# Bounds on $L_p$ Errors in Density Ratio Estimation via $f$-Divergence Loss Functions

**Yoshiaki Kitazawa**
NTT DATA Mathematical Systems Inc.
Data Mining Division
1F Shinanomachi Rengakan, 35, Shinanomachi
Shinjuku-ku, Tokyo, 160-0016, Japan
kitazawa@msi.co.jp

## Abstract

Density ratio estimation (DRE) is a core technique in machine learning used to capture relationships between two probability distributions. $f$-divergence loss functions, which are derived from variational representations of $f$-divergence, have become a standard choice in DRE for achieving cutting-edge performance. This study provides novel theoretical insights into DRE by deriving upper and lower bounds on the $L_p$ errors through $f$-divergence loss functions. These bounds apply to any estimator belonging to a class of Lipschitz continuous estimators, irrespective of the specific $f$-divergence loss function employed. The derived bounds are expressed as a product involving the data dimensionality and the expected value of the density ratio raised to the $p$-th power. Notably, the lower bound includes an exponential term that depends on the Kullback–Leibler (KL) divergence, revealing that the $L_p$ error increases significantly as the KL divergence grows when $p > 1$. This increase becomes even more pronounced as the value of $p$ grows. The theoretical insights are validated through numerical experiments.

## 1 Introduction

Density ratio estimation (DRE) is a key machine learning technique for computing the density ratio $r^*(\mathbf{x}) = q(\mathbf{x})/p(\mathbf{x})$ between two probability distributions, based on samples independently drawn from $p$ and $q$. DRE plays a central role in various machine learning methods, including generative modeling (Goodfellow et al., 2014; Nowozin et al., 2016; Uehara et al., 2016), mutual information estimation and representation learning (Belghazi et al., 2018; Hjelm et al., 2018), energy-based modeling (Gutmann & Hyvärinen, 2010), and covariate shift and domain adaptation (Shimodaira, 2000; Huang et al., 2006).

Recent advancements in DRE have been fueled by approaches employing neural networks as density ratio estimators. These methods use loss functions derived from variational representations of $f$-divergence (Nguyen et al., 2010; Sugiyama et al., 2012), where the optimal function corresponds to the density ratio via the Legendre transform, leading to state-of-the-art performance.

Despite their empirical success, recent research has started to unravel the theoretical connections between optimizing $f$-divergence loss functions and the accuracy of DRE. For integral probability metric (IPM) loss functions, the upper and lower bounds of the $L_p$ error in DRE have been established as the minimax bounds of their optimization (Liang, 2017; Niles-Weed & Berthet, 2022). More recent studies have concentrated on $f$-divergence loss functions, deriving upper bounds (Belomestny et al., 2021) and the minimax upper and lower bounds for optimizing Shannon divergence loss (Belomestny et al., 2021; Puchkin et al., 2024).

Although substantial progress has been made, several aspects of the relationship between the choice of $f$-divergence loss functions and the accuracy of DRE remain unresolved. First, minimax lower bounds do not reflect the true lower bound on the estimation error for the actual density ratio. Second, the relationship between the true magnitudes of the $f$-divergences and the sample size requirements for DRE using divergence loss functions is not fully understood. Specifically, the role of the true

Kullback–Leibler (KL) divergence in determining the sample size needed for DRE with the KL-loss function remains unclear, even though it is known that sample size requirements for KL-divergence estimation increase exponentially as the true value of the divergence grows (Poole et al., 2019; Song & Ermon, 2019; McAllester & Stratos, 2020). Finally, it remains an open question whether different $f$-divergence loss functions, such as total variation loss and KL-divergence loss, yield statistically equivalent $L_p$ errors (e.g., root mean square errors).

This study aims to address uncertainties in DRE using $f$-divergence loss functions by deriving the upper and lower bounds that are independent of the choice of $f$-divergence. However, the theoretical optimization of $f$-divergence loss functions presents challenges due to their dependence on sample sets drawn from two distributions. The lack of overlap between these sample sets often leads to unstable optimization points, resulting in losses that do not reach their theoretical optima. In practice, this issue is commonly mitigated by applying early stopping while monitoring validation losses.

We incorporate this practical approach into our theoretical framework by reformulating the loss functions conceptually, thereby bridging the gap between the practical and theoretical behaviors of these functions. Subsequently, we derive upper and lower bounds for the $L_p$ error in DRE by optimizing $f$-divergence loss functions. These bounds are obtained based on the expectation of the distance between nearest neighbors in observations under the assumptions of $L$-Lipschitz continuity of the energy function of the distributions and the compactness of the support.

The derived bounds are expressed as a product of terms involving the data dimensionality and the expected value of the density ratio raised to the $p$-th power. Notably, the lower bound includes an exponential term dependent on the KL-divergence, showing that the $L_p$ error increases significantly as the KL-divergence grows for $p > 1$, with the rate of increase accelerating for higher values of $p$. These bounds apply to a class of Lipschitz continuous estimators, irrespective of the specific $f$-divergence loss functions used. The theoretical findings are validated through numerical experiments.

In summary, the key contributions of this study are as follows: (1) We derive universal upper and lower bounds for the $L_p$ error in DRE through optimizations of variational representations of $f$-divergences, providing a novel perspective on DRE with $f$-divergence loss functions. (2) We empirically explore the relationship between KL-divergence, data dimensionality, and estimation accuracy in DRE through optimizations of variational representations of $f$-divergences. Specifically, we find that the $L_p$ error increases significantly with larger KL-divergence values for $p > 1$, and this effect is amplified by the magnitude of $p$.

**Related Work.** This study establishes upper and lower bounds on convergence rates for nonparametric density ratio estimation using $f$-divergence optimization. Relevant prior research includes studies on the minimax convergence rates for density estimation in the context of GAN optimization, particularly for Wasserstein GANs (Arjovsky & Bottou, 2017) and vanilla GANs (Goodfellow et al., 2014). In Wasserstein GAN optimization, Liang (2017) and Singh & Póczos (2018) derived the minimax convergence rates for the IPM loss, which encompasses total variation among $f$-divergences. Furthermore, Niles-Weed & Berthet (2022) extended these findings to the Wasserstein-$p$ distance for $p > 1$. In the context of vanilla GAN optimization, Belomestny et al. (2021) and Puchkin et al. (2024) provided minimax upper and lower convergence rates for the Shannon divergence loss, specifically deriving an upper bound for the $L_2$ error. Beyond GAN-based research, Nguyen et al. (2010) proposed an upper bound for the Hellinger distance in DRE using the KL-divergence loss, thereby establishing a minimax upper bound for the $L_1$ error in DRE. Additionally, foundational work by Stone (1980) introduced a minimax convergence rate for nonparametric regression, which is also applicable as an upper bound for the $L_1$ error in nonparametric density estimation. For detailed comparisons between our derived bounds and existing DRE bounds, see Section E.1 in Appendix.

## 2 PRELIMINARIES: NOTATION, SETUP, AND $f$-DIVERGENCE LOSS FUNCTIONS

In this section, we define the notation, outline the problem setup, and present the variational representation of $f$-divergence along with the associated loss functions that form the foundation of the analysis in the subsequent sections.

## 2.1 Notation, Preliminary Concepts, and Setup

**Notation:** Random variables are represented by uppercase letters, such as $X$. Lowercase letters denote specific values of these random variables; for example, $x$ represents a value of the random variable $X$. Boldface letters, $\mathbf{X}$ and $\mathbf{x}$, are used to denote sets of random variables and their corresponding values, respectively. $\|\mathbf{y} - \mathbf{x}\|_\infty$ denotes the maximum norm in $\mathbf{R}^d$. i.e., $\|\mathbf{y} - \mathbf{x}\|_\infty = \max_{1 \le i \le d} |y_i - x_i|$ for $\mathbf{y} = (y_1, y_2, \ldots, y_d)$ and $\mathbf{x} = (x_1, x_2, \ldots, x_d)$. $\operatorname{diam}(\Omega)$ denotes the diameter of $\bar{\Omega}$. Specifically, let $\operatorname{diam}(\Omega) = \inf\{r > 0 \mid \exists \mathbf{a} \in \Omega \text{ s.t. } \Omega \subseteq \Delta(\mathbf{a}, r)\}$, where $\Delta(\mathbf{a}, r)$ denotes the $d$-dimensional interval centered at $\mathbf{a}$ with each side of length $r$: $\Delta(\mathbf{a}, r) = \{\mathbf{x} \in \mathbb{R}^d \mid \|\mathbf{x} - \mathbf{a}\|_\infty < r/2\}$. $O_p(a_N)$ denotes stochastic boundedness with rate $a_N$ in $\mu$. i.e., $\mathbf{X} = O_p(a_N)$ (as $N \to \infty$) $\Leftrightarrow$ for all $\varepsilon > 0$, there exist $\delta(\varepsilon) > 0$ and $N(\varepsilon) > 0$ such that $\mu\left(|\mathbf{X}|/a_N \ge \delta(\varepsilon)\right) < \varepsilon$ for all $N \ge N(\varepsilon)$.

**Preliminary Concepts:** Let $P$ and $Q$ represent probability measures on $(\Omega, \mathscr{F})$, where $\mathscr{F}$ denotes the $\sigma$-algebra on $\Omega$. $P$ is called *absolutely continuous* with respect to $Q$, $P(A) = 0$ whenever $Q(A) = 0$ for any $A \in \mathscr{F}$. This relationship is denoted as $P \ll Q$. $\frac{dP}{dQ}$ refers to the Radon–Nikodým derivative of $P$ with respect to $Q$ for $P$ and $Q$ with $P \ll Q$. $\mu$ denotes a probability measure on $\Omega$ with $P \ll \mu$ and $Q \ll \mu$. An example of $\mu$ is $(P + Q)/2$. $E_P[\cdot]$ denotes the expectation under the distribution $P$, i.e., $E_P[\phi(\mathbf{x})] = \int_{\Omega_p} \phi(\mathbf{x}) dP(\mathbf{x})$, where $\phi(\mathbf{x})$ is a measurable function over $\Omega$.

**Setup and Assumptions:** $P$ and $Q$ represent probability distributions on $\Omega \subset \mathbb{R}^d$ with unknown probability densities $p$ and $q$, respectively. We assume $p(\mathbf{x}) > 0 \Leftrightarrow q(\mathbf{x}) > 0$ almost everywhere $\mathbf{x} \in \Omega$. [1]

## 2.2 DRE with $f$-divergence variational representation

Here, we introduce the $f$-divergence variational representation and the corresponding loss functions used for DRE.

**Definition 2.1** ($f$-divergence)**.** The $f$-divergence $D_f$ between two probability measures $P$ and $Q$ is defined using a convex function $f$ satisfying $f(1) = 0$. It is expressed as: $D_f(Q||P) = E_P[f(dQ/dP(\mathbf{x}))]$.

Various divergences can be obtained as special cases by selecting an appropriate generator function $f$. For instance, choosing the function $f(u) = u \cdot \log u$ yields the Kullback–Leibler divergence.

We derive the variational representations of $f$-divergences using the Legendre transform of the convex conjugate of a twice differentiable convex function $f$, $f^*(\psi) = \sup_{u \in \mathbb{R}} \{\psi \cdot u - f(u)\}$ (Nguyen et al., 2007):

$$D_f(Q||P) = \sup_{\phi \ge 0} \left\{ E_Q\left[f'(\phi)\right] - E_P\left[f^*(f'(\phi))\right] \right\}, \tag{1}$$

where the supremum is required over all measurable functions $\phi : \Omega \to \mathbb{R}$ with $E_Q[|f'(\phi)|] < \infty$ and $E_P[|f^*(f'(\phi))|] < \infty$. The maximum value is achieved when $\phi(\mathbf{x}) = dQ/dP(\mathbf{x})$. Pairs of the terms $f'(\phi)$ and $f^*(f'(\phi))$ in Equation (1) for major $f$-divergences, along with their corresponding convex functions $f$, are provided in Table 2 in the Appendix.

By substituting $\phi$ with a neural network model $\phi_\theta$ and replacing the expectation $E$ with sample means $\hat{E}$, the optimal function for Equation (1) is trained through back-propagation using an $f$-divergence loss function.

$$\mathcal{L}_f(\phi_\theta) = -\left\{ \hat{E}_Q\left[f'(\phi_\theta)\right] - \hat{E}_P\left[f^*(f'(\phi_\theta))\right] \right\}. \tag{2}$$

Formally, we define the $f$-divergence loss function within a probabilistic theoretical framework as follows:

**Definition 2.2** ($f$-Divergence Loss)**.** Let $\hat{\mathbf{X}}_{P[R]} = \{\mathbf{X}_P^1, \mathbf{X}_P^2, \ldots, \mathbf{X}_P^R\}$, $\mathbf{X}_P^i \overset{\text{iid}}{\sim} P$ denote $R$ i.i.d. random variables from $P$, and let $\hat{\mathbf{X}}_{Q[S]} = \{\mathbf{X}_Q^1, \mathbf{X}_Q^2, \ldots, \mathbf{X}_Q^S\}$, $\mathbf{X}_Q^i \overset{\text{iid}}{\sim} Q$ denote $S$ i.i.d. random variables from $Q$. Thereafter, for a twice differentiable convex function $f$, $f$-divergence loss $\mathcal{L}_f^{(R,S)}(\cdot)$

---

[1] In this study, $q(\mathbf{x})/p(\mathbf{x})$ is used as a notation for $\frac{dQ}{dP}(\mathbf{x})$ based on the Radon–Nikodým density representation for improved readability.

is defined as follows:

$$\mathcal{L}_f^{(R,S)}(\phi) = \frac{1}{S} \cdot \sum_{i=1}^{S} -f' \left( \phi(\mathbf{X}_Q^i) \right) + \frac{1}{R} \sum_{i=1}^{R} f^* \left( f' \left( \phi(\mathbf{X}_P^i) \right) \right), \tag{3}$$

where $\phi$ denotes a measurable function over $\Omega$ such that $\phi : \Omega \to \mathbb{R}_{>0}$.

## 3 MAIN RESULTS

This study presents two key contributions. First, we derive common upper and lower bounds for the $L_p$ error in DRE by employing variational $f$-divergence optimization. Second, we empirically explore the relationship between KL-divergence, data dimensionality, and estimation accuracy in DRE through variational $f$-divergence optimization. Specifically, we find that the $L_p$ error increases significantly as the KL-divergence rises for $p > 1$, and this increase becomes more pronounced with larger values of $p$.

### 3.1 THEORETICAL RESULTS.

In this study, we outline the assumptions required to derive the upper and lower bounds for DRE. These assumptions are straightforward and primarily pertain to Lipschitz continuity of estimators. Specifically, we assume the $L$-Lipschitz continuity of the energy function of the distributions, $T^*(\mathbf{x}) = -\log dQ/dP(\mathbf{x})$.

**Assumption 3.1** (Assumption for the Upper Bound)**.** The following assumption is imposed on the probability distributions $P$ and $Q$.

    U1. $T^*(\mathbf{x}) = -\log dQ/dP(\mathbf{x})$ is $L$-Lipschitz continuous with $L > 0$ on $\Omega$.

**Assumption 3.2** (Assumptions for the Lower Bound)**.** The following assumptions are imposed on the probability distributions $P$ and $Q$.

    L1. $T^*(\mathbf{x}) = -\log dQ/dP(\mathbf{x})$ is $L$-bi-Lipschitz continuous with $L > 1$ on $\Omega$.

    L2. $E_P \left[ (dQ/dP)^p \right] < \infty$ where $p \leq d$.

For the probability distributions $P$ and $Q$, Assumption L1 plays a crucial role in deriving the lower bound of the $L_p$ error in DRE. Further details regarding this assumption are provided in Remark 4.6 in Section 4.2.

Additionally, Assumptions 3.3 and 3.4 are essential for deriving both the upper and lower bounds of the DRE. A discussion comparing Assumption 3.3 with related assumptions in prior work is provided in Section E.2 in Appendix.

**Assumption 3.3** (Assumptions for the Convex Function $f$)**.** The convex function $f$ is assumed to satisfy the following conditions: (F1) $f$ is three-times differentiable; (F2) $f''(u) > 0$ for all $u > 0$; and (F3) $E_P \left[ f''(dQ/dP) \right] < \infty$.

**Assumption 3.4** (Assumption for the Support)**.** The support $\Omega$ is assumed to satisfy the following conditions: (O1) $\text{diam}(\Omega) < \infty$.

Under these conditions, we derive the upper and lower bounds for the $L_p$ error in DRE using variational $f$-divergence optimization.

**Theorem 3.5** (Informal. See Theorem 4.5 and 4.8)**.** *Assume $\Omega$ is a compact set in $\mathbb{R}^d$, where $d \geq 3$, and $f$ satisfies Assumption 3.3. Let $P$ and $Q$ denote the probability measures on $\Omega$, and let $\phi$ represent a $K$-Lipschitz function that minimizes the $f$-divergence loss functions defined in Equation (3) using early stopping. Additionally, let $N = \min\{R, S\}$.*

*(Upper Bound) Assume Assumption 3.1: Thereafter, Equation (4) holds for $1 \leq p \leq d/2$ such that*

$$\left\| \frac{q(\mathbf{x})}{p(\mathbf{x})} - \phi(\mathbf{x}) \right\|_{L_p(\Omega, P)} \lesssim \frac{\text{diam}(\Omega)}{N^{1/d}} \cdot \left\{ L \cdot E \left[ \left( \frac{dQ}{dP} \right)^{2 \cdot p} \right]^{1/(2 \cdot p)} + K \right\}. \tag{4}$$

***(Lower Bound)*** *Assume Assumption 3.2: Equations (5) and (6) hold for $1 \leq p \leq d$ such that*

$$E_{\mathbf{X}_P^1 \cdots \mathbf{X}_P^N} \left[ \left\| \frac{q(\mathbf{x})}{p(\mathbf{x})} - \phi(\mathbf{x}) \right\|_{L_p(\Omega, P)} \right] \gtrsim \frac{1}{N^{1/d}} \cdot \left\{ \frac{1}{L} \cdot \left\{ E_P \left[ \left\{ \frac{dQ}{dP}(\mathbf{x}) \right\}^p \right] \right\}^{1/p} - K \cdot \mathrm{diam}(\Omega) \right\}$$

$$\tag{5}$$

$$\gtrsim \frac{1}{N^{1/d}} \cdot \left\{ \frac{1}{L} \cdot e^{\frac{(p-1)}{p} \cdot KL(P\|Q)-1} - K \cdot \mathrm{diam}(\Omega) \right\},$$

$$\tag{6}$$

*where $|\cdot|_{L_p(\Omega, P)}$ denotes the $L_p$ norm on $\Omega$ with respect to measure $P$, and $KL(P\|Q)$ denotes the KL-divergence between $P$ and $Q$.*

These bounds apply to all $K$-Lipschitz continuous estimators optimized with early stopping using the $f$-divergence loss functions, as discussed in Section 4.3 and supported by Theorem 4.8.

Theorem 3.5 indicates that the curse of dimensionality arises when $p = 1$. For $p > 1$, both the curse of dimensionality and the large sample requirements for high KL-divergence data occur simultaneously. In particular, Equation (6) shows that the $L_p$ error increases exponentially with growing KL-divergence for $p > 1$, and this growth accelerates as $p$ increases. These theoretical insights are validated by numerical experiments, which are presented in the following section.

## 3.2 Experimental Results.

We empirically verified the relationship between KL-divergence and data dimensionality, and their effects on the estimation accuracy of DRE through variational $f$-divergence optimization. The results, which corroborate the implications of Theorem 3.5, are presented in detail in Section D of the Appendix.

$L_p$ **Errors vs. the KL-Divergence in Data**   We conducted experiments to investigate the relationship between $L_1$, $L_2$, and $L_3$ errors in DRE and the KL-divergence of the data. In these experiments, we generated 100 sets of 5-dimensional datasets with KL-divergence values of 1, 2, 4, 6, 8, 10, 12, and 14. For each dataset, DRE was performed using the $\alpha$-divergence and KL-divergence loss functions, and the resulting $L_1$, $L_2$, and $L_3$ errors were recorded. The results are presented in Figure 1. Details of the experimental settings and neural network training procedures are provided in Section D.

As shown in Figure 1, the estimation errors for $p > 1$ increased significantly, with this growth accelerating as $p$ becomes larger. In contrast, when $p = 1$, the increase in estimation error was relatively mild. Consistent with Theorem 3.5, these results demonstrate the significant impact of KL-divergence on $L_p$ errors for $p > 1$ in DRE with $f$-divergence loss functions.

$L_p$ **Errors vs. the Dimensions of Data**   We conducted experiments to investigate the relationship between $L_1$, $L_2$, and $L_3$ errors in DRE and the dimensionality of the data. In the experiments, we generated 100 datasets with dimensions of 50, 100, and 200, each having a KL-divergence value of 3. For each dataset, DRE was performed using $\alpha$-divergence and KL-divergence loss functions, and the resulting $L_1$, $L_2$, and $L_3$ errors were observed. The results are presented in Figure 2. Details of the experimental settings and neural network training procedures are provided in Section D.

As depicted in Figure 2, the estimation errors $L_1$, $L_2$, and $L_3$ increased as the dimensionality of the data increased for both the $\alpha$-divergence and KL-divergence loss functions. These results indicate that the curse of dimensionality affects all $L_p$ errors equally, as suggested by Theorem 3.5.

## 4 Overview of Upper and Lower Bound Derivations

In this section, we outline the derivation of the upper and lower bounds. We begin by introducing a conceptual reformulation of the $f$-divergence loss function, which serves as the foundation of our theoretical framework. Next, we derive the upper and lower bounds for DRE in terms of the $L_p$ error, based on this reformulation. Finally, we extend these results to the practical optimization scenario using the $f$-divergence loss function, incorporating early stopping by monitoring validation losses.

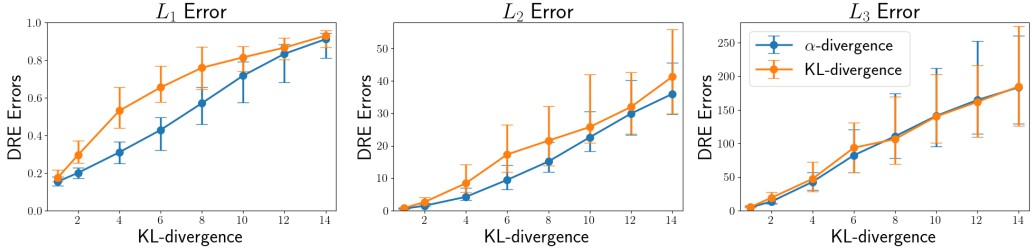

Figure 1: The experimental results of $L_p$ errors versus the magnitude of KL-divergence in the data are presented in Section 3.2. The $x$-axis represents the magnitude of KL-divergence in synthetic datasets of fixed dimensionality. The $y$-axes of the left, center, and right graphs correspond to the $L_1$, $L_2$, and $L_3$ errors in DRE, respectively. The plots depict the median values of the $y$-axis, while the error bars indicate the interquartile range (25th to 75th percentiles). The blue line represents errors computed using the $\alpha$-divergence loss function, whereas the orange line corresponds to errors computed using the KL-divergence loss function.

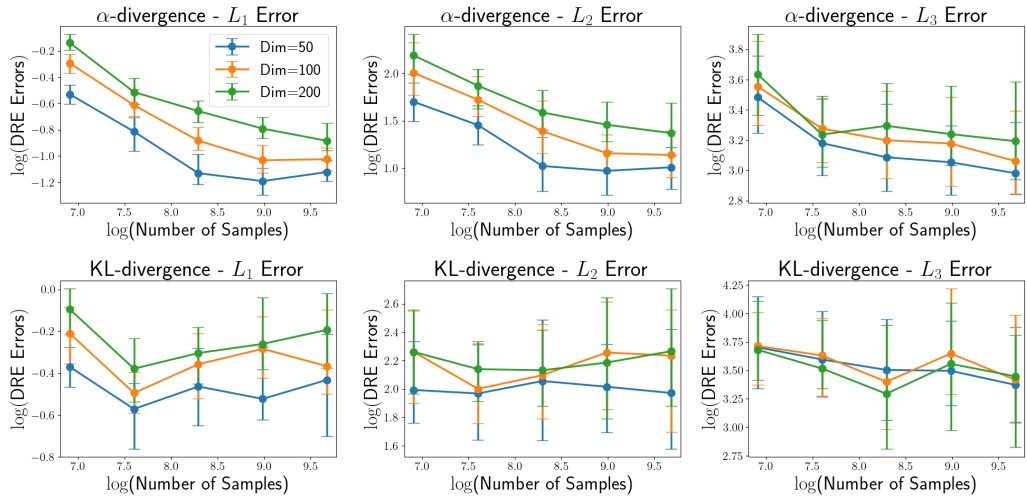

Figure 2: The experimental results on $L_p$ errors versus the dimensionality of the data are presented in Section 3.2. The top row displays results using the $\alpha$-divergence loss function, whereas the bottom row presents results using the KL-divergence loss function. The $x$-axis represents the logarithm of the number of samples utilized in the optimizations of DRE. The $y$-axes of the left, center, and right graphs correspond to the $L_1$, $L_2$, and $L_3$ errors in DRE, respectively. The plots show the median $y$-axis values, while the error bars represent the interquartile range (25th to 75th percentiles). The blue, orange, and green lines correspond to data dimensions of 50, 100, and 200, respectively.

This extension represents the core theoretical contribution of this study. Detailed statements and proofs for the theorems discussed in this section are provided in Section C of the Appendix.

## 4.1 CONCEPTUAL REFORMULATION OF THE $f$-DIVERGENCE LOSS FUNCTIONS

The optimization of $f$-divergence loss functions, denoted as $\mathcal{L}_f^{(R,S)}(\phi)$ in Equation (3), poses both practical and theoretical challenges due to its propensity to overfit the training data.

To better understand this issue, consider a deterministic setting as described in Definition 2.2, where $(\mathbf{x}_P^1, \mathbf{x}_P^2, \ldots, \mathbf{x}_P^R) = (1, 2, \ldots, R)$ and $(\mathbf{x}_Q^1, \mathbf{x}_Q^2, \ldots, \mathbf{x}_Q^S) = (R+1, R+2, \ldots, R+S)$. Notably, $\{\mathbf{x}_P^i\}_{i=1}^R \cap \{\mathbf{x}_Q^i\}_{i=1}^S = \emptyset$. In this setup, we observe that $\hat{\mathcal{L}}_f^{(R,S)}(\phi) \to -\infty$ as $f^*\big(f'\big(\phi(\mathbf{x}_P^i)\big)\big) \to -\infty$ and $-f'\big(\phi(\mathbf{x}_Q^j)\big) \to -\infty$ for all $1 \le i \le R$ and $1 \le j \le S$. In practice, this issue is mitigated by implementing early stopping, where validation losses are monitored during optimization. The

present theoretical framework incorporates this practical strategy, allowing for a deeper analysis of both the optimization process and its implications for downstream tasks such as DRE.

To bridge the gap between the practical and theoretical behaviors of $f$-divergence loss functions within our framework, we propose a conceptual reformulation of the loss function.

**Definition 4.1** ($\mu$-Representation $f$-Divergence Loss). Let $\mu$ be a probability measure with $P \ll \mu$ and $Q \ll \mu$. Let $N = \min\{R, S\}$ and let $\hat{\mathbf{X}}_{\mu[N]} = \{\mathbf{X}_\mu^1, \ldots, \mathbf{X}_\mu^N\}$ denote $N$ i.i.d. random variables from $\mu$. For a twice differentiable convex function $f$, let

$$\widetilde{l}_f(u; \mathbf{x}) = -f'(u) \cdot \frac{dQ}{d\mu}(\mathbf{x}) + f^*(f'(u)) \cdot \frac{dP}{d\mu}(\mathbf{x}), \tag{7}$$

where $f^*$ denotes the Legendre transform of $f$: $f^*(\psi) = \sup_{u \in \mathbb{R}}\{\psi \cdot u - f(u)\}$. The $\mu$-representation of the $f$-divergence loss $\mathcal{L}_f^{(R,S)}(\cdot)$ in Equation (3) at the points $\hat{\mathbf{X}}_{\mu[N]}$ is defined as

$$\widetilde{\mathcal{L}}_f^{(N)}(\phi) \quad = \quad \frac{1}{N} \cdot \sum_{i=1}^{N} \widetilde{l}_f(\phi; \mathbf{X}_\mu^i), \tag{8}$$

where $\phi$ is a measurable function over $\Omega$ such that $\phi : \Omega \to \mathbb{R}_{>0}$.

This representation introduces an error of $1/\sqrt{N}$ between the practical $f$-divergence loss function $\mathcal{L}_f^{(R,S)}(\phi)$ and the $\mu$-representation $f$-divergence loss $\widetilde{\mathcal{L}}_f^{(N)}(\phi)$. However, this error is negligible when $d \geq 3$, which will be discussed in Section 4.3.

The optimization properties of this conceptual loss function are encapsulated in Proposition 4.2.

**Proposition 4.2.** *Assume that $f$ satisfies Assumption 3.3. Let $\phi_* = \arg\min_{\phi:\Omega\to\mathbb{R}_{>0}} \widetilde{\mathcal{L}}_f^{(N)}(\phi)$. Then,* $\phi_*(\mathbf{X}_\mu^i) = \frac{dQ}{dP}(\mathbf{X}_\mu^i)$, *for $i = 1, 2, \ldots, N$.*

This reformulation ensures that the conceptual loss function remains bounded. Furthermore, all optimal points of the conceptual loss function are aligned with the ideal density ratios.

## 4.2 DERIVATION OF UPPER AND LOWER BOUNDS FOR OPTIMAL FUNCTIONS OF THE $\mu$-REPRESENTATION $f$-DIVERGENCE LOSS FUNCTIONS

In this section, we derive the upper and lower bounds for the $L_p$ error in DRE for the optimal function of $\mathcal{L}_f^{(N)}(\cdot)$ as defined in the previous section. These bounds are based on the expected distance between the nearest neighbors of each $\mathbf{X}_\mu^i$, for $1 \leq i \leq N$.

Hereafter, $\mathbf{X}_{\mu[N]}^{(1)}(\mathbf{x})$ denotes the nearest neighbor of $\mathbf{x}$ in $\hat{\mathbf{X}}_{\mu[N]} = \{\mathbf{X}_\mu^1, \ldots, \mathbf{X}_\mu^N\}$. Specifically, define $\mathbf{X}_{\mu[N]}^{(1)}(\mathbf{x})$ as $\mathbf{X}_\mu^i$ in $\hat{\mathbf{X}}_{\mu[N]}$ such that $\|\mathbf{X}_\mu^l - \mathbf{x}\|_\infty > \|\mathbf{X}_\mu^i - \mathbf{x}\|_\infty$, for all $l < i$, and $\|\mathbf{X}_\mu^u - \mathbf{x}\|_\infty \geq \|\mathbf{X}_\mu^i - \mathbf{x}\|_\infty$ for all $u > i$. As in the previous section, let $\phi_* = \arg\min_{\phi:\Omega\to\mathbb{R}_{>0}} \widetilde{\mathcal{L}}_f^{(N)}(\phi)$.

As presented in Proposition 4.2, the optimal points of the $\mu$-representation $f$-divergence loss functions $\widetilde{\mathcal{L}}_f^{(N)}(\phi)$ coincide with the ideal density ratios. This fact provides the following equation, serving as the key bridge between the density ratio and its estimation.

$$\phi_*(\mathbf{X}_\mu^i) = \frac{dQ}{dP}(\mathbf{X}_\mu^i) = \frac{dQ}{dP}\big(\mathbf{X}_{\mu[N]}^{(1)}(\mathbf{X}_\mu^i)\big). \tag{9}$$

Based on this equation, we can obtain

$$\left|\phi_*\big(\mathbf{X}_{\mu[N]}^{(1)}(\mathbf{x})\big) - \phi_*(\mathbf{x})\right|^p = \left|\frac{dQ}{dP}\big(\mathbf{X}_{\mu[N]}^{(1)}(\mathbf{x})\big) - \phi_*(\mathbf{x})\right|^p. \tag{10}$$

Using the triangle inequality in the $L_p$ norm for the density ratios at $\mathbf{x}$ and its nearest neighbor, we obtain

$$\left\{E_P \left|\frac{dQ}{dP}(\mathbf{x}) - \frac{dQ}{dP}\big(\mathbf{X}_{\mu[N]}^{(1)}(\mathbf{x})\big)\right|^p\right\}^{1/p} - \left\{E_P \left|\frac{dQ}{dP}\big(\mathbf{X}_{\mu[N]}^{(1)}(\mathbf{x})\big) - \phi_*(\mathbf{x})\right|^p\right\}^{1/p}$$

$$\leq \left\{ E_P \left| \frac{dQ}{dP}(\mathbf{x}) - \phi_*(\mathbf{x}) \right|^p \right\}^{1/p}$$

$$\leq \left\{ E_P \left| \frac{dQ}{dP}(\mathbf{x}) - \frac{dQ}{dP}(\mathbf{X}_{\mu[N]}^{(1)}(\mathbf{x})) \right|^p \right\}^{1/p} + \left\{ E_P \left| \frac{dQ}{dP}(\mathbf{X}_{\mu[N]}^{(1)}(\mathbf{x})) - \phi_*(\mathbf{x}) \right|^p \right\}^{1/p}. \quad (11)$$

Assuming the $L$-bi-Lipschitz continuity of the energy function of the density ratio, $T^*(\mathbf{x}) = -\log q(\mathbf{x})/p(\mathbf{x})$, we yield

$$\frac{1}{L^p} \cdot \left( \frac{dQ}{dP}(\mathbf{x}) \right)^p \left\| \mathbf{X}_{\mu[N]}^{(1)}(\mathbf{x}) - \mathbf{x} \right\|_\infty^p + O_p \left( \frac{1}{N^{1/(2d)}} \right)$$

$$\leq \left| \frac{dQ}{dP}(\mathbf{x}) - \frac{dQ}{dP}(\mathbf{X}_{\mu[N]}^{(1)}(\mathbf{x})) \right|^p$$

$$\leq L^p \cdot \left( \frac{dQ}{dP}(\mathbf{x}) \right)^p \left\| \mathbf{X}_{\mu[N]}^{(1)}(\mathbf{x}) - \mathbf{x} \right\|_\infty^p + O_p \left( \frac{1}{N^{1/(2d)}} \right). \quad (12)$$

Additionally, from the $K$-Lipschitz continuity of $\phi_*(\cdot)$ and Equation (9),

$$\left| \frac{dQ}{dP}(\mathbf{X}_{\mu[N]}^{(1)}(\mathbf{x})) - \phi_*(\mathbf{x}) \right|^p = \left| \phi_*(\mathbf{X}_{\mu[N]}^{(1)}(\mathbf{x})) - \phi_*(\mathbf{x}) \right|^p \leq K^p \cdot \left\| \mathbf{X}_{\mu[N]}^{(1)}(\mathbf{x}) - \mathbf{x} \right\|_\infty^p. \quad (13)$$

Equations (12) and (13) provide the upper and lower bounds of the difference in density ratios between $\mathbf{x}$ and its nearest neighbor $\mathbf{X}_{\mu[N]}^{(1)}(\mathbf{x})$ using their distance.

To evaluate the expectation of the distance between $\mathbf{x}$ and its nearest neighbor $\mathbf{X}_{\mu[N]}^{(1)}(\mathbf{x})$, we present the following theorems: Theorem 4.3 provides an upper bound for the expectation on the right-hand side of Equation (12); and Theorem 4.4 establishes a lower bound for the expectation on the left-hand side.

**Theorem 4.3.** *Under the assumption that $\Omega$ is compact, for $1 \leq p \leq d/2$,*

$$\varlimsup_{N \to \infty} N^{1/d} \cdot \left\{ E_P \left[ \left\{ \frac{dQ}{dP}(\mathbf{x}) \right\}^p \cdot \left\| \mathbf{X}_{P[N]}^{(1)}(\mathbf{x}) - \mathbf{x} \right\|_\infty^p \right] \right\}^{1/p}$$

$$\leq \mathrm{diam}(\Omega) \cdot \left( E_P \left[ \left\{ \frac{dQ}{dP}(\mathbf{x}) \right\}^{2 \cdot p} \right] \right)^{1/(2 \cdot p)}. \quad (14)$$

**Theorem 4.4.** *Let $P$ and $Q$ be probability measures on a compact set $\Omega$ in $\mathbb{R}^d$ with $d \geq 1$. Assume that $P \ll \lambda$ and $Q \ll \lambda$, where $\lambda$ denotes the Lebesgue measure on $\mathbb{R}^d$. Let $p$ be a positive constant such that $p \geq 1$. Assume $E[(dQ/dP)^p] < \infty$. Then,*

$$\lim_{N \to \infty} N^{1/d} \cdot \left\{ E_{\hat{\mathbf{X}}_{P[N]}} \left[ E_P \left[ \left\{ \frac{dQ}{dP}\left(\mathbf{X}_{P[N]}^{(1)}(\mathbf{x})\right) \right\}^p \cdot \left\| \mathbf{X}_{P[N]}^{(1)}(\mathbf{x}) - \mathbf{x} \right\|_\infty^p \right] \right] \right\}^{1/p}$$

$$\geq e^{-1} \cdot \left\{ E_P \left[ \left\{ \frac{dQ}{dP}(\mathbf{x}) \right\}^p \right] \right\}^{1/p}, \quad (15)$$

*where $E_{\hat{\mathbf{X}}_{P[N]}}[\cdot]$ denotes the expectation over each variable in $\hat{\mathbf{X}}_{P[N]} = \{\mathbf{X}_P^1, \mathbf{X}_P^2, \ldots, \mathbf{X}_P^N\}$.*

Notably, using Jensen's inequality on the right-hand side of Equation (15) in Theorem 4.4, the KL-divergence between $P$ and $Q$ appears in the lower bound such that

$$e^{-1} \cdot \left\{ E_P \left[ \left\{ \frac{dQ}{dP}(\mathbf{x}) \right\}^p \right] \right\}^{1/p} = e^{-1} \cdot \left\{ E_Q \left[ \left\{ \frac{dQ}{dP}(\mathbf{x}) \right\}^{p-1} \right] \right\}^{1/p}$$

$$= e^{-1} \cdot \left\{ E_Q \left[ e^{(p-1) \cdot \log \frac{dQ}{dP}(\mathbf{x})} \right] \right\}^{1/p}$$

$$\geq e^{-1} \cdot \left\{ e^{E_Q \left[ (p-1) \cdot \log \frac{dQ}{dP}(\mathbf{x}) \right]} \right\}^{1/p} = e^{\frac{p-1}{p} \cdot KL(Q \| P) - 1}. \quad (16)$$

We derive the upper and lower bounds for the $L_p$ error in DRE for the optimally estimated functions $\widetilde{\mathcal{L}}_f^{(N)}(\cdot)$, as stated in Theorem 4.5.

**Theorem 4.5.** *Assume $\Omega$ is a compact set in $\mathbb{R}^d$ with $d \geq 3$, and that $f$ satisfies Assumption 3.3. Let $P$ and $Q$ be probability measures on $\Omega$, assuming that $P \ll \lambda$ and $Q \ll \lambda$, where $\lambda$ denotes the Lebesgue measure on $\mathbb{R}^d$. Let $T^*(\mathbf{x})$ be the energy function of $dQ/dP(\mathbf{x})$ defined as $T^*(\mathbf{x}) = -\log dQ/dP(\mathbf{x})$. Let $\widetilde{\mathcal{F}}_{K\text{-}Lip}^{(N)}$ denote the set of all $K$-Lipschitz continuous functions on $\Omega$ that minimize $\widetilde{\mathcal{L}}_f^{(N)}(\cdot)$. Specifically, define*

$$\widetilde{\mathcal{F}}^{(N)} = \left\{ \phi_* : \Omega \to \mathbb{R}_{>0} \,\Big|\, \widetilde{\mathcal{L}}_f^{(N)}(\phi_*) = \min_\phi \widetilde{\mathcal{L}}_f^{(N)}(\phi) \right\}, \tag{17}$$

*and*

$$\mathcal{F}_{K\text{-}Lip} = \left\{ \phi : \Omega \to \mathbb{R}_{>0} \,\Big|\, |\phi(\mathbf{y}) - \phi(\mathbf{x})| \leq K \cdot \|\mathbf{y} - \mathbf{x}\|_\infty \text{ for all } \mathbf{y}, \mathbf{x} \in \Omega \right\}. \tag{18}$$

*Subsequently, let $\widetilde{\mathcal{F}}_{K\text{-}Lip}^{(N)} = \widetilde{\mathcal{F}}^{(N)} \cap \mathcal{F}_{K\text{-}Lip}$.*

*(**Upper Bound**) Assume that $T^*(\mathbf{x})$ satisfies Assumption 3.1. Thereafter, for $1 \leq p \leq d/2$, Equation (19) holds for any $\phi \in \widetilde{\mathcal{F}}_{K\text{-}Lip}^{(N)}$ such that*

$$\varlimsup_{N \to \infty} N^{1/d} \cdot \left\{ E_P \left| \frac{dQ}{dP}(\mathbf{x}) - \phi(\mathbf{x}) \right|^p \right\}^{1/p}$$

$$\leq L \cdot \text{diam}(\Omega) \cdot \left\{ E_P \left[ \left\{ \frac{dQ}{dP}(\mathbf{x}) \right\}^{2 \cdot p} \right] \right\}^{1/(2 \cdot p)} + K \cdot \text{diam}(\Omega). \tag{19}$$

*(**Lower Bound**) Assume that $T^*(\mathbf{x})$ satisfies Assumption 3.2. Then, Equations (20) and (21) hold for any $\phi \in \widetilde{\mathcal{F}}_{K\text{-}Lip}^{(N)}$ such that*

$$\varliminf_{N \to \infty} N^{1/d} \cdot E_{\hat{\mathbf{X}}_{P[N]}} \left[ \left\{ E_P \left| \frac{dQ}{dP}(\mathbf{x}) - \phi(\mathbf{x}) \right|^p \right\}^{1/p} \right]$$

$$\geq \frac{1}{L} \cdot \left\{ E_P \left[ \left\{ \frac{dQ}{dP}(\mathbf{x}) \right\}^p \right] \right\}^{1/p} - K \cdot \text{diam}(\Omega) \tag{20}$$

$$\geq \frac{1}{L} \cdot e^{\frac{p-1}{p} \cdot KL(Q\|P) - 1} - K \cdot \text{diam}(\Omega) \tag{21}$$

*Remark* 4.6. In the case $L = 1$, Equation (12) with $p = 1$ implies that $|dQ/dP(\mathbf{y}) - dQ/dP(\mathbf{x})| = dQ/dP(\mathbf{x}) \cdot \|\mathbf{y} - \mathbf{x}\|_\infty$, for all $\mathbf{x}, \mathbf{y} \in \Omega$. A typical case is when $dQ/dP(x_1, x_2, \ldots, x_d) \equiv dQ/dP(x_1, x_2, \ldots, x_{d'})$ with $d' < d$. Thus, this typically occurs when $dQ/dP(\mathbf{x})$ is a replication of its lower-dimensional distribution. In such cases, the upper and lower bounds for the $L_p$ error in DRE are determined by the lower-dimensional distribution.

## 4.3 DERIVATION OF UPPER AND LOWER BOUNDS FOR OPTIMAL FUNCTIONS OF THE $f$-DIVERGENCE LOSS FUNCTIONS

To establish upper and lower bounds for practical DRE using $f$-divergence loss function optimization, we first statistically evaluate the discrepancy between the outputs of the practically optimized functions $\mathcal{L}_f^{(R,S)}(\cdot)$, which employ early stopping based on validation losses, and the theoretically optimized functions $\widetilde{\mathcal{L}}_f^{(N)}(\cdot)$. Next, we demonstrate that this discrepancy becomes negligible when $d \geq 3$. Finally, the upper and lower bounds for DRE are expressed in terms of the $L_p$ error for $f$-divergence loss function optimization with early stopping. This result constitutes the final theoretical contribution of this study.

First, according to the central limit theorem, an error of order $1/\sqrt{N}$ in probability arises when measuring validation losses.

$$\mathcal{L}_f^{(R,S)}(\phi) - E_\mu \left[ \mathcal{L}_f^{(R,S)}(\phi) \right] = O_p \left( \frac{1}{\sqrt{N}} \right). \tag{22}$$

Equation (22) implies that there is an error margin of $O_p\left(\frac{1}{\sqrt{N}}\right)$ when monitoring validation losses for early stopping in the optimization of $\mathcal{L}_f^{(R,S)}(\phi)$.

Subsequently, we use the following theorem to demonstrate that the optimization of Equation (22), employing early stopping based on validation losses, is governed by the optimization of the $\mu$-representation $f$-divergence loss functions $\widetilde{\mathcal{L}}_f^{(N)}(\cdot)$.

**Theorem 4.7.** *Assume the same assumptions as in Proposition 4.2. Let $\phi_* = \arg\min_{\phi:\Omega\to\mathbb{R}_{>0}} \widetilde{\mathcal{L}}_f^{(N)}(\phi)$. Therefore, for any measurable function $\phi:\Omega\to\mathbb{R}_{>0}$,*

$$\phi(\mathbf{X}_\mu^i) - \phi_*(\mathbf{X}_\mu^i) = O_p\left(\frac{1}{\sqrt{N}}\right), \quad for\ 1 \le i \le N,$$

$$\iff \mathcal{L}_f^{(R,S)}(\phi) - \min_{\phi:\Omega\to\mathbb{R}_{>0}} E_\mu\left[\mathcal{L}_f^{(R,S)}(\phi)\right] = O_p\left(\frac{1}{\sqrt{N}}\right), \tag{23}$$

*where $\{\mathbf{X}_\mu^1, \mathbf{X}_\mu^2, \ldots, \mathbf{X}_\mu^N\}$ is defined in Definition 4.1.*

In Equation (23), the first term on the right-hand side represents the empirical risk of $\mathcal{L}_f^{(R,S)}(\phi)$ using validation data, while the second term denotes the minimum value of its true error. This equation illustrates that when $\mathcal{L}_f^{(R,S)}(\phi)$ is within the actual early stopping margin, specifically $O_p\left(\frac{1}{\sqrt{N}}\right)$, the function $\phi$ deviates from the optimal function of $\widetilde{\mathcal{L}}_f^{(N)}(\phi)$ by no more than $O_p\left(\frac{1}{\sqrt{N}}\right)$.

Based on Equation (23), we define the optimal function of $\mathcal{L}_f^{(R,S)}(\phi)$ for use with early stopping while monitoring validation losses as follows:

$$\phi_{\text{val}} \text{ is optimal in the optimization of } \mathcal{L}_f^{(R,S)}(\phi) \text{ using early stopping}$$

$$\triangleq \phi_* + O_p\left(\frac{1}{\sqrt{N}}\right), \quad \text{where} \quad \phi_* = \arg\min_{\phi:\Omega\to\mathbb{R}_{>0}} E_\mu\left[\widetilde{\mathcal{L}}_f^{(N)}(\phi)\right]. \tag{24}$$

The difference $O_p\left(\frac{1}{\sqrt{N}}\right)$, appearing in Equation (24), is negligible for DRE when $d \ge 3$. Indeed, using the triangle inequality in the $L_p$ norm for $\phi_* = \arg\min_{\phi:\Omega\to\mathbb{R}_{>0}} \widetilde{\mathcal{L}}_f^{(N)}(\phi)$ and Equation (20), we observe

$$\left\{E_P\left|\frac{dQ}{dP}(\mathbf{x}) - \phi_{\text{val}}(\mathbf{x})\right|^p\right\}^{1/p} \ge \underbrace{\left\{E_P\left|\frac{dQ}{dP}(\mathbf{x}) - \phi^*(\mathbf{x})\right|^p\right\}^{1/p}}_{=O\left(\frac{1}{N^{1/d}}\right)} - \underbrace{\left\{E_P\left|\phi_{\text{val}}(\mathbf{x}) - \phi^*(\mathbf{x})\right|^p\right\}^{1/p}}_{=O\left(\frac{1}{\sqrt{N}}\right)\ll\frac{1}{N^{1/d}}}. \tag{25}$$

Therefore, we finally obtain the following Theorem 4.8.

**Theorem 4.8.** *Assume the same assumptions and notations as in Theorem 4.5. Additionally, define*

$$\mathcal{F}_{K\text{-}Lip}^{(N)} = \left\{\phi \in \mathcal{F}_{K\text{-}Lip} \;\middle|\; \exists \phi_* \in \widetilde{\mathcal{F}}_{K\text{-}Lip}^{(N)} \text{ such that } \phi = \phi_* + O_p\left(\frac{1}{\sqrt{N}}\right)\right\}. \tag{26}$$

*That is, $\mathcal{F}_{K\text{-}Lip}^{(N)}$ denotes the set of all functions that differ by at most $O_p\left(\frac{1}{\sqrt{N}}\right)$ from some functions that minimize $\widetilde{\mathcal{L}}_f^{(N)}(\cdot)$. Therefore, the same results as in Theorem 4.5 hold for all $\phi \in \mathcal{F}_{K\text{-}Lip}^{(N)}$.*

## 5 CONCLUSIONS

We have established upper and lower bounds on the $L_p$ errors in DRE through the optimization of $f$-divergence loss functions. These bounds are applicable to any member of a group of Lipschitz continuous estimators, irrespective of the specific $f$-divergence loss function employed. These bounds provide new insights into how data dimensionality and the KL divergence between distributions

influence the accuracy of DRE. Furthermore, numerical experiments validate these theoretical findings, demonstrating that the relationship between $L_p$ errors, KL divergence, and data dimensionality aligns with the theoretical implications derived from the bounds. However, challenges remain, especially in high-dimensional settings, due to the curse of dimensionality and large sample requirements. Future studies could refine the theoretical framework to explore loss functions that enhance DRE performance in complex, high-dimensional tasks.

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

## A    ORGANIZATION OF THE SUPPLEMENTARY DOCUMENT

This supplementary document is organized as follows: Section B provides a list of notations used in this study. Section C presents the proofs referenced in Sections 3 and 4. Section D provides details of the experiments conducted. Section E explores further discussions related to this study.

Additionally, the code used for the numerical experiment is included as supplementary material.

## B    NOTATIONS

We outline all the notations used in the Appendix of this study in Table 1.

## C    PROOFS

In this section, we present the theorems and proofs referenced in this study. We begin by summarizing the definitions and assumptions introduced in previous sections, followed by the detailed theorems and proofs utilized throughout this study.

### C.1    DEFINITIONS AND ASSUMPTIONS IN SECTIONS 2, 3, AND 4

#### C.1.1    DEFINITIONS

**Definition C.1** ($f$-Divergence (Definition 2.1 restated))**.** The $f$-divergence $D_f$ between two probability measures $P$ and $Q$, induced by a convex function $f$ satisfying $f(1) = 0$, is defined as $D_f(Q\|P) = E_P[f(q(\mathbf{x})/p(\mathbf{x}))]$.

**Definition C.2** ($f$-Divergence Loss (Definition 2.2 restated))**.** Let $\hat{\mathbf{X}}_{P[R]} = \{\mathbf{X}_P^1, \mathbf{X}_P^2, \ldots, \mathbf{X}_P^R\}$, $\mathbf{X}_P^i \overset{\text{iid}}{\sim} P$ denote $R$ i.i.d. random variables from $P$, and let $\hat{\mathbf{X}}_{Q[S]} = \{\mathbf{X}_Q^1, \mathbf{X}_Q^2, \ldots, \mathbf{X}_Q^S\}$, $\mathbf{X}_Q^i \overset{\text{iid}}{\sim} Q$ denote $S$ i.i.d. random variables from $Q$. Then, for a twice differentiable convex function $f$, $f$-divergence loss $\mathcal{L}_f^{(R,S)}(\cdot)$ is defined as follows:

$$\mathcal{L}_f^{(R,S)}(\phi) = \frac{1}{S} \cdot \sum_{i=1}^{S} -f'\left(\phi(\mathbf{X}_Q^i)\right) + \frac{1}{R} \sum_{i=1}^{R} f^*\left(f'\left(\phi(\mathbf{X}_P^i)\right)\right), \tag{27}$$

where $\phi$ is a measurable function over $\Omega$ such that $\phi : \Omega \to \mathbb{R}_{>0}$.

**Definition C.3** ($\mu$-Representation $f$-Divergence Loss (Definition 4.1 restated))**.** Let $f$ be a twice differentiable convex function $f$. Then, $\mu$-representation function of $f$ for $u > 0$ at a point $\mathbf{x} \in \Omega$, which is written for $\widetilde{l}_f(u)$ in an abbreviated form, is defined as

$$\widetilde{l}_f(u; \mathbf{x}) = -f'(u) \cdot \frac{dQ}{d\mu}(\mathbf{x}) + f^*(f'(u)) \cdot \frac{dP}{d\mu}(\mathbf{x}), \tag{28}$$

where $f^*$ denotes the Legendre transform of $f$: $f^*(\psi) = \sup_{u \in \mathbb{R}} \{\psi \cdot u - f(u)\}$. Let $N = \min\{R, S\}$, and let $\hat{\mathbf{X}}_{\mu[N]} = \{\mathbf{X}_\mu^1, \ldots, \mathbf{X}_\mu^N\}$ denote $N$ i.i.d. random variables from $\mu$. Then, $\mu$-representation of the $f$-divergence loss $\mathcal{L}_f^{(R,S)}(\cdot)$ in Equation (27) at the points $\hat{\mathbf{X}}_{\mu[N]}$ is defined as

$$\widetilde{\mathcal{L}}_f^{(N)}(\phi) = \frac{1}{N} \cdot \sum_{i=1}^{N} \widetilde{l}_f(u; \mathbf{X}_\mu^{\mathbf{i}}) \tag{29}$$

where $\phi$ is a measurable function over $\Omega$ such that $\phi : \Omega \to \mathbb{R}_{>0}$.

#### C.1.2    ASSUMPTIONS

**Assumption C.4** (Assumption for the Upper Bound (Assumption 3.1 restated))**.** The following assumption is imposed on the probability distributions $P$ and $Q$.

    U1.  $T^*(\mathbf{x}) = -\log dQ/dP(\mathbf{x})$ is $L$-Lipschitz continuous with $L > 0$ on $\Omega$. i.e., $\exists L > 0$ s.t. $\left|T^*(\mathbf{y}) - T^*(\mathbf{x})\right| \leq L \cdot \left\|\mathbf{y} - \mathbf{x}\right\|_\infty$ for any $\mathbf{y}, \mathbf{x} \in \Omega$.

Table 1: Notations and definitions used in the proofs

| Notations | Definitions, Meanings |
|---|---|
| (Capital, small, and bold letters) | Random variables are denoted by capital letters, such as $A$. Corresponding values of the random variables are represented by small letters. Bold letters $\mathbf{A}$ and $\mathbf{a}$ denote sets of random variables and their corresponding values, respectively. |
| $\mathbb{R}, \mathbb{R}^d$ | The set of all real numbers and the $d$-dimensional vector space over the real numbers, respectively. |
| $\mathbb{R}_{>0}$ | The set of all positive real numbers: $\mathbb{R}_{>0} = \{x \in \mathbb{R} \mid x > 0\}$. |
| $\Omega$ | A subset of $\mathbb{R}^d$: $\Omega \subset \mathbb{R}^d$. |
| $f(x) = O(g(x))$, as $x \to a$ | Asymptotic boundedness with rate $g(x)$ as $x \to a$: $f(x) = O(g(x)) \Leftrightarrow \limsup_{x \to a} |f(x)/g(x)| \leq C$, where $C > 0$. |
| $f(x) = o(g(x))$, as $x \to a$ | Asymptotic domination with rate $g(x)$ as $x \to a$: $f(x) = o(g(x)) \Leftrightarrow \lim_{x \to a} f(x)/g(x) = 0$. |
| $\mathbf{X} = O_p(a_N)$, as $N \to \infty$ | Stochastic boundedness with rate $a_N$ in $\mu$: $\mathbf{X} = O_p(a_N) \Leftrightarrow$ for all $\varepsilon > 0$, there exist $\delta(\varepsilon) > 0$ and $N(\varepsilon) > 0$ such that $\mu\left(|\mathbf{X}|/a_N \geq \delta(\varepsilon)\right) < \varepsilon$ for all $N \geq N(\varepsilon)$. |
| $\mathbf{X} = o_p(a_N)$, as $N \to \infty$ | Convergence in probability with rate $a_N$ in $\mu$: $\mathbf{X} = o_p(a_N) \Leftrightarrow$ for all $\varepsilon > 0$, for all $\delta > 0$, there exists $N(\varepsilon, \delta) > 0$ such that $\mu(|\mathbf{X}|/a_N \geq \delta) < \varepsilon$ for all $N \geq N(\varepsilon, \delta)$. |
| $P \ll Q$ | $P$ is absolutely continuous with respect to $Q$. |
| $P, Q$ | A pair of probability measures with $P \ll Q$ and $Q \ll P$. |
| $\mu$ | A probability measure with $P \ll \mu$ and $Q \ll \mu$. |
| $\frac{dP}{dQ}$ | The Radon–Nikodým derivative of $P$ with respect to $Q$. |
| $\hat{\mathbf{X}}_{P[R]}$ | $R$ i.i.d. random variables from $P$: $\hat{\mathbf{X}}_{P[R]} = \{\mathbf{X}_P^1, \mathbf{X}_P^2, \ldots, \mathbf{X}_P^R\}$, where $\mathbf{X}_P^i \overset{\text{iid}}{\sim} P$. |
| $\hat{\mathbf{X}}_{Q[S]}$ | $S$ i.i.d. random variables from $Q$: $\hat{\mathbf{X}}_{Q[S]} = \{\mathbf{X}_Q^1, \mathbf{X}_Q^2, \ldots, \mathbf{X}_Q^S\}$, where $\mathbf{X}_Q^i \overset{\text{iid}}{\sim} Q$. |
| $N$ | $N = \min\{R, S\}$. |
| $\hat{\mathbf{X}}_{\mu[N]}$ | $N$ i.i.d. random variables from $\mu$: $\hat{\mathbf{X}}_{\mu[N]} = \{\mathbf{X}_\mu^1, \mathbf{X}_\mu^2, \ldots, \mathbf{X}_\mu^N\}$, where $\mathbf{X}_\mu^i \overset{\text{iid}}{\sim} \mu$. |
| $\mathbf{X}_{\mu[N]}^{(1)}(\mathbf{x})$ | The nearest neighbor variable of $\mathbf{x}$ in $\hat{\mathbf{X}}_{\mu[N]}$: $\mathbf{X}_{\mu[N]}^{(1)}(\mathbf{x})$ is the $\mathbf{X}_\mu^i$ such that $\|\mathbf{X}_\mu^i - \mathbf{x}\| < \|\mathbf{X}_\mu^j - \mathbf{x}\|$ for all $j \neq i$. |
| $D_f(Q\|P)$ | $f$-divergence: $D_f(Q\|P) = E_P[f(q(\mathbf{x})/p(\mathbf{x}))]$. See Definition C.1. |
| $\mathcal{L}_f^{(R,S)}(\cdot)$ | $f$-divergence loss function. See Definition C.2. |
| $\widetilde{l}_f(u; \mathbf{x})$ | $\mu$-representation of the $f$-divergence loss function at $\mathbf{x}$: $\widetilde{l}_f(u; \mathbf{x}) = -f'(u) \cdot \frac{dQ}{d\mu}(\mathbf{x}) + f^*(f'(u)) \cdot \frac{dP}{d\mu}(\mathbf{x})$. |
| $\widetilde{\mathcal{L}}_f^{(N)}(\cdot)$ | $\mu$-representation of the $f$-divergence loss function $\mathcal{L}_f^{(R,S)}(\cdot)$. See Definition 4.1. |
| $\bar{\mathcal{L}}_f(\phi)$ | The expectation of the $\mu$-representation of the $f$-divergence loss on $\mu$. See Lemma C.11. |
| $\|\cdot\|$ | The Euclidean norm. |
| $\|\cdot\|_\infty$ | The maximum norm in $\mathbb{R}^d$: $\|\mathbf{y} - \mathbf{x}\|_\infty = \max_{1 \leq i \leq d} |y_i - x_i|$. |
| $\Delta(\mathbf{a}, r)$ | The $d$-dimensional interval centered at $\mathbf{a}$ with each side of length $r$: $\Delta(\mathbf{a}, r) = \{\mathbf{x} \in \mathbb{R}^d \mid \|\mathbf{x} - \mathbf{a}\|_\infty < r/2\}$. |
| $\text{diam}(\mathcal{B})$ | The diameter of $\Omega$: $\text{diam}(\Omega) = \inf\{r > 0 \mid \exists \mathbf{a} \in \Omega \text{ s.t. } \Omega \subseteq \Delta(\mathbf{a}, r)\}$. |

**Assumption C.5** (Assumptions for the Lower Bound (Assumption 3.2 restated))**.** The following assumptions are imposed on the probability distributions $P$ and $Q$.

L1. $T^*(\mathbf{x}) = -\log dQ/dP(\mathbf{x})$ is $L$-bi-Lipschitz continuous on $\Omega$. i.e., $\exists L > 1$ s.t. $(1/L) \leq \left| T^*(\mathbf{y}) - T^*(\mathbf{x}) \right| \leq L \cdot \left\| \mathbf{y} - \mathbf{x} \right\|_\infty$ for any $\mathbf{y}, \mathbf{x} \in \Omega$.

L2. $E_P\left[ (dQ/dP)^p \right] < \infty$ where $p \leq d$.

**Assumption C.6** (Assumptions for the Convex Function $f$ (Assmption 3.3 restated))**.** The following assumptions are assumed for the convex function $f$.

F1. $f$ is three-time differentiable.

F2. $f''(u) > 0$ for all $u > 0$.

F3. $E_P\left[ f''(dQ/dP) \right] < \infty$.

**Assumption C.7** (Assumption for the Support (Assmption 3.4 restated))**.** The following assumption is assumed for $\Omega$.

O1. $\mathrm{diam}(\Omega) < \infty$.

## C.2 THEOREMS AND PROOFS IN SECTIONS 2, 3, AND 4

**Lemma C.8.** *Let $f$ be a twice differentiable function. Consider $\widetilde{l}_f(u; \mathbf{x})$ defined as in Equation (28). Then, the first derivative of $\widetilde{l}_f(u; \mathbf{x})$ with respect to $u$ is given by:*

$$\frac{d}{du}\widetilde{l}_f(u; \mathbf{x}) = \left\{ u - \frac{dQ}{dP}(\mathbf{x}) \right\} \cdot f''(u) \cdot \frac{dP}{d\mu}(\mathbf{x}). \tag{30}$$

*Additionally, if $\widetilde{l}_f(u; \mathbf{x})$ is thrice differentiable, its second derivative with respect to $u$ is given by:*

$$\frac{d^2}{du^2}\widetilde{l}_f(u; \mathbf{x}) = \left\{ \left( u - \frac{dQ}{dP}(\mathbf{x}) \right) \cdot f'''(u) + f''(u) \right\} \cdot \frac{dP}{d\mu}(\mathbf{x}). \tag{31}$$

*Proof of Lemma C.8.* First, note that

$$\widetilde{l}_f(u; \mathbf{x}) = -f'(u) \cdot \frac{dQ}{d\mu}(\mathbf{x}) + f^*(f'(u)) \cdot \frac{dP}{d\mu}(\mathbf{x})$$
$$= -f'(u) \cdot \frac{dQ}{d\mu}(\mathbf{x}) + \left\{ f'(u) \cdot u - f(u) \right\} \cdot \frac{dP}{d\mu}(\mathbf{x}). \tag{32}$$

Differentiating Equation (32) with respect to $u$, we obtain the first and second derivatives of $\widetilde{l}_f(u; \mathbf{x})$ as follows:

$$\frac{d}{du}\widetilde{l}_f(u; \mathbf{x}) = -f''(u) \cdot \frac{dQ}{d\mu}(\mathbf{x}) + u \cdot f''(u) \cdot \frac{dP}{d\mu}(\mathbf{x})$$
$$= \left\{ u - \frac{dQ}{dP}(\mathbf{x}) \right\} \cdot f''(u) \cdot \frac{dP}{d\mu}(\mathbf{x}), \tag{33}$$

and

$$\frac{d^2}{du^2}\widetilde{l}_f(u; \mathbf{x}) = -f'''(u) \cdot \frac{dQ}{d\mu}(\mathbf{x}) + f''(u) \cdot \frac{dP}{d\mu}(\mathbf{x}) + u \cdot f'''(u) \cdot \frac{dP}{d\mu}(\mathbf{x})$$
$$= \left\{ \left( u - \frac{dQ}{dP}(\mathbf{x}) \right) \cdot f'''(u) + f''(u) \right\} \cdot \frac{dP}{d\mu}(\mathbf{x}). \tag{34}$$

This completes the proof. $\square$

**Theorem C.9.** *Assume that $f$ satisfies Assumption C.6. Then, $\widetilde{l}_f(u; \mathbf{x})$, as defined in Equation (28), is minimized if and only if $u^*(\mathbf{x}) = \frac{dQ}{dP}(\mathbf{x})$. Additionally, for $u > 0$, the following holds:*

$$
\widetilde{l}_f(u; \mathbf{x}) - \widetilde{l}_f\left(\frac{dQ}{dP}(\mathbf{x}); \mathbf{x}\right)
$$

$$
= \frac{1}{2} \cdot f''\left(\frac{dQ}{dP}(\mathbf{x})\right) \cdot \frac{dP}{d\mu}(\mathbf{x}) \cdot \left|u - \frac{dQ}{dP}(\mathbf{x})\right|^2 + o\left(\left|u - \frac{dQ}{dP}(\mathbf{x})\right|^2\right), \tag{35}
$$

*where $f(a) = o(a)$ (as $a \to 0$) denotes asymptotic domination such that $\lim_{a \to 0} \frac{f(a)}{a} \to 0$.*

*Proof of Theorem C.9.* Let $\mathrm{sign}(x)$ denote the sign of the value $x$: specifically, $\mathrm{sign}(x) = 1$ if $x > 0$, $\mathrm{sign}(x) = -1$ if $x < 0$, and $\mathrm{sign}(x) = 0$ if $x = 0$.

From Equation (30) in Lemma C.8, we have

$$
\mathrm{sign}\left(\frac{d}{du}\widetilde{l}_f(u; \mathbf{x})\right) = \mathrm{sign}\left(\left\{u - \frac{dQ}{dP}(\mathbf{x})\right\} \cdot f''(u) \cdot \frac{dP}{d\mu}(\mathbf{x})\right)
$$

$$
= \mathrm{sign}\left(\left\{u - \frac{dQ}{dP}(\mathbf{x})\right\}\right) \cdot \mathrm{sign}\left(f''(u)\right) \cdot \mathrm{sign}\left(\frac{dP}{d\mu}(\mathbf{x})\right)
$$

$$
= \mathrm{sign}\left(u - \frac{dQ}{dP}(\mathbf{x})\right). \tag{36}
$$

Thus, $\widetilde{l}_f(u; \mathbf{x})$ is minimized only when $u^* = \frac{dQ}{dP}(\mathbf{x})$.

Next, from Equation (30),

$$
\frac{d}{du}\widetilde{l}_f\left(\frac{dQ}{dP}(\mathbf{x}); \mathbf{x}\right) = 0, \tag{37}
$$

and from Equation (31),

$$
\frac{d^2}{du^2}\widetilde{l}_f\left(\frac{dQ}{dP}(\mathbf{x}); \mathbf{x}\right) = f''\left(\frac{dQ}{dP}(\mathbf{x})\right) \cdot \frac{dP}{d\mu}(\mathbf{x}). \tag{38}
$$

Thus, using the second-order Taylor expansion of $\widetilde{l}_f(u; \mathbf{x})$ around $u = \frac{dQ}{dP}(\mathbf{x})$, we have

$$
\widetilde{l}_f(u; \mathbf{x}) - \widetilde{l}_f\left(\frac{dQ}{dP}(\mathbf{x}); \mathbf{x}\right)
$$

$$
= \frac{1}{2} \cdot f''\left(\frac{dQ}{dP}(\mathbf{x})\right) \cdot \frac{dP}{d\mu}(\mathbf{x}) \cdot \left|u - \frac{dQ}{dP}(\mathbf{x})\right|^2 + o\left(\left|u - \frac{dQ}{dP}(\mathbf{x})\right|^2\right). \tag{39}
$$

This completes the proof. $\qquad \square$

**Proposition C.10** (Proposition 4.2 restated). *Assume that $f$ satisfies Assumption C.6. Let $\widetilde{\mathcal{L}}_f^{(N)}(\phi)$ denote the $\mu$-representation $f$-divergence loss as defined in Definition C.3. Then, the minimum value of $\widetilde{\mathcal{L}}_f^{(N)}(\phi)$ over all measurable functions $\phi : \Omega \to \mathbb{R}_{>0}$ is achieved if and only if $\phi$ satisfies*

$$
\phi(\mathbf{X}_\mu^i) = \frac{dQ}{dP}(\mathbf{X}_\mu^i), \quad \text{for } i = 1, 2, \dots, N. \tag{40}
$$

*proof of Proposition C.10.* From Theorem C.9, we observe that, for $i = 1, 2, \dots, N$,

$$
\min_{u > 0} \widetilde{l}_f(u; \mathbf{X}_\mu^i) = \widetilde{l}_f\left(\frac{dQ}{dP}(\mathbf{X}_\mu^i); \mathbf{X}_\mu^i\right), \tag{41}
$$

where the minimum value is achieved only at $u = \frac{dQ}{dP}(\mathbf{X}_\mu^i)$.

Thus,

$$
\begin{aligned}
\min_{\phi:\Omega\to\mathbb{R}_{>0}} \widetilde{\mathcal{L}}_f^{(N)}(\phi) &= \min_{\phi:\Omega\to\mathbb{R}_{>0}} \frac{1}{N}\cdot\sum_{i=1}^{N}\widetilde{l}_f(\phi(\mathbf{X}_\mu^i);\mathbf{X}_\mu^i) \\
&= \min_{\substack{\phi(\mathbf{X}_\mu^i)>0,\\ i=1,2,\ldots,N}} \frac{1}{N}\cdot\sum_{i=1}^{N}\widetilde{l}_f(\phi(\mathbf{X}_\mu^i);\mathbf{X}_\mu^i) \\
&= \min_{\substack{u_i>0,\\ i=1,2,\ldots,N}} \frac{1}{N}\cdot\sum_{i=1}^{N}\widetilde{l}_f(u_i;\mathbf{X}_\mu^i) \\
&= \frac{1}{N}\cdot\sum_{i=1}^{N}\widetilde{l}_f\left(\frac{dQ}{dP}(\mathbf{X}_\mu^i);\mathbf{X}_\mu^i\right).
\end{aligned}
\tag{42}
$$

Suppose that $\widetilde{\phi}(\mathbf{x})$ is a function on $\Omega$ that satisfies Equation (40). From Equation (42), we have

$$
\begin{aligned}
&\widetilde{\mathcal{L}}_f^{(N)}\left(\widetilde{\phi}\right) - \min_{\phi:\Omega\to\mathbb{R}_{>0}} \widetilde{\mathcal{L}}_f^{(N)}(\phi) \\
&= \frac{1}{N}\cdot\sum_{i=1}^{N}\widetilde{l}_f\left(\widetilde{\phi}(\mathbf{X}_\mu^i);\mathbf{X}_\mu^i\right) - \frac{1}{N}\cdot\sum_{i=1}^{N}\widetilde{l}_f\left(\frac{dQ}{dP}(\mathbf{X}_\mu^i);\mathbf{X}_\mu^i\right) \\
&= \frac{1}{N}\cdot\sum_{i=1}^{N}\widetilde{l}_f\left(\frac{dQ}{dP}(\mathbf{X}_\mu^i);\mathbf{X}_\mu^i\right) - \frac{1}{N}\cdot\sum_{i=1}^{N}\widetilde{l}_f\left(\frac{dQ}{dP}(\mathbf{X}_\mu^i);\mathbf{X}_\mu^i\right) \\
&= 0.
\end{aligned}
\tag{43}
$$

Here, we show that the minimum value of $\widetilde{\mathcal{L}}_f^{(N)}(\phi)$ over all measurable functions $\phi:\Omega\to\mathbb{R}_{>0}$ is achieved if $\phi:\Omega\to\mathbb{R}_{>0}$ satisfies Equation (40).

Next, we show that the minimum value of $\widetilde{\mathcal{L}}_f^{(N)}(\phi)$ over all measurable functions $\phi:\Omega\to\mathbb{R}_{>0}$ is achieved only if $\phi:\Omega\to\mathbb{R}_{>0}$ satisfies Equation (40).

We have, for any function $\phi:\Omega\to(0,\infty)$,

$$
\begin{aligned}
&\widetilde{\mathcal{L}}_f^{(N)}(\phi) - \min_{\phi:\Omega\to\mathbb{R}_{>0}} \widetilde{\mathcal{L}}_f^{(N)}(\phi) \\
&= \frac{1}{N}\cdot\sum_{i=1}^{N}\widetilde{l}_f\left(\phi(\mathbf{X}_\mu^i);\mathbf{X}_\mu^i\right) - \frac{1}{N}\cdot\sum_{i=1}^{N}\min_{\substack{u_i>0,\\ i=1,2,\ldots,N}}\widetilde{l}_f(u_i;\mathbf{X}_\mu^i) \\
&= \frac{1}{N}\cdot\sum_{i=1}^{N}\left\{\widetilde{l}_f\left(\phi(\mathbf{X}_\mu^i);\mathbf{X}_\mu^i\right) - \min_{u>0}\widetilde{l}_f(u;\mathbf{X}_\mu^i)\right\}.
\end{aligned}
\tag{44}
$$

Suppose that $\phi(\mathbf{X}_\mu^i)\neq\frac{dQ}{dP}(\mathbf{X}_\mu^i)$. Then, from Equation (41), we have

$$
\widetilde{l}_f\left(\phi(\mathbf{X}_\mu^i);\mathbf{X}_\mu^i\right) > \min_{u>0}\widetilde{l}_f(u;\mathbf{X}_\mu^i).
\tag{45}
$$

From Equations (44) and (45), we observe that

$$
\begin{aligned}
&\widetilde{\mathcal{L}}_f^{(N)}(\phi) - \min_{\phi:\Omega\to\mathbb{R}_{>0}} \widetilde{\mathcal{L}}_f^{(N)}(\phi) \\
&= \frac{1}{N}\cdot\sum_{i=1}^{N}\left\{\widetilde{l}_f\left(\phi(\mathbf{X}_\mu^i);\mathbf{X}_\mu^i\right) - \min_{u>0}\widetilde{l}_f(u;\mathbf{X}_\mu^i)\right\} \\
&\geq \frac{1}{N}\cdot\left\{\widetilde{l}_f\left(\phi(\mathbf{X}_\mu^i);\mathbf{X}_\mu^i\right) - \min_{u>0}\widetilde{l}_f(u;\mathbf{X}_\mu^i)\right\} \\
&> 0
\end{aligned}
\tag{46}
$$

Thus, we see that the minimum value of $\widetilde{\mathcal{L}}_f^{(N)}(\phi)$ over all measurable functions $\phi : \Omega \to \mathbb{R}_{>0}$ is achieved only if $\phi : \Omega \to \mathbb{R}_{>0}$ satisfies Equation (40).

This completes the proof. $\qquad\square$

**Lemma C.11.** *Assume that $f$ satisfies Assumption C.6. Let $\widetilde{\mathcal{L}}_f^{(N)}(\phi)$ denote the $\mu$-representation $f$-divergence loss as defined in Definition C.3. Define*

$$
\begin{aligned}
\bar{\mathcal{L}}_f(\phi) &= E_\mu \left[ \widetilde{\mathcal{L}}_f^{(N)}(\phi) \right] \\
&= \frac{1}{N} \cdot \sum_{i=1}^N E_\mu \left[ -f'(\phi(\mathbf{x}_i)) \cdot \frac{dQ}{d\mu}(\mathbf{x}_i) \right] \\
&\quad + \frac{1}{N} \cdot \sum_{i=1}^N E_\mu \left[ f^*(f'(\phi(\mathbf{x}_i))) \cdot \frac{dP}{d\mu}(\mathbf{x}_i) \right].
\end{aligned}
\tag{47}
$$

*Then,*

$$
E_\mu \left[ \min_{\phi:\Omega\to\mathbb{R}_{>0}} \widetilde{\mathcal{L}}_f^{(N)}(\phi) \right] = \min_{\phi:\Omega\to\mathbb{R}_{>0}} \bar{\mathcal{L}}_f(\phi) = \min_{\phi:\Omega\to\mathbb{R}_{>0}} E_\mu \left[ \mathcal{L}_f^{(R,S)}(\phi) \right],
\tag{48}
$$

*where the infimum is taken over all measurable functions $\phi : \Omega \to \mathbb{R}_{>0}$ such that $E_P[f(\phi(\mathbf{X}))] < \infty$. Additionally, the equality in Equation (48) hold when $\phi(\mathbf{x}) = \frac{dQ}{dP}(\mathbf{x})$.*

*proof of Lemma C.11.* Let, $\widetilde{l}_f^*(\mathbf{x}) = \min_{u\in\mathbb{R}_{>0}} \widetilde{l}_f(u; \mathbf{x})$. From Theorem C.9, we see $\widetilde{l}_f^*(\mathbf{x}) = \widetilde{l}_f(dQ/dP(\mathbf{x}); \mathbf{x})$. Then, we have

$$
\begin{aligned}
\widetilde{l}_f^*(\mathbf{x}) &= \widetilde{l}_f \left( \frac{dQ}{dP}(\mathbf{x}); \mathbf{x} \right) \\
&= -f' \left( \frac{dQ}{dP}(\mathbf{x}) \right) \cdot \frac{dQ}{d\mu}(\mathbf{x}) + \left\{ f' \left( \frac{dQ}{dP}(\mathbf{x}) \right) \cdot \frac{dQ}{dP}(\mathbf{x}) - f \left( \frac{dQ}{dP}(\mathbf{x}) \right) \right\} \cdot \frac{dP}{d\mu}(\mathbf{x}) \\
&= -f \left( \frac{dQ}{dP}(\mathbf{x}) \right) \cdot \frac{dP}{d\mu}(\mathbf{x}).
\end{aligned}
\tag{49}
$$

Now, we have

$$
\begin{aligned}
\min_{\phi:\Omega\to\mathbb{R}_{>0}} \widetilde{\mathcal{L}}_f^{(N)}(\phi) &= \min_{\phi:\Omega\to\mathbb{R}_{>0}} \frac{1}{N} \cdot \sum_{i=1}^N \widetilde{l}_f(\phi(\mathbf{X}_\mu^i); \mathbf{X}_\mu^i) \\
&= \min_{\substack{\phi(\mathbf{X}_\mu^i)>0, \\ i=1,2,\dots,N}} \frac{1}{N} \cdot \sum_{i=1}^N \widetilde{l}_f(\phi(\mathbf{X}_\mu^i); \mathbf{X}_\mu^i) \\
&= \min_{\substack{u_i>0, \\ i=1,2,\dots,N}} \frac{1}{N} \cdot \sum_{i=1}^N \widetilde{l}_f(u_i; \mathbf{X}_\mu^i) \\
&= \frac{1}{N} \cdot \sum_{i=1}^N \widetilde{l}_f^*(\mathbf{X}_\mu^i).
\end{aligned}
\tag{50}
$$

Additionally, we have

$$
\begin{aligned}
E_\mu \left[ \widetilde{\mathcal{L}}_f^{(N)}(\phi) \right] = E_\mu \Bigg[ &\frac{1}{N} \cdot \sum_{i=1}^N -f'(\phi(\mathbf{x}_i)) \cdot \frac{dQ}{d\mu}(\mathbf{x}_i) \\
&+ \frac{1}{N} \cdot \sum_{i=1}^N f^*(f'(\phi(\mathbf{x}_i))) \cdot \frac{dP}{d\mu}(\mathbf{x}_i) \Bigg]
\end{aligned}
$$

$$
= -\frac{1}{N} \cdot \sum_{i=1}^{N} E_\mu \left[ f'\left(\phi(\mathbf{x}_i)\right) \cdot \frac{dQ}{d\mu}(\mathbf{x}_i) \right]
$$

$$
+ \frac{1}{N} \cdot \sum_{i=1}^{N} E_\mu \left[ f^*\left(f'\left(\phi(\mathbf{x}_i)\right)\right) \cdot \frac{dP}{d\mu}(\mathbf{x}_i) \right]
$$

$$
= -\frac{1}{N} \cdot \sum_{i=1}^{N} E_Q\left[f'\left(\phi\right)\right] + \frac{1}{N} \cdot \sum_{i=1}^{N} E_P\left[f^*\left(f'\left(\phi\right)\right)\right]
$$

$$
= -E_Q\left[f'\left(\phi\right)\right] + E_P\left[f^*\left(f'\left(\phi\right)\right)\right], \tag{51}
$$

and

$$
E\left[\mathcal{L}_f^{(R,S)}(\phi)\right] = E\left[\frac{1}{R} \cdot \sum_{i=1}^{S} -f'\left(\phi(\mathbf{x}_i^q)\right)\right.
$$

$$
\left. + \frac{1}{S} \cdot \sum_{i=1}^{R} f^*\left(f'\left(\phi(\mathbf{x}_i^p)\right)\right)\right]
$$

$$
= -\frac{1}{S} \cdot \sum_{i=1}^{S} E_Q\left[f'\left(\phi(\mathbf{x}_i)\right)\right]
$$

$$
+ \frac{1}{R} \cdot \sum_{i=1}^{R} E_P\left[f^*\left(f'\left(\phi(\mathbf{x}_i)\right)\right)\right]
$$

$$
= -\frac{1}{S} \cdot \sum_{i=1}^{S} E_Q\left[f'\left(\phi\right)\right] + \frac{1}{R} \cdot \sum_{i=1}^{R} E_P\left[f^*\left(f'\left(\phi\right)\right)\right]
$$

$$
= -E_Q\left[f'\left(\phi\right)\right] + E_P\left[f^*\left(f'\left(\phi\right)\right)\right]. \tag{52}
$$

Now, note that, from Equation (1) (Nguyen et al. (2007)), we see

$$
\min_{\phi:\Omega\to\mathbb{R}_{>0}} -E_Q\left[f'\left(\phi\right)\right] + E_P\left[f^*\left(f'\left(\phi\right)\right)\right] = -D_f(Q\|P), \tag{53}
$$

where $D_f(Q\|P)$ denotes $f$-divergence defined in Definition C.1 and the equality in Equation (53) holds for $\phi(\mathbf{x}) = dQ/dP(\mathbf{x})$.

From Equations (51), (52) and (53), we have

$$
\min_{\phi:\Omega\to\mathbb{R}_{>0}} E_\mu\left[\widetilde{\mathcal{L}}_f^{(N)}(\phi)\right] = \min_{\phi:\Omega\to\mathbb{R}_{>0}} E\left[\mathcal{L}_f^{(R,S)}(\phi)\right] = -D_f(Q\|P), \tag{54}
$$

and the equality in Equation (54) holds for $\phi(\mathbf{x}) = dQ/dP(\mathbf{x})$.

Substituting Equation (49) into Equation (50), we have

$$
\min_{\phi:\Omega\to\mathbb{R}_{>0}} \widetilde{\mathcal{L}}_f^{(N)}(\phi) = \frac{1}{N} \cdot \sum_{i=1}^{N} \widetilde{l}_f^*(\mathbf{X}_\mu^i)
$$

$$
= \frac{1}{N} \cdot \sum_{i=1}^{N} -f\left(\frac{dQ}{dP}(\mathbf{X}_\mu^i)\right) \cdot \frac{dP}{d\mu}(\mathbf{X}_\mu^i). \tag{55}
$$

Thus,

$$
E_\mu\left[\min_{\phi:\Omega\to\mathbb{R}_{>0}} \widetilde{\mathcal{L}}_f^{(N)}(\phi)\right] = E_\mu\left[\frac{1}{N} \cdot \sum_{i=1}^{N} -f\left(\frac{dQ}{dP}(\mathbf{x}_i)\right) \cdot \frac{dP}{d\mu}(\mathbf{x}_i)\right]
$$

$$
= -\frac{1}{N} \cdot \sum_{i=1}^{N} E_\mu\left[f\left(\frac{dQ}{dP}(\mathbf{x}_i)\right) \cdot \frac{dP}{d\mu}(\mathbf{x}_i)\right]
$$

$$= -\frac{1}{N} \cdot \sum_{i=1}^{N} D_f(Q\|P)$$

$$= -D_f(Q\|P), \tag{56}$$

From Equations (54) and (56), we have

$$E_\mu \left[ \min_{\phi:\Omega\to\mathbb{R}_{>0}} \widetilde{\mathcal{L}}_f^{(N)}(\phi) \right] = \min_{\phi:\Omega\to\mathbb{R}_{>0}} \overline{\mathcal{L}}_f(\phi) = \min_{\phi:\Omega\to\mathbb{R}_{>0}} E_\mu \left[ \mathcal{L}_f^{(R,S)}(\phi) \right], \tag{57}$$

and the equality in each Equation (57) holds for $\phi(\mathbf{x}) = dQ/dP(\mathbf{x})$.

This completes the proof. $\qquad\square$

The following theorem presents the convergence rate of the expected value of the distance between two neighboring samples. Similar theorems have been discussed in studies on the order statistics of multidimensional continuous random variables (e.g., Biau & Devroye (2015), p. 17, Theorem 2.1).

**Theorem C.12** (Theorem 4.3 restated). *Assume that $\Omega$ is a compact set, as stated in Assumption C.7. Let $\mathbf{X}_{\mu[N]}^{(1)}(\mathbf{x})$ denote the nearest neighbor of $\mathbf{x}$ in $\hat{\mathbf{X}}_{\mu[N]}$. Specifically, let $\mathbf{X}_{\mu[N]}^{(1)}(\mathbf{x})$ be $\mathbf{X}_\mu^i$ in $\hat{\mathbf{X}}_{\mu[N]}$ such that*

$$\|\mathbf{X}_\mu^i - \mathbf{x}\|_\infty < \|\mathbf{X}_\mu^j - \mathbf{x}\|_\infty \;\; (\forall\, j < i), \quad \textit{and} \quad \|\mathbf{X}_\mu^i - \mathbf{x}\|_\infty \leq \|\mathbf{X}_\mu^j - \mathbf{x}\|_\infty \;\; (\forall\, j > i). \tag{58}$$

*Additionally, let $\mathrm{diam}(\Omega)$ denote the diameter of $\Omega$. i.e., $\mathrm{diam}(\mathcal{B}) = \inf_{r\in\mathbb{R}}\{\mathcal{B} \subseteq \Delta(\mathbf{a}, r) \mid \exists\mathbf{a} \in \mathcal{B}\}$, where $\Delta(\mathbf{a}, r)$ denotes the $d$-dimensional interval centered at $\mathbf{a}$ with each side of length $r$: $\Delta(\mathbf{a}, r) = \{\mathbf{x} \in \mathbb{R}^d \mid \|\mathbf{x} - \mathbf{a}\|_\infty < r/2\}$.*

*Then, for $1 \leq \kappa \leq d$,*

$$E_\mu \left\| \mathbf{X}_{\mu[N]}^{(1)}(\mathbf{x}) - \mathbf{x} \right\|_\infty^\kappa \leq \mathrm{diam}(\Omega)^\kappa \cdot \left( \frac{1}{N+1} \right)^{\kappa/d}, \quad \textit{for all } N \geq 1. \tag{59}$$

*proof of Theorem C.12.* Let we rewrite $\mathbf{x}$ in Equation (59) as $\mathbf{X}_\mu^{N+1}$. Subsequently, let $\hat{\mathbf{X}}_{\mu[N+1]} = \hat{\mathbf{X}}_{\mu[N]} \cup \{\mathbf{X}_\mu^{N+1}\}$. Let $\Delta_i = \Omega \cap \Delta(\mathbf{X}_\mu^i, \|\mathbf{X}_{\mu[N]}^{(1)}(\mathbf{X}_\mu^i) - \mathbf{X}_\mu^i\|_\infty)$, where $\Delta(\mathbf{a}, r) = \{\mathbf{x} \in \mathbb{R}^d \mid \|\mathbf{x} - \mathbf{a}\|_\infty < r/2\}$. Note that, $\Delta_i \cap \Delta_j = \phi$ if $i \neq j$. Thus, $\sqcup_{i=1}^{N+1}\Delta_i \subseteq \Omega$.

Now, let $\lambda$ denote the Lebesgue measure on $\mathbb{R}^d$. Then, we have

$$\sum_{i=1}^{N+1} \lambda\left(\Delta_i\right) = \lambda\left(\sqcup_{i=1}^{N+1}\Delta_i\right) \leq \lambda\left(\Omega\right) \leq \mathrm{diam}(\Omega)^d, \tag{60}$$

Subsequently, since $\lambda\left(\Delta_i\right) = \left\| \mathbf{X}_{\mu[N]}^{(1)}(\mathbf{X}_\mu^i) - \mathbf{X}_\mu^i \right\|_\infty^d$, we have

$$\sum_{i=1}^{N+1} \lambda\left(\Delta_i\right) = \sum_{i=1}^{N+1} \left\| \mathbf{X}_{\mu[N]}^{(1)}(\mathbf{X}_\mu^i) - \mathbf{X}_\mu^i \right\|_\infty^d. \tag{61}$$

Thus, from Equations (60) and (61), we have

$$\sum_{i=1}^{N+1} \left\| \mathbf{X}_{\mu[N]}^{(1)}(\mathbf{X}_\mu^i) - \mathbf{X}_\mu^i \right\|_\infty^d \leq \mathrm{diam}(\Omega)^d. \tag{62}$$

Note that it follows from Jensen's inequality that

$$\frac{1}{N+1}\sum_{i=1}^{N+1} \left\| \mathbf{X}_{\mu[N]}^{(1)}(\mathbf{X}_\mu^i) - \mathbf{X}_\mu^i \right\|_\infty^\kappa \leq \left\{ \frac{1}{N+1}\sum_{i=1}^{N+1} \left\| \mathbf{X}_{\mu[N]}^{(1)}(\mathbf{X}_\mu^i) - \mathbf{X}_\mu^i \right\|_\infty^d \right\}^{\kappa/d}. \tag{63}$$

From Equations (62) and (63), we have

$$\frac{1}{N+1}\sum_{i=1}^{N+1}\left\|\mathbf{X}_{\mu[N]}^{(1)}(\mathbf{X}_{\mu}^i)-\mathbf{X}_{\mu}^i\right\|_{\infty}^{\kappa} \leq \left\{\frac{1}{N+1}\sum_{i=1}^{N+1}\left\|\mathbf{X}_{\mu[N]}^{(1)}(\mathbf{X}_{\mu}^i)-\mathbf{X}_{\mu}^i\right\|_{\infty}^{d}\right\}^{\kappa/d}$$

$$\leq \left\{\frac{1}{N+1}\cdot\mathrm{diam}(\Omega)^d\right\}^{\kappa/d}$$

$$= \mathrm{diam}(\Omega)^{\kappa}\cdot\left(\frac{1}{N+1}\right)^{\kappa/d}.$$

$$(64)$$

Thus,

$$\frac{1}{N+1}\sum_{i=1}^{N+1}E_{\mathbf{X}_{\mu}^i}\left\|\mathbf{X}_{\mu[N]}^{(1)}(\mathbf{x})-\mathbf{x}\right\|_{\infty}^{\kappa} \leq \mathrm{diam}(\Omega)^{\kappa}\cdot\left(\frac{1}{N+1}\right)^{\kappa/d}, \tag{65}$$

where $E_{\mathbf{X}_{\mu}^i}\left\|\mathbf{X}_{\mu[N]}^{(1)}(\mathbf{x})-\mathbf{x}\right\|_{\infty}^{\kappa}$ denotes the expectation of $\left\|\mathbf{X}_{\mu[N]}^{(1)}(\mathbf{X}_{\mu}^i)-\mathbf{X}_{\mu}^i\right\|_{\infty}^{\kappa}$ with respect to $\mathbf{X}_{\mu}^i$.

Note that,

$$E_{\mu}\left\|\mathbf{X}_{\mu[N]}^{(1)}(\mathbf{x})-\mathbf{x}\right\|_{\infty}^{\kappa} = E_{\mathbf{X}_{\mu}^i}\left\|\mathbf{X}_{\mu[N]}^{(1)}(\mathbf{x})-\mathbf{x}\right\|_{\infty}^{\kappa}. \tag{66}$$

Therefore,

$$E_{\mu}\left\|\mathbf{X}_{\mu[N]}^{(1)}(\mathbf{x})-\mathbf{x}\right\|_{\infty}^{\kappa} = \frac{1}{N+1}\sum_{i=1}^{N+1}E_{\mathbf{X}_{\mu}^i}\left\|\mathbf{X}_{\mu[N]}^{(1)}(\mathbf{x})-\mathbf{x}\right\|_{\infty}^{\kappa}. \tag{67}$$

Finally, from Equations (65) and (67), we have

$$E_{\mu}\left\|\mathbf{X}_{\mu[N]}^{(1)}(\mathbf{x})-\mathbf{x}\right\|_{\infty}^{\kappa} = \frac{1}{N+1}\sum_{i=1}^{N+1}E_{\mathbf{X}_{\mu}^i}\left\|\mathbf{X}_{\mu[N]}^{(1)}(\mathbf{x})-\mathbf{x}\right\|_{\infty}^{\kappa} \leq \mathrm{diam}(\Omega)^{\kappa}\cdot\left(\frac{1}{N+1}\right)^{\kappa/d}. \tag{68}$$

This completes the proof. □

**Corollary C.13.** *Assume the same assumption as in Theorem C.12. Then, for $1 \leq p \leq d$,*

$$\varlimsup_{N\to\infty} N^{1/d}\cdot\left\{E_{\mu}\left[\left\|\mathbf{X}_{\mu[N]}^{(1)}(\mathbf{x})-\mathbf{x}\right\|_{\infty}^{p}\right]\right\}^{1/p} \leq \mathrm{diam}(\Omega). \tag{69}$$

*proof of Corollary C.13.* First, from Theorem C.12 when $\kappa = p$,

$$E_{\mu}\left\|\mathbf{X}_{\mu[N]}^{(1)}(\mathbf{x})-\mathbf{x}\right\|_{\infty}^{p} \leq \mathrm{diam}(\Omega)^p\cdot\left(\frac{1}{N+1}\right)^{p/d}, \quad \text{for all } N \geq 1. \tag{70}$$

Thus, for all $N \geq 1$,

$$\left\{E_{\mu}\left\|\mathbf{X}_{\mu[N]}^{(1)}(\mathbf{x})-\mathbf{x}\right\|_{\infty}^{p}\right\}^{1/p} \leq \left\{\mathrm{diam}(\Omega)^p\cdot\left(\frac{1}{N+1}\right)^{p/d}\right\}^{1/p}$$

$$= \mathrm{diam}(\Omega)\cdot\left(\frac{1}{N+1}\right)^{1/d} \tag{71}$$

Taking $\varlimsup_{N\to\infty}$ on both sides of the above inequality, we have

$$\varlimsup_{N\to\infty} N^{1/d}\cdot\left\{E_{\mu}\left[\left\|\mathbf{X}_{\mu[N]}^{(1)}(\mathbf{x})-\mathbf{x}\right\|_{\infty}^{p}\right]\right\}^{1/p}$$

$$\leq \varlimsup_{N\to\infty}\left\{N^{1/d}\cdot\mathrm{diam}(\Omega)\cdot\left(\frac{1}{N+1}\right)^{1/d}\right\}$$

$$= \mathrm{diam}(\Omega). \tag{72}$$

This completes the proof. □

**Corollary C.14.** *Assume the same assumption as in Theorem C.12. Then, for $1 \le p \le d/2$,*

$$\overline{\lim_{N \to \infty}} N^{1/d} \cdot \left\{ E_P \left[ \left\{ \frac{dQ}{dP}(\mathbf{x}) \right\}^p \cdot \left\| \mathbf{X}^{(1)}_{P[N]}(\mathbf{x}) - \mathbf{x} \right\|_\infty^p \right] \right\}^{1/p}$$

$$\le \operatorname{diam}(\Omega) \cdot \left( E_P \left[ \left\{ \frac{dQ}{dP}(\mathbf{x}) \right\}^{2 \cdot p} \right] \right)^{1/(2 \cdot p)}. \tag{73}$$

*proof of Corollary C.14.* First from Theorem C.12 when $\kappa = 2 \cdot p$ and $\mu = P$,

$$E_P \left\| \mathbf{X}^{(1)}_{P[N]}(\mathbf{x}) - \mathbf{x} \right\|_\infty^{2 \cdot p} \le \operatorname{diam}(\Omega)^{2 \cdot p} \cdot \left( \frac{1}{N+1} \right)^{2 \cdot p/d}, \quad \text{for all } N \ge 1. \tag{74}$$

Thus, for all $N \ge 1$,

$$\left\{ E_P \left\| \mathbf{X}^{(1)}_{P[N]}(\mathbf{x}) - \mathbf{x} \right\|_\infty^{2 \cdot p} \right\}^{1/(2 \cdot p)} \le \left\{ \operatorname{diam}(\Omega)^{2 \cdot p} \cdot \left( \frac{1}{N+1} \right)^{2 \cdot p/d} \right\}^{1/(2 \cdot p)}$$

$$= \operatorname{diam}(\Omega) \cdot \left( \frac{1}{N+1} \right)^{1/d} \tag{75}$$

Now, using Hölder's inequality, we have

$$E_P \left[ \left\{ \frac{dQ}{dP}(\mathbf{x}) \right\}^p \cdot \left\| \mathbf{X}^{(1)}_{P[N]}(\mathbf{x}) - \mathbf{x} \right\|_\infty^p \right]$$

$$\le \left( E_P \left[ \left\{ \frac{dQ}{dP}(\mathbf{x}) \right\}^{2 \cdot p} \right] \right)^{1/(2 \cdot p)} \cdot \left( E_P \left[ \left\| \mathbf{X}^{(1)}_{P[N]}(\mathbf{x}) - \mathbf{x} \right\|_\infty^{2 \cdot p} \right] \right)^{1/(2 \cdot p)}$$

$$\le \left( E_P \left[ \left\{ \frac{dQ}{dP}(\mathbf{x}) \right\}^{2 \cdot p} \right] \right)^{1/(2 \cdot p)} \cdot \operatorname{diam}(\Omega) \cdot \left( \frac{1}{N+1} \right)^{1/d} \tag{76}$$

Taking $\overline{\lim}_{N \to \infty}$ on both sides of the above inequality, we have

$$\overline{\lim_{N \to \infty}} N^{1/d} \cdot \left\{ E_P \left[ \left\{ \frac{dQ}{dP}(\mathbf{x}) \right\}^p \cdot \left\| \mathbf{X}^{(1)}_{P[N]}(\mathbf{x}) - \mathbf{x} \right\|_\infty^p \right] \right\}^{1/p}$$

$$\le \overline{\lim_{N \to \infty}} \left\{ N^{1/d} \cdot \left( E_P \left[ \left\{ \frac{dQ}{dP}(\mathbf{x}) \right\}^{2 \cdot p} \right] \right)^{1/(2 \cdot p)} \cdot \operatorname{diam}(\Omega) \cdot \left( \frac{1}{N+1} \right)^{1/d} \right\}$$

$$= \operatorname{diam}(\Omega) \cdot \left( E_P \left[ \left\{ \frac{dQ}{dP}(\mathbf{x}) \right\}^{2 \cdot p} \right] \right)^{1/(2 \cdot p)} \tag{77}$$

This completes the proof. $\qquad\square$

**Lemma C.15.** *Let $\mu$ be a probability measure on $\mathbb{R}^d$ with $d \ge 1$. Assume that $\mu \ll \lambda$, where $\lambda$ denotes the Lebesgue measure on $\mathbb{R}^d$. Let $\| \cdot \|_\infty$ denote the maximum norm in $\mathbb{R}^d$: $\|\mathbf{y} - \mathbf{x}\|_\infty = \max_{1 \le i \le d} |y^i - x^i|$, where $\mathbf{y} = (y^1, y^2, \ldots, y^N)$ and $\mathbf{x} = (x^1, x^2, \ldots, x^N)$. Additionally, let $\Delta(\mathbf{x}, r)$ denote the $d$-dimensional interval centered at $\mathbf{x}$ with each side of length $r$: $\Delta(\mathbf{x}, r) = \{\mathbf{x}' \in \mathbb{R}^d \mid \|\mathbf{x}' - \mathbf{x}\|_\infty \le r/2\}$.*

*Then, for any interior point $\mathbf{x}$ in $\Omega$,*

$$\mu\big(\Delta(\mathbf{x}, r)\big) = \frac{d\mu}{d\lambda}(\mathbf{x}) \cdot r^d + o\left(r^d\right), \quad \text{as } r \to 0, \tag{78}$$

*where $f(r) = o(g(r))$, as $r \to 0$, denotes asymptotic domination such that $\lim_{r \to 0} f(r)/g(r) = 0$.*

*proof of Lemma C.15.* Note that, if $\mathbf{x}$ is an interior point in $\Omega$, it holds that

$$\lim_{r \to \infty} \frac{\mu\big(\Delta(\mathbf{x}, r)\big)}{\lambda\big(\Delta(\mathbf{x}, r)\big)} = \frac{d\mu}{d\lambda}(\mathbf{x}). \tag{79}$$

From Equation (79), we have

$$\begin{aligned}
\lim_{r \to \infty} \frac{\mu\big(\Delta(\mathbf{x}, r)\big)}{r^d} &= \lim_{r \to \infty} \frac{\mu\big(\Delta(\mathbf{x}, r)\big)}{r^d} \\
&= \lim_{r \to \infty} \frac{\mu\big(\Delta(\mathbf{x}, r)\big)}{\lambda\big(\Delta(\mathbf{x}, r)\big)} \tag{80} \\
&= \frac{d\mu}{d\lambda}(\mathbf{x}). \tag{81}
\end{aligned}$$

Here, we use an equation where $\lambda(\Delta(\mathbf{x}, r)) = r^d$ in Equation (80).

From Equation (81), we observe that

$$\mu\big(\Delta(\mathbf{x}, r)\big) = \frac{d\mu}{d\lambda}(\mathbf{x}) \cdot r^d + o\left(r^d\right), \quad \text{as } r \to 0. \tag{82}$$

This completes the proof. □

**Corollary C.16.** *Assume the same assumptions as in Lemma C.15. Let $\mathbf{X}$ be a random variable drawn from $\mu$, and let $E_{\mathbf{X}}$ denote the expectation with respect to $\mathbf{X}$.*

*Then, for any interior point $\mathbf{x}_0$ in $\Omega$,*

$$E_{\mathbf{X}}\left[\|\mathbf{x}_0 - \mathbf{X}\|_{\infty}^p \cdot I\big(\Delta(\mathbf{x}_0, r)\big)(\mathbf{X})\right] = \frac{d\mu}{d\lambda}(\mathbf{x}_0) \cdot r^{p+d+1} + o\left(r^{p+d+1}\right), \quad \text{as } r \to 0, \tag{83}$$

*where $I\big(A\big)(\cdot)$ is the indicator function for A: $I\big(A\big)(\mathbf{x}) = 1$ if $\mathbf{x} \in A$, and $0$ otherwise.*

*proof of Corollary C.16.* Consider the integration variable from $\mathbf{x}$ to $r$ such that

$$\|\mathbf{x}_0 - \mathbf{x}\|_{\infty}^p = r. \tag{84}$$

Then, from Lemma C.15, we have, as $r \to 0$,

$$I\big(\Delta(\mathbf{x}_0, r)\big)(\mathbf{x}) \cdot \frac{d\mu}{d\lambda}(\mathbf{x}) \, d\mathbf{x} = \frac{d\mu}{d\lambda}(\mathbf{x}_0) \cdot r^d + o\left(r^d\right). \tag{85}$$

From the definition of expectation with the density $d\mu/d\lambda$ and Equation (85), we have, as $r \to 0$,

$$\begin{aligned}
&E_{\mathbf{X}}\left[\|\mathbf{x}_0 - \mathbf{X}\|_{\infty}^p \cdot I\big(\Delta(\mathbf{x}_0, r)\big)(\mathbf{X})\right] \\
&= \int \|\mathbf{x}_0 - \mathbf{x}\|_{\infty}^p \cdot I\big(\Delta(\mathbf{x}_0, r)\big)(\mathbf{x}) \cdot \frac{d\mu}{d\lambda}(\mathbf{x}) \, d\mathbf{x} \\
&= \int r^p \cdot \left(\frac{d\mu}{d\lambda}(\mathbf{x}_0) \cdot r^d + o\left(r^d\right)\right) dr \\
&= \frac{d\mu}{d\lambda}(\mathbf{x}_0) \cdot r^{p+d+1} + o\left(r^{p+d+1}\right). \tag{86}
\end{aligned}$$

This completes the proof. □

**Theorem C.17** (Theorem 4.4 restated). *Let $P$ and $Q$ be probability measures on a compact set $\Omega$ in $\mathbb{R}^d$ with $d \geq 1$. Assume that $P \ll \lambda$ and $Q \ll \lambda$, where $\lambda$ denotes the Lebesgue measure on $\mathbb{R}^d$. Let $p$ be a positive constant such that $p \geq 1$. Assume $E[(dQ/dP)^p] < \infty$.*

*Then,*

$$\lim_{N \to \infty} N^{1/d} \cdot \left\{ E_{\hat{\mathbf{X}}_{P[N]}} \left[ E_P \left[ \left\{ \frac{dQ}{dP}\left(\mathbf{X}^{(1)}_{P[N]}(\mathbf{x})\right) \right\}^p \cdot \left\|\mathbf{X}^{(1)}_{P[N]}(\mathbf{x}) - \mathbf{x}\right\|_{\infty}^p \right] \right] \right\}^{1/p}$$

$$\geq e^{-1} \cdot \left\{ E_P \left[ \left\{ \frac{dQ}{dP}(\mathbf{x}) \right\}^p \right] \right\}^{1/p}, \tag{87}$$

*where $E_{\hat{\mathbf{X}}_{P[N]}}[\cdot]$ denotes the expectation on each variable in $\hat{\mathbf{X}}_{P[N]} = \{\mathbf{X}_P^1, \mathbf{X}_P^2, \ldots, \mathbf{X}_P^N\}$.*

*proof of Theorem C.17.* Let

$$B_i = \left\{ \mathbf{x} \in \Omega \,\Big|\, \left\| \mathbf{X}_P^i - \mathbf{x} \right\|_\infty \leq \left( \frac{1}{N} \right)^{1/d} \right\}. \tag{88}$$

Since $\mathbf{X}_{P[N]}^{(1)}(\mathbf{x})$ is the nearest neighbor in $\{\mathbf{X}_P^1, \mathbf{X}_P^2, \ldots, \mathbf{X}_P^N\}$ for $\mathbf{x}$,

$$1 \leq \exists i \leq N \ \text{ s.t. } \ \left\| \mathbf{X}_P^i - \mathbf{x} \right\|_\infty \leq \left( \frac{1}{N} \right)^{1/d}$$

$$\iff \qquad \left\| \mathbf{X}_{P[N]}^{(1)}(\mathbf{x}) - \mathbf{x} \right\|_\infty \leq \left( \frac{1}{N} \right)^{1/d} \tag{89}$$

Thus,

$$\left\{ \mathbf{x} \in \Omega \,\Big|\, \left\| \mathbf{X}_{P[N]}^{(1)}(\mathbf{x}) - \mathbf{x} \right\|_\infty \leq \left( \frac{1}{N} \right)^{1/d} \right\}$$

$$= \bigcup_{i=1}^N \left\{ \mathbf{x} \in \Omega \,\Big|\, \left\| \mathbf{X}_P^i - \mathbf{x} \right\|_\infty \leq \left( \frac{1}{N} \right)^{1/d} \right\} = \bigcup_{i=1}^N B_i \tag{90}$$

Next, define

$$Z_N(\mathbf{x}) = \sum_{i=1}^N I\left(B_i\right)(\mathbf{x}). \tag{91}$$

Let $\mathbf{X}_P$ be a random variable drawn from $P$ with $\mathbf{X}_P \perp\!\!\!\perp \mathbf{X}_P^i$, for $1 \leq i \leq N$.

From Lemma C.15,

$$P\big(I\left(B_i\right)(\mathbf{X}_P) = 1\big) = P\big(B_i\big)$$

$$= \frac{dP}{d\lambda}(\mathbf{X}_P) \cdot \left( \frac{1}{N^{1/d}} \right)^d + o\left( \frac{1}{N^{1/d}} \right)^d$$

$$= \frac{dP}{d\lambda}(\mathbf{X}_P) \cdot \frac{1}{N} + o\left( \frac{1}{N} \right)$$

$$= \frac{1}{N} + o\left( \frac{1}{N} \right), \tag{92}$$

and $I\left(B_i\right)(\mathbf{X}_P) \in \{0, 1\}$ and $I\left(B_i\right)(\mathbf{X}_P) \perp\!\!\!\perp I\left(B_j\right)(\mathbf{X}_P)$ for $i \neq j$. Namely, $Z_N\left(\mathbf{X}_P\right)$ follows a binomial distribution with $N$ trials and a success probability for each trial of $1/N + o(1/N)$.

Then, we obtain

$$E_{\hat{\mathbf{X}}_{P[N]}} \left[ I\big(\{Z_N(\mathbf{X}_P) = 0\}\big) \right] = \left( 1 - \frac{1}{N} - o\left( \frac{1}{N} \right) \right)^N. \tag{93}$$

Since $\lim_{N \to \infty} 1 - \frac{1}{N} - o\left( \frac{1}{N} \right) = 1$, we have

$$\left( 1 - \frac{1}{N} - o\left( \frac{1}{N} \right) \right)^N = \left( 1 - \frac{1}{N} - o\left( \frac{1}{N} \right) \right)^{N-1},$$

as $N \longrightarrow \infty$.

Thus,

$$E_{\hat{\mathbf{X}}_{P[N]}}\left[I\left(\{Z_N(\mathbf{X}_P)=0\}\right)\right] = \left(1 - \frac{1}{N} - o\left(\frac{1}{N}\right)\right)^{N-1} \quad \text{(as } N \longrightarrow \infty). \tag{94}$$

In addition, note that

$$Z_N(\mathbf{x}) \geq I\left(\bigcup_{i=1}^N B_i\right)(\mathbf{x}),$$

and

$$Z_N(\mathbf{x}) \geq 1 \implies I\left(\bigcup_{i=1}^N B_i\right)(\mathbf{x}) = 1.$$

In particular,

$$Z_N(\mathbf{x}) = 1 \implies \sum_{i=1}^N I\left(B_i\right)(\mathbf{x}) = 1.$$

Therefore,

$$Z_N(\mathbf{x}) = 1 \iff \sum_{i=1}^N I\left(B_i\right)(\mathbf{x}) = 1. \tag{95}$$

Now, we obtain

$$N^{p/d} \cdot E_P\left[\left\{\frac{dQ}{dP}\left(\mathbf{X}_{P[N]}^{(1)}(\mathbf{x})\right)\right\}^p \cdot \left\|\mathbf{X}_{P[N]}^{(1)}(\mathbf{x}) - \mathbf{x}\right\|_\infty^p\right]$$

$$\geq N^{p/d} \cdot E_P\left[\left\{\frac{dQ}{dP}\left(\mathbf{X}_{P[N]}^{(1)}(\mathbf{x})\right)\right\}^p \cdot \left\|\mathbf{X}_{P[N]}^{(1)}(\mathbf{x}) - \mathbf{x}\right\|_\infty^p\right.$$

$$\times I\left(\left\{\mathbf{x} \in \Omega \mid \left\|\mathbf{X}_{P[N]}^{(1)}(\mathbf{x}) - \mathbf{x}\right\|_\infty \leq \left(\frac{1}{N}\right)^{1/d}\right\}\right)$$

$$\left. \times I\left(\left\{\mathbf{x} \in \Omega \mid Z_N(\mathbf{x}) = 1\right\}\right)\right]$$

$$= N^{p/d} \cdot E_P\left[\left\{\frac{dQ}{dP}\left(\mathbf{X}_{P[N]}^{(1)}(\mathbf{x})\right)\right\}^p \cdot \left\|\mathbf{X}_{P[N]}^{(1)}(\mathbf{x}) - \mathbf{x}\right\|_\infty^p\right.$$

$$\left. \times I\left(\bigcup_{i=1}^N B_i\right) \cdot I\left(\left\{\mathbf{x} \in \Omega \mid Z_N(\mathbf{x}) = 1\right\}\right)\right] \qquad \text{(by Equation (90))}$$

$$= N^{p/d} \cdot E_P\left[\left\{\frac{dQ}{dP}\left(\mathbf{X}_{P[N]}^{(1)}(\mathbf{x})\right)\right\}^p \cdot \left\|\mathbf{X}_{P[N]}^{(1)}(\mathbf{x}) - \mathbf{x}\right\|_\infty^p\right.$$

$$\left. \times \sum_{i=1}^N I\left(B_i\right) \cdot I\left(\left\{\mathbf{x} \in \Omega \mid Z_N(\mathbf{x}) = 1\right\}\right)\right] \qquad \text{(by Equation (95))}$$

$$= N^{p/d} \cdot \sum_{i=1}^N E_P\left[\left\{\frac{dQ}{dP}\left(\mathbf{X}_{P[N]}^{(1)}(\mathbf{x})\right)\right\}^p \cdot \left\|\mathbf{X}_{P[N]}^{(1)}(\mathbf{x}) - \mathbf{x}\right\|_\infty^p\right.$$

$$\left. \times I\left(B_i\right) \cdot I\left(\left\{\mathbf{x} \in \Omega \mid Z_N(\mathbf{x}) = 1\right\}\right)\right]$$

$$= N^{p/d} \cdot \sum_{i=1}^N E_P\left[\left\{\frac{dQ}{dP}\left(\mathbf{X}_P^i\right)\right\}^p \cdot \left\|\mathbf{X}_P^i - \mathbf{x}\right\|_\infty^p\right.$$

$$\left. \times I\left(B_i\right) \cdot I\left(\left\{\mathbf{x} \in \Omega \mid Z_N(\mathbf{x}) = 1\right\}\right)\right]. \tag{96}$$

Now, let

$$Z_N^{-j}(\mathbf{x}) = \sum_{i \neq j}^{N} I(B_i)(\mathbf{x}).$$

Then,

$$I(B_i) \cdot I\left(\left\{\mathbf{x} \in \Omega \mid Z_N(\mathbf{x}) = 1\right\}\right) = I(B_i) \cdot I\left(\left\{\mathbf{x} \in \Omega \mid Z_N^{-i}(\mathbf{x}) = 0\right\}\right).$$

(97)

Additionally, let $\hat{\mathbf{X}}_{P[N]}^{-i}$ denote the subset of $\hat{\mathbf{X}}_{P[N]}$ excluding $\mathbf{X}_P^i$. i.e., $\hat{\mathbf{X}}_{P[N]}^{-i} = \hat{\mathbf{X}}_{P[N]} \setminus \{\mathbf{X}_P^i\}$. Let $E_N^{-i}[\cdot]$ denote the expectation over the variables in $\hat{\mathbf{X}}_{P[N]}^{-i}$, which is equivalent to $E_{\hat{\mathbf{X}}_{P[N]}^{-i}}$.

From Equation (94),

$$E_N^{-i}\left[I(B_i) \cdot I\left(\left\{\mathbf{x} \in \Omega \mid Z_N^{-i}(\mathbf{x}) = 0\right\}\right)\right] = \left(1 - \frac{1}{N-1} - o\left(\frac{1}{N-1}\right)\right)^{N-2}.$$

(98)

From Equations (97) and (98), we have

$$E_N^{-i}\left[E_P\left[\left\{\frac{dQ}{dP}(\mathbf{X}_P^i)\right\}^p \cdot \left\|\mathbf{X}_P^i - \mathbf{x}\right\|_\infty^p \times I(B_i) \cdot I\left(\left\{\mathbf{x} \in \Omega \mid Z_N(\mathbf{x}) = 1\right\}\right)\right]\right]$$

$$= E_N^{-i}\left[E_P\left[\left\{\frac{dQ}{dP}(\mathbf{X}_P^i)\right\}^p \cdot \left\|\mathbf{X}_P^i - \mathbf{x}\right\|_\infty^p \times I(B_i) \cdot I\left(\left\{\mathbf{x} \in \Omega \mid Z_N(\mathbf{x})^{-i} = 0\right\}\right)\right]\right]$$

$$= E_P\left[\left\{\frac{dQ}{dP}(\mathbf{X}_P^i)\right\}^p \cdot \left\|\mathbf{X}_P^i - \mathbf{x}\right\|_\infty^p \times I(B_i) \cdot E_N^{-i}\left[I\left(\left\{\mathbf{x} \in \Omega \mid Z_N(\mathbf{x})^{-i} = 0\right\}\right)\right]\right]$$

$$= E_P\left[\left\{\frac{dQ}{dP}(\mathbf{X}_P^i)\right\}^p \cdot \left\|\mathbf{X}_P^i - \mathbf{x}\right\|_\infty^p \times I(B_i) \times \left(1 - \frac{1}{N-1} - o\left(\frac{1}{N-1}\right)\right)^{N-2}\right]$$

$$= \left(1 - \frac{1}{N-1} - o\left(\frac{1}{N-1}\right)\right)^{N-2} \times E_P\left[\left\{\frac{dQ}{dP}(\mathbf{X}_P^i)\right\}^p \cdot \left\|\mathbf{X}_P^i - \mathbf{x}\right\|_\infty^p \times I(B_i)\right].$$

(99)

From Corollary C.16, we have

$$E_P\left[\left\{\frac{dQ}{dP}(\mathbf{X}_P^i)\right\}^p \cdot \left\|\mathbf{X}_P^i - \mathbf{x}\right\|_\infty^p \times I(B_i)\right]$$

$$= \left\{\frac{dQ}{dP}(\mathbf{X}_P^i)\right\}^p \cdot \left\{\frac{dP}{d\mu}(\mathbf{X}_P^i) \cdot \left(\frac{1}{N^{1/d}}\right)^{p+d+1} + o\left(\left(\frac{1}{N^{1/d}}\right)^{p+d+1}\right)\right\}$$

$$= \frac{dP}{d\mu}(\mathbf{X}_P^i) \cdot \left\{\frac{dQ}{dP}(\mathbf{X}_P^i)\right\}^p \cdot \left(\frac{1}{N}\right)^{1+p/d} + o\left(\left(\frac{1}{N}\right)^{1+p/d}\right)$$

$$= \left\{\frac{dQ}{dP}(\mathbf{X}_P^i)\right\}^p \cdot \left(\frac{1}{N}\right)^{1+p/d} + o\left(\left(\frac{1}{N}\right)^{1+p/d}\right).$$

(100)

From Equations (99) and (100), we obtain

$$E_N^{-i}\left[E_P\left[\left\{\frac{dQ}{dP}(\mathbf{X}_P^i)\right\}^p \cdot \left\|\mathbf{X}_P^i - \mathbf{x}\right\|_\infty^p \times I(B_i) \cdot I\left(\left\{\mathbf{x} \in \Omega \mid Z_N(\mathbf{x}) = 1\right\}\right)\right]\right]$$

$$= \left(1 - \frac{1}{N-1} - o\left(\frac{1}{N-1}\right)\right)^{N-2}$$

$$\times \left\{ \frac{dQ}{dP} \left( \mathbf{X}_P^i \right) \right\}^p \cdot \left( \frac{1}{N} \right)^{1+p/d} + o \left( \left( \frac{1}{N} \right)^{1+p/d} \right). \tag{101}$$

From Equations (96) and (101), we obtain, as $N \longrightarrow \infty$,

$$N^{p/d} \cdot E_{\hat{\mathbf{X}}_{P[N]}} \left[ E_P \left[ \left\{ \frac{dQ}{dP} \left( \mathbf{X}_{P[N]}^{(1)}(\mathbf{x}) \right) \right\}^p \cdot \left\| \mathbf{X}_{P[N]}^{(1)}(\mathbf{x}) - \mathbf{x} \right\|_\infty^p \right] \right]$$

$$\geq N^{p/d} \cdot E_{\hat{\mathbf{X}}_{P[N]}} \left[ \sum_{i=1}^N E_P \left[ \left\{ \frac{dQ}{dP} \left( \mathbf{X}_P^i \right) \right\}^p \cdot \left\| \mathbf{X}_P^i - \mathbf{x} \right\|_\infty^p \right. \right.$$

$$\left. \left. \times \, I\left(B_i\right) \cdot I \left( \left\{ \mathbf{x} \in \Omega \;\middle|\; Z_N(\mathbf{x}) = 1 \right\} \right) \right] \right]$$

$$= \sum_{i=1}^N N^{p/d} \cdot E_{\mathbf{X}_P^i} \left[ E_N^{-i} \left[ E_P \left[ \left\{ \frac{dQ}{dP} \left( \mathbf{X}_P^i \right) \right\}^p \cdot \left\| \mathbf{X}_P^i - \mathbf{x} \right\|_\infty^p \right. \right. \right.$$

$$\left. \left. \left. \times \, I\left(B_i\right) \cdot I \left( \left\{ \mathbf{x} \in \Omega \;\middle|\; Z_N(\mathbf{x}) = 1 \right\} \right) \right] \right] \right] \qquad \text{(by Equation (96))}$$

$$= \sum_{i=1}^N N^{p/d} \cdot E_{\mathbf{X}_P^i} \left[ \left( 1 - \frac{1}{N-1} - o \left( \frac{1}{N-1} \right) \right)^{N-2} \right.$$

$$\left. \times \left\{ \frac{dQ}{dP} \left( \mathbf{X}_P^i \right) \right\}^p \cdot \left( \frac{1}{N} \right)^{1+p/d} + o \left( \left( \frac{1}{N} \right)^{1+p/d} \right) \right]$$

$$= N \cdot \left\{ \left( 1 - \frac{1}{N-1} - o \left( \frac{1}{N-1} \right) \right)^{N-2} \right.$$

$$\left. \times \, E_P \left[ \left\{ \frac{dQ}{dP} (\mathbf{x}) \right\}^p \right] \cdot \left( \frac{1}{N} \right) + o \left( \frac{1}{N} \right) \right\} \qquad \text{(by Equation (101))}$$

$$= \left( 1 - \frac{1}{N-1} - o \left( \frac{1}{N-1} \right) \right)^{N-2} \cdot \left\{ E_P \left[ \left\{ \frac{dQ}{dP} (\mathbf{x}) \right\}^p \right] + o\left(1\right) \right\}. \tag{102}$$

As $N \to \infty$, we observe

$$\left( 1 - \frac{1}{N-1} - o \left( \frac{1}{N-1} \right) \right)^{N-2} \longrightarrow e^{-1}. \tag{103}$$

Then, from Equation (102), we obtain

$$\lim_{N \to \infty} N^{p/d} \cdot E_{\hat{\mathbf{X}}_{P[N]}} \left[ E_P \left[ \left\{ \frac{dQ}{dP} \left( \mathbf{X}_{P[N]}^{(1)}(\mathbf{x}) \right) \right\}^p \cdot \left\| \mathbf{X}_{P[N]}^{(1)}(\mathbf{x}) - \mathbf{x} \right\|_\infty^p \right] \right]$$

$$\geq e^{-1} \cdot E_P \left[ \left\{ \frac{dQ}{dP} (\mathbf{x}) \right\}^p \right]. \tag{104}$$

This completes the proof. $\qquad\square$

**Theorem C.18.** *Assume that $f$ satisfies Assumption C.6. For $\widetilde{\mathcal{L}}_f^{(N)}(\phi)$ defined in Definition C.3, let*
$\phi_*^{(N)} = \arg\min_{\phi:\Omega \to \mathbb{R}_{>0}} \widetilde{\mathcal{L}}_f^{(N)}(\phi)$.

*Then, for any measurable function $\phi : \Omega \to \mathbb{R}_{>0}$, the following equivalence holds:*

$$\phi(\mathbf{X}_\mu^i) - \phi_*^{(N)}(\mathbf{X}_\mu^i) = O_p \left( \frac{1}{\sqrt{N}} \right), \quad \text{for } 1 \leq i \leq N$$

$$\iff \widetilde{\mathcal{L}}_f^{(N)}(\phi) - \min_{\phi:\Omega \to \mathbb{R}_{>0}} \bar{\mathcal{L}}_f(\phi) = O_p \left( \frac{1}{\sqrt{N}} \right), \tag{105}$$

*where $\{\mathbf{X}_\mu^1, \mathbf{X}_\mu^2, \ldots, \mathbf{X}_\mu^N\}$ is defined in Definition C.3, and $\bar{\mathcal{L}}_f(\phi)$ is defined in Lemma C.11.*

*proof of Theorem C.18.* First, we enumerate several facts used in this proof.

I. From the Central Limit Theorem, we have:

$$\widetilde{\mathcal{L}}_f^{(N)}\left(\phi_*^{(N)}\right) - E_\mu\left[\widetilde{\mathcal{L}}_f^{(N)}\left(\phi_*^{(N)}\right)\right] = O_p\left(\frac{1}{\sqrt{N}}\right). \tag{106}$$

II. From Proposition C.10, we have, for all $\mathbf{x} \in \hat{\mathbf{X}}_{\mu[N]}$:

$$\phi_*^{(N)}(\mathbf{x}) = \frac{dQ}{dP}(\mathbf{x}), \tag{107}$$

where $\hat{\mathbf{X}}_{\mu[N]}$ is defined in Definition C.3.

III. From Equation (107), it follows that:

$$\widetilde{\mathcal{L}}_f^{(N)}\left(\phi_*^{(N)}\right) = \widetilde{\mathcal{L}}_f^{(N)}\left(\frac{dQ}{dP}\right), \tag{108}$$

and

$$E_\mu\left[\widetilde{\mathcal{L}}_f^{(N)}\left(\phi_*^{(N)}\right)\right] = E_\mu\left[\widetilde{\mathcal{L}}_f^{(N)}\left(\frac{dQ}{dP}\right)\right]. \tag{109}$$

IV. From Lemma C.11, we have:

$$\min_{\phi:\Omega\to\mathbb{R}_{>0}} \bar{\mathcal{L}}_f(\phi) = \bar{\mathcal{L}}_f\left(\frac{dQ}{dP}\right) = E_\mu\left[\widetilde{\mathcal{L}}_f^{(N)}\left(\frac{dQ}{dP}\right)\right]. \tag{110}$$

V. From Lemma C.8, for $\widetilde{l}_f(u;\mathbf{x})$ defined in Equation (28), we obtain:

$$\frac{d}{du}\widetilde{l}_f\left(\frac{dQ}{dP}(\mathbf{x});\mathbf{x}\right) = 0, \tag{111}$$

and

$$\frac{d^2}{du^2}\widetilde{l}_f\left(\frac{dQ}{dP}(\mathbf{x});\mathbf{x}\right) = f''\left(\frac{dQ}{dP}(\mathbf{x})\right) \cdot \frac{dP}{d\mu}(\mathbf{x}). \tag{112}$$

VI. From Theorem C.9, we have:

$$\widetilde{l}_f\left(u;\mathbf{x}\right) - \widetilde{l}_f\left(\frac{dQ}{dP}(\mathbf{x});\mathbf{x}\right) = \frac{1}{2} \cdot f''\left(\frac{dQ}{dP}(\mathbf{x})\right) \cdot \frac{dP}{d\mu}(\mathbf{x}) \cdot \left|u - \frac{dQ}{dP}(\mathbf{x})\right|^2$$
$$+ o\left(\left|u - \frac{dQ}{dP}(\mathbf{x})\right|^2\right), \tag{113}$$

where $f(a) = o(a)$ (as $a \to 0$) denotes asymptotic domination such that $\lim_{a\to0} f(a)/a = 0$.

VII. From the assumption that $E_P[f''(dQ/dP)] < \infty$,

$$f''\left(\frac{dQ}{dP}(\mathbf{X}_\mu^i)\right) \cdot \frac{dP}{d\mu}(\mathbf{X}_\mu^i) = O_p\left(1\right), \quad \text{as } N \to \infty. \tag{114}$$

Now, we show the direction "$\implies$" in Equation (105).

Assume that $\phi(\mathbf{X}_\mu^i) = \phi_*^{(N)}(\mathbf{X}_\mu^i) + O_p\left(1/\sqrt{N}\right)$ for $1 \le i \le N$.

From Equations (35) in Theorem C.9 and (114), we have

$$\widetilde{l}_f\left(\phi(\mathbf{X}_\mu^i);\mathbf{X}_\mu^i\right) - \widetilde{l}_f\left(\phi_*^{(N)}(\mathbf{X}_\mu^i);\mathbf{X}_\mu^i\right)$$

$$
\begin{aligned}
&= \widetilde{l}_f\left(\phi(\mathbf{X}_\mu^i); \mathbf{X}_\mu^i\right) - \widetilde{l}_f\left(\frac{dQ}{dP}(\mathbf{X}_\mu^i); \mathbf{X}_\mu^i\right) \\
&= \frac{1}{2} \cdot f''\left(\frac{dQ}{dP}(\mathbf{X}_\mu^i)\right) \cdot \frac{dP}{d\mu}(\mathbf{X}_\mu^i) \cdot \left|\phi(\mathbf{X}_\mu^i) - \frac{dQ}{dP}(\mathbf{X}_\mu^i)\right|^2 + o\left(\left|\phi(\mathbf{X}_\mu^i) - \frac{dQ}{dP}(\mathbf{X}_\mu^i)\right|^2\right) \\
&= \frac{1}{2} \cdot f''\left(\frac{dQ}{dP}(\mathbf{X}_\mu^i)\right) \cdot \frac{dP}{d\mu}(\mathbf{X}_\mu^i) \cdot \left|\phi(\mathbf{X}_\mu^i) - \phi_*^{(N)}(\mathbf{X}_\mu^i)\right|^2 + o\left(\left|\phi(\mathbf{X}_\mu^i) - \phi_*^{(N)}(\mathbf{X}_\mu^i)\right|^2\right) \\
&= O_p\left(1\right) \cdot O_p\left(\left\{\frac{1}{\sqrt{N}}\right\}^2\right) \\
&= O_p\left(\frac{1}{N}\right).
\end{aligned}
\tag{115}
$$

Thus, we have:

$$
\begin{aligned}
\widetilde{\mathcal{L}}_f^{(N)}(\phi) - \widetilde{\mathcal{L}}_f^{(N)}\left(\phi_*^{(N)}\right) &= \frac{1}{N} \cdot \sum_{i=1}^{N}\left\{\widetilde{l}_f\left(\phi(\mathbf{X}_\mu^i); \mathbf{X}_\mu^i\right) - \widetilde{l}_f\left(\frac{dQ}{dP}(\mathbf{X}_\mu^i); \mathbf{X}_\mu^i\right)\right\} \\
&= \frac{1}{N} \cdot \sum_{i=1}^{N} O_p\left(\frac{1}{N}\right) \\
&= O_p\left(\frac{1}{N}\right).
\end{aligned}
\tag{116}
$$

From Equations (106), (108), (110), and (116), we obtain:

$$
\widetilde{\mathcal{L}}_f^{(N)}(\phi) - \min_{\phi:\Omega\to\mathbb{R}_{>0}} \bar{\mathcal{L}}_f(\phi)
$$

$$
\begin{aligned}
&= \left\{\widetilde{\mathcal{L}}_f^{(N)}(\phi) - \widetilde{\mathcal{L}}_f^{(N)}\left(\phi_*^{(N)}\right)\right\} + \left\{\widetilde{\mathcal{L}}_f^{(N)}\left(\phi_*^{(N)}\right) - \min_{\phi:\Omega\to\mathbb{R}_{>0}} \bar{\mathcal{L}}_f(\phi)\right\} \\
&= \left\{\widetilde{\mathcal{L}}_f^{(N)}(\phi) - \widetilde{\mathcal{L}}_f^{(N)}\left(\phi_*^{(N)}\right)\right\} + \left\{\widetilde{\mathcal{L}}_f^{(N)}\left(\frac{dQ}{dP}\right) - \min_{\phi:\Omega\to\mathbb{R}_{>0}} \bar{\mathcal{L}}_f(\phi)\right\} \quad \text{(by Equation (108))} \\
&= \left\{\widetilde{\mathcal{L}}_f^{(N)}(\phi) - \widetilde{\mathcal{L}}_f^{(N)}\left(\phi_*^{(N)}\right)\right\} + \left\{\widetilde{\mathcal{L}}_f^{(N)}\left(\frac{dQ}{dP}\right) - E\left[\widetilde{\mathcal{L}}_f^{(N)}\left(\frac{dQ}{dP}\right)\right]\right\} \quad \text{(by Equation (110))} \\
&= \left\{\widetilde{\mathcal{L}}_f^{(N)}(\phi) - \widetilde{\mathcal{L}}_f^{(N)}\left(\phi_*^{(N)}\right)\right\} + \left\{\widetilde{\mathcal{L}}_f^{(N)}\left(\phi_*^{(N)}\right) - E\left[\widetilde{\mathcal{L}}_f^{(N)}\left(\phi_*^{(N)}\right)\right]\right\} \quad \text{(by Equation (108))} \\
&= O_p\left(\frac{1}{N}\right) + O_p\left(\frac{1}{\sqrt{N}}\right) \quad \text{(by Equations (106) and (116))} \\
&= O_p\left(\frac{1}{\sqrt{N}}\right).
\end{aligned}
\tag{117}
$$

Thus, we have proved "$\Longrightarrow$".

Next, we prove the direction "$\Longleftarrow$" in Equation (105).

Suppose

$$
\widetilde{\mathcal{L}}_f^{(N)}(\phi) - \min_{\phi:\Omega\to\mathbb{R}_{>0}} \bar{\mathcal{L}}_f(\phi) = O_p\left(\frac{1}{\sqrt{N}}\right).
\tag{118}
$$

From Equations (106), (110), (109), and (118), we obtain

$$
\widetilde{\mathcal{L}}_f^{(N)}(\phi) - \widetilde{\mathcal{L}}_f^{(N)}\left(\phi_*^{(N)}\right)
$$

$$
= \left\{\widetilde{\mathcal{L}}_f^{(N)}(\phi) - \min_{\phi:\Omega\to\mathbb{R}_{>0}} \bar{\mathcal{L}}_f(\phi)\right\} + \left\{\min_{\phi:\Omega\to\mathbb{R}_{>0}} \bar{\mathcal{L}}_f(\phi) - \widetilde{\mathcal{L}}_f^{(N)}\left(\phi_*^{(N)}\right)\right\}
$$

$$= \left\{ \widetilde{\mathcal{L}}_f^{(N)}(\phi) - \min_{\phi:\Omega \to \mathbb{R}_{>0}} \bar{\mathcal{L}}_f(\phi) \right\} + \left\{ E\left[ \widetilde{\mathcal{L}}_f^{(N)} \left( \frac{dQ}{dP} \right) \right] - \widetilde{\mathcal{L}}_f^{(N)} \left( \phi_*^{(N)} \right) \right\} \qquad \text{(by Equation (110))}$$

$$= \left\{ \widetilde{\mathcal{L}}_f^{(N)}(\phi) - \min_{\phi:\Omega \to \mathbb{R}_{>0}} \bar{\mathcal{L}}_f(\phi) \right\} + \left\{ E\left[ \widetilde{\mathcal{L}}_f^{(N)} \left( \phi_*^{(N)} \right) \right] - \widetilde{\mathcal{L}}_f^{(N)} \left( \phi_*^{(N)} \right) \right\} \qquad \text{(by Equation (109))}$$

$$= O_p\left( \frac{1}{\sqrt{N}} \right) + O_p\left( \frac{1}{\sqrt{N}} \right) \qquad \text{(by Equations (106) and (118))}$$

$$= O_p\left( \frac{1}{\sqrt{N}} \right). \tag{119}$$

From Equation (107), we have

$$\widetilde{\mathcal{L}}_f^{(N)}(\phi) - \widetilde{\mathcal{L}}_f^{(N)} \left( \phi_*^{(N)} \right) = \frac{1}{N} \cdot \sum_{i=1}^{N} \widetilde{l}_f \left( \phi(\mathbf{X}_\mu^i); \mathbf{X}_\mu^i \right) - \frac{1}{N} \cdot \sum_{i=1}^{N} \widetilde{l}_f \left( \phi_*^{(N)}(\mathbf{X}_\mu^i); \mathbf{X}_\mu^i \right)$$

$$= \frac{1}{N} \cdot \sum_{i=1}^{N} \left\{ \widetilde{l}_f \left( \phi(\mathbf{X}_\mu^i); \mathbf{X}_\mu^i \right) - \widetilde{l}_f \left( \phi_*^{(N)}(\mathbf{X}_\mu^i); \mathbf{X}_\mu^i \right) \right\}. \tag{120}$$

From Equations (119) and (120), we have

$$\frac{1}{N} \cdot \sum_{i=1}^{N} \left\{ \widetilde{l}_f \left( \phi(\mathbf{X}_\mu^i); \mathbf{X}_\mu^i \right) - \widetilde{l}_f \left( \frac{dQ}{dP}(\mathbf{X}_\mu^i); \mathbf{X}_\mu^i \right) \right\} = O_p\left( \frac{1}{\sqrt{N}} \right). \tag{121}$$

Let $a_N^i = E_P \left[ \left| \phi(\mathbf{X}_\mu^i) - \phi_*^{(k)}(\mathbf{X}_\mu^i) \right| \right]$. Since $\mathbf{X}_\mu^i$ is identically distributed for $1 \le i \le N$, we have $a_N^i = a_N^1$ for any $1 \le i \le N$. Thus, define $A_N = \sup_{k \ge N} a_k^i = \sup_{k \ge N} a_k^1$.

Using Chebyshev's inequality, we have for any $\varepsilon > 0$,

$$P\left( \left| \phi(\mathbf{X}_\mu^i) - \phi_*^{(k)}(\mathbf{X}_\mu^i) \right| / A_N > \frac{1}{\varepsilon} \right)$$

$$\le \frac{\varepsilon \cdot E_P \left[ \left| \phi(\mathbf{X}_\mu^i) - \phi_*^{(k)}(\mathbf{X}_\mu^i) \right| \right]}{A_N}$$

$$\le \frac{\varepsilon \cdot a_N^i}{A_N}$$

$$\le \varepsilon. \tag{122}$$

Thus, $\phi(\mathbf{X}_\mu^i) - \phi_*^{(k)}(\mathbf{X}_\mu^i) = O_p(A_N)$.

Now, we calculate

$$\frac{1}{N} \sum_{i=1}^{N} \left\{ \widetilde{l}_f \left( \phi(\mathbf{X}_\mu^i); \mathbf{X}_\mu^i \right) - \widetilde{l}_f \left( \phi_*^{(N)}(\mathbf{X}_\mu^i); \mathbf{X}_\mu^i \right) \right\}$$

$$= \frac{1}{N} \sum_{i=1}^{N} \left\{ \widetilde{l}_f \left( \phi(\mathbf{X}_\mu^i); \mathbf{X}_\mu^i \right) - \widetilde{l}_f \left( \frac{dQ}{dP}(\mathbf{X}_\mu^i); \mathbf{X}_\mu^i \right) \right\}$$

$$= \frac{1}{N} \sum_{i=1}^{N} \left\{ \frac{1}{2} \cdot \lambda(\mathbf{X}_\mu^i) \cdot O_p\left( \left| \phi(\mathbf{X}_\mu^i) - \frac{dQ}{dP}(\mathbf{X}_\mu^i) \right|^2 \right) + o_p\left( \left| \phi(\mathbf{X}_\mu^i) - \frac{dQ}{dP}(\mathbf{X}_\mu^i) \right|^4 \right) \right\}$$

$$= \frac{1}{N} \sum_{i=1}^{N} \left\{ \frac{1}{2} \cdot \lambda(\mathbf{X}_\mu^i) \cdot O_p\left( \left| \phi(\mathbf{X}_\mu^i) - \phi_*^{(N)}(\mathbf{X}_\mu^i) \right|^2 \right) + o_p\left( \left| \phi(\mathbf{X}_\mu^i) - \phi_*^{(N)}(\mathbf{X}_\mu^i) \right|^4 \right) \right\}$$

$$= \frac{1}{N} \cdot N \cdot \frac{1}{2} \cdot O_p\left( \sqrt{N} \right) \cdot O_p\left( A_N^2 \right) + \frac{1}{N} \cdot N \cdot \frac{1}{2} \cdot o_p\left( A_N^4 \right)$$

$$= O_p\left(\sqrt{N}\right) \cdot O_p\left(A_N^2\right) + o_p\left(A_N^4\right). \tag{123}$$

Here, $\mathbf{X} = o_p(a_N)$ denotes the convergence in probability with rate $a_N$ in $\mu$ as $N \to \infty$: $\mathbf{X} = o_p(a_N)$ (as $N \to \infty$) $\Leftrightarrow \forall \varepsilon, \forall \delta > 0, \exists N(\varepsilon, \delta) > 0$ such that $\mu(|\mathbf{X}|/a_N \geq \delta) < \varepsilon$ for $\forall N \geq N(\varepsilon, \delta)$.

From Equations (121) and (123), we have

$$O_p\left(\frac{1}{\sqrt{N}}\right) \geq O_p\left(\sqrt{N}\right) \cdot O_p\left(A_N^2\right) + o_p\left(A_N^4\right). \tag{124}$$

From the definition of $A_N$, we observe that $A_N$ decreases as $N$ increases. Thus, $\lim_{N\to\infty} A_N$ exists and $0 \leq \lim_{N\to\infty} A_N < \infty$.

Suppose that $\lim_{N\to\infty} A_N > 0$. Then, we have

$$O_p\left(\sqrt{N}\right) \cdot O_p\left(A_N^2\right) + o_p\left(A_N^4\right) = O_p\left(\sqrt{N}\right) + o_p(1). \tag{125}$$

This contradicts Equation (124). Therefore, $\lim_{N\to\infty} A_N = 0$.

From Equation (124), we have

$$O_p\left(\frac{1}{N}\right) \geq O_p\left(A_N^2\right) + o_p\left(\frac{A_N^4}{\sqrt{N}}\right)$$
$$= O_p\left(A_N^2\right). \tag{126}$$

Thus, $A_N = O\left(1/\sqrt{N}\right)$.

Finally, we have

$$\phi(\mathbf{X}_\mu^i) - \phi_*^{(N)}(\mathbf{X}_\mu^i) = O_p(A_N) = O_p\left(\frac{1}{\sqrt{N}}\right). \tag{127}$$

Here, we have proved the direction "$\Longleftarrow$".

This completes the proof. $\qquad\square$

**Corollary C.19** (Theorem 4.7 restated). *Assume the same assumption as in Theorem C.18. let $\phi_*^{(N)} = \arg\min_{\phi:\Omega\to\mathbb{R}_{>0}} \widetilde{\mathcal{L}}_f^{(N)}(\phi)$.*

*Then, for any measurable function $\phi : \Omega \to \mathbb{R}_{>0}$,*

$$\phi(\mathbf{X}_\mu^i) - \phi_*^{(N)}(\mathbf{X}_\mu^i) = O_p\left(\frac{1}{\sqrt{N}}\right), \quad \text{for } 1 \leq i \leq N.$$

$$\Longleftrightarrow \mathcal{L}_f^{(R,S)}(\phi) - \inf_{\phi:\Omega\to\mathbb{R}_{>0}} E_\mu\left[\mathcal{L}_f^{(R,S)}(\phi)\right] = O_p\left(\frac{1}{\sqrt{N}}\right), \tag{128}$$

*where $\{\mathbf{X}_\mu^1, \mathbf{X}_\mu^2, \ldots, \mathbf{X}_\mu^N\}$ is defined in Definition C.3, and $\mathcal{L}_f^{(R,S)}(\phi)$ is defined in Definition C.2.*

*proof of Corollary C.19.* From Lemma C.11, we have $\mathcal{L}_f^{(R,S)}(\phi) = \bar{\mathcal{L}}_f(\phi)$.

Thus, Equation (128) follows directly from Equation (105).

This completes the proof. $\qquad\square$

**Theorem C.20** (Theorem 4.5 restated). *Assume that $\Omega$ is a compact set in $\mathbb{R}^d$ with $d \geq 3$ and that $f$ satisfies Assumption C.6. Let $P$ and $Q$ be probability measures on $\Omega$. Assume that $P \ll \lambda$ and $Q \ll \lambda$, where $\lambda$ denotes the Lebesgue measure on $\mathbb{R}^d$. Let $T^*(\mathbf{x})$ be the energy function of $dQ/dP(\mathbf{x})$ defined as $T^*(\mathbf{x}) = -\log dQ/dP(\mathbf{x})$.*

Let $\widetilde{\mathcal{F}}_{K\text{-}Lip}^{(N)}$ denote the set of all K-Lipschitz continuous functions on $\Omega$ that minimize $\widetilde{\mathcal{L}}_f^{(N)}(\cdot)$. Specifically, define

$$\widetilde{\mathcal{F}}^{(N)} = \left\{ \phi_* : \Omega \to \mathbb{R}_{>0} \;\middle|\; \widetilde{\mathcal{L}}_f^{(N)}(\phi_*) = \min_\phi \widetilde{\mathcal{L}}_f^{(N)}(\phi) \right\}, \tag{129}$$

and

$$\mathcal{F}_{K\text{-}Lip} = \left\{ \phi : \Omega \to \mathbb{R}_{>0} \;\middle|\; \left|\phi(\mathbf{y}) - \phi(\mathbf{x})\right| \le K \cdot \left\|\mathbf{y} - \mathbf{x}\right\|_\infty \text{ for all } \mathbf{y}, \mathbf{x} \in \Omega \right\}. \tag{130}$$

Subsequently, let

$$\widetilde{\mathcal{F}}_{K\text{-}Lip}^{(N)} = \widetilde{\mathcal{F}}^{(N)} \cap \mathcal{F}_{K\text{-}Lip}. \tag{131}$$

**(Upper Bound)** *Assume Assumption C.4: there exists $L > 0$ such that $\left|T^*(\mathbf{y}) - T^*(\mathbf{x})\right| \le L \cdot \|\mathbf{y} - \mathbf{x}\|_\infty$ for any $\mathbf{y}, \mathbf{x} \in \Omega$, i.e., $T^*(\mathbf{x})$ is L-Lipschitz continuous on $\Omega$.*

*Then, Equation (132) holds for $1 \le p \le d/2$, such that for any $\phi \in \widetilde{\mathcal{F}}_{K\text{-}Lip}^{(N)}$,*

$$\varlimsup_{N\to\infty} N^{1/d} \cdot \left\{ E_P \left| \frac{dQ}{dP}(\mathbf{x}) - \phi(\mathbf{x}) \right|^p \right\}^{1/p}$$

$$\le L \cdot \mathrm{diam}(\Omega) \cdot \left\{ E_P \left[ \left\{ \frac{dQ}{dP}(\mathbf{x}) \right\}^{2\cdot p} \right] \right\}^{1/(2\cdot p)} + K \cdot \mathrm{diam}(\Omega). \tag{132}$$

**(Lower Bound)** *Assume Assumption C.5: there exists $L > 1$ such that $(1/L) \cdot \|\mathbf{y} - \mathbf{x}\|_\infty \le \left|T^*(\mathbf{y}) - T^*(\mathbf{x})\right| \le L \cdot \|\mathbf{y} - \mathbf{x}\|_\infty$ for any $\mathbf{y}, \mathbf{x} \in \Omega$, i.e., $T^*(\mathbf{x})$ is L-bi-Lipschitz continuous on $\Omega$; and $E_P\left[dQ/dP\right] < \infty$ with $1 \le p \le d$.*

*Then, Equation (133) holds for any $\phi \in \widetilde{\mathcal{F}}_{K\text{-}Lip}^{(N)}$, such that*

$$\lim_{N\to\infty} N^{1/d} \cdot E_{\hat{\mathbf{X}}_{P[N]}} \left[ \left\{ E_P \left| \frac{dQ}{dP}(\mathbf{x}) - \phi(\mathbf{x}) \right|^p \right\}^{1/p} \right]$$

$$\ge \frac{1}{L} \cdot \left\{ E_P \left[ \left\{ \frac{dQ}{dP}(\mathbf{x}) \right\}^p \right] \right\}^{1/p} - K \cdot \mathrm{diam}(\Omega) \tag{133}$$

$$\ge \frac{1}{L} \cdot e^{\frac{p-1}{p} \cdot KL(Q\|P)-1} - K \cdot \mathrm{diam}(\Omega) \tag{134}$$

*proof of Theorem C.20.* First, we list the equations used in this proof.

    I. Using the second-order Taylor expansion of $e^{-t}$, we have

$$e^{-t} = 1 - t + \frac{1}{2} \cdot e^{-c(t)} \cdot t^2, \quad \text{where } 0 \le |c(t)| \le |t|. \tag{135}$$

    II. From Equation (135), it follows that

$$\left| \frac{dQ}{dP}(\mathbf{y}) - \frac{dQ}{dP}(\mathbf{x}) \right|$$

$$= e^{-T^*(\mathbf{y})} \cdot \left| 1 - e^{T^*(\mathbf{y})-T^*(\mathbf{x})} \right|$$

$$= e^{-T^*(\mathbf{y})} \left\{ (T^*(\mathbf{y}) - T^*(\mathbf{x})) + \frac{1}{2} \cdot e^{C(\mathbf{y},\mathbf{x},T^*)} \cdot (T^*(\mathbf{y}) - T^*(\mathbf{x}))^2 \right\}$$

$$= \frac{dQ}{dP}(\mathbf{y}) \left\{ (T^*(\mathbf{y}) - T^*(\mathbf{x})) + \frac{1}{2} \cdot e^{C(\mathbf{y},\mathbf{x},T^*)} \cdot (T^*(\mathbf{y}) - T^*(\mathbf{x}))^2 \right\},$$

$$\text{where } 0 \le |C(\mathbf{y}, \mathbf{x}, T^*)| \le |T^*(\mathbf{y}) - T^*(\mathbf{x})|. \tag{136}$$

III. From Corollary C.13, for $0 \le p \le d/2$,

$$\overline{\lim_{N \to \infty}} \, N^{1/d} \cdot \left\{ E_P \left\| \mathbf{X}_{\mu[N]}^{(1)}(\mathbf{x}) - \mathbf{x} \right\|_\infty^p \right\}^{1/p} \le \operatorname{diam}(\Omega).$$

(137)

IV. From Corollary C.14, for $0 \le p \le d/2$,

$$\overline{\lim_{N \to \infty}} \, N^{1/d} \cdot \left\{ E_P \left[ \left\{ \frac{dQ}{dP}(\mathbf{x}) \right\}^p \cdot \left\| \mathbf{X}_{P[N]}^{(1)}(\mathbf{x}) - \mathbf{x} \right\|_\infty^p \right] \right\}^{1/p}$$

$$\le \operatorname{diam}(\Omega) \cdot \left\{ E_P \left[ \left\{ \frac{dQ}{dP}(\mathbf{x}) \right\}^{2 \cdot p} \right] \right\}^{1/(2 \cdot p)}$$

(138)

V. From Equation (138), for $0 \le p \le d/2$,

$$\overline{\lim_{N \to \infty}} \, N^{1/d} \cdot \left\{ E_P \left[ \left\{ \frac{dQ}{dP}(\mathbf{x}) \right\}^p \cdot \left\| \mathbf{X}_{P[N]}^{(1)}(\mathbf{x}) - \mathbf{x} \right\|_\infty^{2 \cdot p} \right] \right\}^{1/p}$$

$$\le \overline{\lim_{N \to \infty}} \, N^{1/d} \cdot \left\{ E_P \left[ \left\{ \frac{dQ}{dP}(\mathbf{x}) \right\}^{2 \cdot p} \right] \right\}^{1/(2 \cdot p)} \left\{ E_P \left[ \left\| \mathbf{X}_{P[N]}^{(1)}(\mathbf{x}) - \mathbf{x} \right\|_\infty^{4 \cdot p} \right] \right\}^{1/(2 \cdot p)}$$

$$\le \left\{ E_P \left[ \left\{ \frac{dQ}{dP}(\mathbf{x}) \right\}^{2 \cdot p} \right] \right\}^{1/(2 \cdot p)} \cdot \operatorname{diam}(\Omega) \cdot \overline{\lim_{N \to \infty}} \, \frac{N^{1/d}}{N^{2/d}}$$

$$= 0.$$

(139)

VI. From Theorem C.17, for $0 \le p \le d$,

$$\lim_{N \to \infty} N^{1/d} \cdot \left\{ E_{\hat{\mathbf{X}}_{P[N]}} \left[ E_P \left[ \left\{ \frac{dQ}{dP} \left( \mathbf{X}_{P[N]}^{(1)}(\mathbf{x}) \right) \right\}^p \cdot \left\| \mathbf{X}_{P[N]}^{(1)}(\mathbf{x}) - \mathbf{x} \right\|_\infty^p \right] \right] \right\}^{1/p}$$

$$\ge e^{-1} \cdot \left\{ E_P \left[ \left\{ \frac{dQ}{dP}(\mathbf{x}) \right\}^p \right] \right\}^{1/p},$$

(140)

where $E_{\hat{\mathbf{X}}_{P[N]}}[\cdot]$ denotes the expectation on each variable in $\hat{\mathbf{X}}_{P[N]} = \{\mathbf{X}_P^1, \mathbf{X}_P^2, \ldots, \mathbf{X}_P^N\}$.

VII. Let $\hat{\mathbf{X}}_{\mu[N]}$ denote the set of random variables defined in Proposition C.10. From Proposition C.10,

$$\phi \in \widetilde{\mathcal{F}}_{K\text{-}Lip}^{(N)} \iff \phi(\mathbf{X}_\mu^i) = \frac{dQ}{dP}(\mathbf{X}_\mu^i), \quad \text{for} \quad 1 \le \forall i \le N.$$

(141)

Now, we prove Equation (132). Let $\phi(\mathbf{x})$ be a member of $\widetilde{\mathcal{F}}_{K\text{-}Lip}^{(N)}$.

By applying the triangle inequality in the $L_p$ norm, we have

$$\left\{ E_P \left| \frac{dQ}{dP}(\mathbf{x}) - \phi(\mathbf{x}) \right|^p \right\}^{1/p}$$

$$\le \left\{ E_P \left| \frac{dQ}{dP}(\mathbf{x}) - \frac{dQ}{dP} \left( \mathbf{X}_{\mu[N]}^{(1)}(\mathbf{x}) \right) \right|^p \right\}^{1/p} + \left\{ E_P \left| \frac{dQ}{dP} \left( \mathbf{X}_{\mu[N]}^{(1)}(\mathbf{x}) \right) - \phi(\mathbf{x}) \right|^p \right\}^{1/p}.$$

(142)

From the $K$-Lipschitz continuity of $\phi$ and Equation (141),

$$\left\{ E_P \left| \frac{dQ}{dP} \left( \mathbf{X}_{\mu[N]}^{(1)}(\mathbf{x}) \right) - \phi(\mathbf{x}) \right|^p \right\}^{1/p} = \left\{ E_P \left| \phi \left( \mathbf{X}_{\mu[N]}^{(1)}(\mathbf{x}) \right) - \phi(\mathbf{x}) \right|^p \right\}^{1/p} \quad \text{(by Equation 141)}$$

$$\le K \cdot \left\{ E_P \left\| \mathbf{X}_{\mu[N]}^{(1)}(\mathbf{x}) - \mathbf{x} \right\|_\infty^p \right\}^{1/p}.$$

(143)

From Equations (137) and (143),

$$\varlimsup_{N\to\infty} \left\{ E_P \left| \frac{dQ}{dP}(\mathbf{X}^{(1)}_{\mu[N]}(\mathbf{x})) - \phi(\mathbf{x}) \right|^p \right\}^{1/p} \le K \cdot \mathrm{diam}(\Omega). \tag{144}$$

Next, by substituting $\mathbf{y} = \mathbf{X}^{(1)}_{\mu[N]}(\mathbf{x})$ and multiplying by $\frac{dP}{d\mu}(\mathbf{x})$ in Equation (136), and using the $L$-Lipschitz continuity of $T^*$, we have

$$\left\{ E_P \left| \frac{dQ}{dP}(\mathbf{X}^{(1)}_{\mu[N]}(\mathbf{x})) - \frac{dQ}{dP}(\mathbf{x}) \right|^p \right\}^{1/p}$$

$$= \left[ E_P \left| \frac{dQ}{dP}(\mathbf{x}) \times \left\{ \left( T^*(\mathbf{X}^{(1)}_{\mu[N]}(\mathbf{x})) - T^*(\mathbf{x}) \right) \right. \right. \right.$$

$$\left. \left. \left. + \frac{1}{2} \cdot e^{C_1(\mathbf{x})} \cdot \left( T^*(\mathbf{X}^{(1)}_{\mu[N]}(\mathbf{x})) - T^*(\mathbf{x}) \right)^2 \right\} \right|^p \right]^{1/p},$$

$$\text{where } 0 \le C_1(\mathbf{x}) \le \left| T^*\left(\mathbf{X}^{(1)}_{\mu[N]}(\mathbf{x})\right) - T^*(\mathbf{x}) \right|.$$

$$= \left\{ E_P \left| \frac{dQ}{dP}(\mathbf{x}) \times \left\{ \left( T^*(\mathbf{X}^{(1)}_{\mu[N]}(\mathbf{x})) - T^*(\mathbf{x}) \right) \right\} \right. \right.$$

$$\left. \left. + \frac{dQ}{dP}(\mathbf{x}) \times \left\{ \frac{1}{2} \cdot e^{C_1(\mathbf{x})} \cdot \left( T^*(\mathbf{X}^{(1)}_{\mu[N]}(\mathbf{x})) - T^*(\mathbf{x}) \right)^2 \right\} \right|^p \right\}^{1/p}$$

$$\le \left\{ E_P \left[ \left\{ \frac{dQ}{dP}(\mathbf{x}) \right\}^p \cdot \left| T^*(\mathbf{X}^{(1)}_{\mu[N]}(\mathbf{x})) - T^*(\mathbf{x}) \right|^p \right] \right\}^{1/p}$$

$$+ \left\{ E_P \left[ \left\{ \frac{dQ}{dP}(\mathbf{x}) \right\}^p \cdot \frac{1}{2^p} \cdot e^{p \cdot C_1(\mathbf{x})} \cdot \left| T^*(\mathbf{X}^{(1)}_{\mu[N]}(\mathbf{x})) - T^*(\mathbf{x}) \right|^{2 \cdot p} \right] \right\}^{1/p}$$

$$\le \left\{ E_P \left[ \left\{ \frac{dQ}{dP}(\mathbf{x}) \right\}^p \cdot \left| T^*(\mathbf{X}^{(1)}_{\mu[N]}(\mathbf{x})) - T^*(\mathbf{x}) \right|^p \right] \right\}^{1/p}$$

$$+ \left\{ E_P \left[ \left\{ \frac{dQ}{dP}(\mathbf{x}) \right\}^p \right. \right.$$

$$\left. \left. \times \frac{1}{2^p} \cdot e^{p \cdot \left| T^*(\mathbf{X}^{(1)}_{\mu[N]}(\mathbf{x})) - T^*(\mathbf{x}) \right|} \cdot \left| T^*(\mathbf{X}^{(1)}_{\mu[N]}(\mathbf{x})) - T^*(\mathbf{x}) \right|^{2 \cdot p} \right] \right\}^{1/p}$$

$$\left( \because C_1(\mathbf{x}) \le \left| T^*\left(\mathbf{X}^{(1)}_{\mu[N]}(\mathbf{x})\right) - T^*(\mathbf{x}) \right| \right)$$

$$\le \left\{ E_P \left[ \left\{ \frac{dQ}{dP}(\mathbf{x}) \right\}^p \cdot L^p \cdot \left\| \mathbf{X}^{(1)}_{\mu[N]}(\mathbf{x}) - \mathbf{x} \right\|_\infty^p \right] \right\}^{1/p}$$

$$+ \left\{ E_P \left[ \left\{ \frac{dQ}{dP}(\mathbf{x}) \right\}^p \right. \right.$$

$$\left. \left. \times \frac{1}{2^p} \cdot e^{p \cdot L \cdot \left\| \mathbf{X}^{(1)}_{\mu[N]}(\mathbf{x}) - \mathbf{x} \right\|_\infty} \cdot L^p \cdot \left\| \mathbf{X}^{(1)}_{\mu[N]}(\mathbf{x}) - \mathbf{x} \right\|_\infty^{2 \cdot p} \right] \right\}^{1/p}$$

$$\le \left\{ E_P \left[ \left\{ \frac{dQ}{dP}(\mathbf{x}) \right\}^p \cdot L^p \cdot \left\| \mathbf{X}^{(1)}_{\mu[N]}(\mathbf{x}) - \mathbf{x} \right\|_\infty^p \right] \right\}^{1/p}$$

$$+ \left\{ E_P \left[ \left\{ \frac{dQ}{dP}(\mathbf{x}) \right\}^p \right. \right.$$

$$\left. \left. \times \frac{1}{2^p} \cdot e^{p \cdot L \cdot \mathrm{diam}(\Omega)} \cdot L^p \cdot \left\| \mathbf{X}^{(1)}_{\mu[N]}(\mathbf{x}) - \mathbf{x} \right\|_\infty^{2 \cdot p} \right] \right\}^{1/p}$$

$$= L \cdot \left\{ E_P \left[ \left\{ \frac{dQ}{dP}(\mathbf{x}) \right\}^p \cdot \left\| \mathbf{X}^{(1)}_{\mu[N]}(\mathbf{x}) - \mathbf{x} \right\|_\infty^p \right] \right\}^{1/p}$$

$$+ \frac{1}{2} \cdot e^{L \cdot \operatorname{diam}(\Omega)} \cdot \left\{ E_P \left[ \left\{ \frac{dQ}{dP}(\mathbf{x}) \right\}^p \cdot \left\| \mathbf{X}_{\mu[N]}^{(1)}(\mathbf{x}) - \mathbf{x} \right\|_\infty^{2 \cdot p} \right] \right\}^{1/p} \tag{145}$$

From Equations (138), (139) and (145), we have

$$\varlimsup_{N \to \infty} N^{1/d} \cdot \left\{ E_P \left| \frac{dQ}{dP}(\mathbf{x}) - \phi\big(\mathbf{X}_{\mu[N]}^{(1)}(\mathbf{x})\big) \right|^p \right\}^{1/p}$$

$$\leq \varlimsup_{N \to \infty} N^{1/d} \cdot L \cdot \left\{ E_P \left[ \left\{ \frac{dQ}{dP}(\mathbf{x}) \right\}^p \cdot \left\| \mathbf{X}_{\mu[N]}^{(1)}(\mathbf{x}) - \mathbf{x} \right\|_\infty^p \right] \right\}^{1/p}$$

$$+ \varlimsup_{N \to \infty} N^{1/d} \cdot \frac{1}{2} \cdot e^{L \cdot \operatorname{diam}(\Omega)} \cdot \left\{ E_P \left[ \left\{ \frac{dQ}{dP}(\mathbf{x}) \right\}^p \cdot \left\| \mathbf{X}_{\mu[N]}^{(1)}(\mathbf{x}) - \mathbf{x} \right\|_\infty^{2 \cdot p} \right] \right\}^{1/p}$$

$$= L \cdot \operatorname{diam}(\Omega) \cdot \left\{ E_P \left[ \left\{ \frac{dQ}{dP}(\mathbf{x}) \right\}^{2 \cdot p} \right] \right\}^{1/(2 \cdot p)}. \tag{146}$$

Finally, from Equations (144), (142), and (146), we have

$$\lim_{N \to \infty} N^{1/d} \cdot \left\{ E_P \left| \frac{dQ}{dP}(\mathbf{x}) - \phi(\mathbf{x}) \right|^p \right\}^{1/p}$$

$$\leq L \cdot \operatorname{diam}(\Omega) \cdot \left\{ E_P \left[ \left\{ \frac{dQ}{dP}(\mathbf{x}) \right\}^{2 \cdot p} \right] \right\}^{1/(2 \cdot p)} + \operatorname{diam}(\Omega) \cdot K. \tag{147}$$

Thus, it is shown that Equation (132) holds.

Next, we prove Equation (133). By applying the triangle inequality in the $L_p$ norm, we have

$$\left\{ E_P \left| \frac{dQ}{dP}(\mathbf{x}) - \phi(\mathbf{x}) \right|^p \right\}^{1/p}$$

$$\geq \left\{ E_P \left| \frac{dQ}{dP}(\mathbf{x}) - \frac{dQ}{dP}\big(\mathbf{X}_{\mu[N]}^{(1)}(\mathbf{x})\big) \right|^p \right\}^{1/p} - \left\{ E_P \left| \frac{dQ}{dP}\big(\mathbf{X}_{\mu[N]}^{(1)}(\mathbf{x})\big) - \phi(\mathbf{x}) \right|^p \right\}^{1/p}. \tag{148}$$

By substituting $\mathbf{y} = \mathbf{X}_{\mu[N]}^{(1)}(\mathbf{x})$ and multiplying by $\frac{dP}{d\mu}(\mathbf{x})$ in Equation (136) along with the $L$-bi-Lipschitz continuity of $T^*$, we have

$$\left\{ E_P \left| \frac{dQ}{dP}\big(\mathbf{X}_{\mu[N]}^{(1)}(\mathbf{x})\big) - \frac{dQ}{dP}(\mathbf{x}) \right|^p \right\}^{1/p}$$

$$= \left\{ E_P \left| \frac{dQ}{dP}\big(\mathbf{X}_{\mu[N]}^{(1)}(\mathbf{x})\big) \right. \right.$$

$$\times \left\{ \Big( T^*\big(\mathbf{X}_{\mu[N]}^{(1)}(\mathbf{x})\big) - T^*(\mathbf{x}) \Big) \right.$$

$$\left. \left. + \frac{1}{2} \cdot e^{C_1(\mathbf{x})} \cdot \Big( T^*\big(\mathbf{X}_{\mu[N]}^{(1)}(\mathbf{x})\big) - T^*(\mathbf{x}) \Big)^2 \right\} \right|^p \right\}^{1/p}$$

$$\text{where } 0 \leq C_1(\mathbf{x}) \leq \left| T^*\big(\mathbf{X}_{\mu[N]}^{(1)}(\mathbf{x})\big) - T^*(\mathbf{x}) \right|$$

$$\geq \left\{ E_P \left[ \left\{ \frac{dQ}{dP}\big(\mathbf{X}_{\mu[N]}^{(1)}(\mathbf{x})\big) \right\}^p \cdot \left| T^*\big(\mathbf{X}_{\mu[N]}^{(1)}(\mathbf{x})\big) - T^*(\mathbf{x}) \right|^p \right] \right\}^{1/p}$$

$$- \left\{ E_P \left[ \left\{ \frac{dQ}{dP}\big(\mathbf{X}_{\mu[N]}^{(1)}(\mathbf{x})\big) \right\}^p \right. \right.$$

$$\times \frac{1}{2^p} \cdot e^{p \cdot \left| T^*(\mathbf{X}_{\mu[N]}^{(1)}(\mathbf{x})) - T^*(\mathbf{x}) \right|} \cdot \left| T^*(\mathbf{X}_{\mu[N]}^{(1)}(\mathbf{x})) - T^*(\mathbf{x}) \right|^{2 \cdot p} \Bigg] \Bigg\}^{1/p}$$

$$\geq \left\{ E_P \left[ \left\{ \frac{dQ}{dP}(\mathbf{X}_{\mu[N]}^{(1)}(\mathbf{x})) \right\}^p \cdot \frac{1}{L^p} \cdot \left\| \mathbf{X}_{\mu[N]}^{(1)}(\mathbf{x}) - \mathbf{x} \right\|_\infty^p \right] \right\}^{1/p}$$

$$- \left\{ E_P \left[ \left\{ \frac{dQ}{dP}(\mathbf{X}_{\mu[N]}^{(1)}(\mathbf{x})) \right\}^p \right. \right.$$

$$\left. \left. \times \frac{1}{2^p} \cdot e^{p \cdot L \cdot \left\| \mathbf{X}_{\mu[N]}^{(1)}(\mathbf{x}) - \mathbf{x} \right\|_\infty} \cdot L^p \cdot \left\| \mathbf{X}_{\mu[N]}^{(1)}(\mathbf{x}) - \mathbf{x} \right\|_\infty^{2 \cdot p} \right] \right\}^{1/p}$$

$$\geq \left\{ E_P \left[ \left\{ \frac{dQ}{dP}(\mathbf{X}_{\mu[N]}^{(1)}(\mathbf{x})) \right\}^p \cdot \frac{1}{L^p} \cdot \left\| \mathbf{X}_{\mu[N]}^{(1)}(\mathbf{x}) - \mathbf{x} \right\|_\infty^p \right] \right\}^{1/p}$$

$$- \left\{ E_P \left[ \left\{ \frac{dQ}{dP}(\mathbf{X}_{\mu[N]}^{(1)}(\mathbf{x})) \right\}^p \right. \right.$$

$$\left. \left. \times \frac{1}{2^p} \cdot e^{p \cdot L \cdot \mathrm{diam}(\Omega)} \cdot L^p \cdot \left\| \mathbf{X}_{\mu[N]}^{(1)}(\mathbf{x}) - \mathbf{x} \right\|_\infty^{2 \cdot p} \right] \right\}^{1/p}$$

$$= \frac{1}{L} \cdot \left\{ E_P \left[ \left\{ \frac{dQ}{dP}(\mathbf{X}_{\mu[N]}^{(1)}(\mathbf{x})) \right\}^p \cdot \left\| \mathbf{X}_{\mu[N]}^{(1)}(\mathbf{x}) - \mathbf{x} \right\|_\infty^p \right] \right\}^{1/p}$$

$$- \frac{1}{2} \cdot e^{\mathrm{diam}(\Omega)} \cdot L \cdot \left\{ E_P \left[ \left\{ \frac{dQ}{dP}(\mathbf{X}_{\mu[N]}^{(1)}(\mathbf{x})) \right\}^p \cdot \left\| \mathbf{X}_{\mu[N]}^{(1)}(\mathbf{x}) - \mathbf{x} \right\|_\infty^{2 \cdot p} \right] \right\}^{1/p} \tag{149}$$

From Equations (138), (139) and (149), we have

$$\varliminf_{N \to \infty} N^{1/d} \cdot \left\{ E_{\hat{\mathbf{X}}_{P[N]}} \left[ \left( E_P \left| \frac{dQ}{dP}(\mathbf{x}) - \phi(\mathbf{X}_{\mu[N]}^{(1)}(\mathbf{x})) \right|^p \right)^{1/p} \right] \right\}$$

$$\geq \varliminf_{N \to \infty} N^{1/d} \cdot \left\{ E_{\hat{\mathbf{X}}_{P[N]}} \left[ \frac{1}{L^p} \cdot \left( E_P \left[ \left\{ \frac{dQ}{dP}(\mathbf{X}_{\mu[N]}^{(1)}(\mathbf{x})) \right\}^p \cdot \left\| \mathbf{X}_{\mu[N]}^{(1)}(\mathbf{x}) - \mathbf{x} \right\|_\infty^p \right] \right)^{1/p} \right. \right.$$

$$\left. \left. - \frac{1}{2} \cdot e^{\mathrm{diam}(\Omega)} \cdot L \cdot \left( E_P \left[ \left\{ \frac{dQ}{dP}(\mathbf{X}_{\mu[N]}^{(1)}(\mathbf{x})) \right\}^p \cdot \left\| \mathbf{X}_{\mu[N]}^{(1)}(\mathbf{x}) - \mathbf{x} \right\|_\infty^{2 \cdot p} \right] \right)^{1/p} \right] \right\}$$

$$\geq \varliminf_{N \to \infty} N^{1/d} \cdot \left\{ E_{\hat{\mathbf{X}}_{P[N]}} \left[ \frac{1}{L} \cdot \left( E_P \left[ \left\{ \frac{dQ}{dP}(\mathbf{X}_{\mu[N]}^{(1)}(\mathbf{x})) \right\}^p \cdot \left\| \mathbf{X}_{\mu[N]}^{(1)}(\mathbf{x}) - \mathbf{x} \right\|_\infty^p \right] \right)^{1/p} \right] \right\}$$

$$- \varlimsup_{N \to \infty} N^{1/d} \cdot \left\{ E_{\hat{\mathbf{X}}_{P[N]}} \left[ \frac{1}{2} \cdot e^{\mathrm{diam}(\Omega)} \right. \right.$$

$$\left. \left. \times L \cdot \left( E_P \left[ \left\{ \frac{dQ}{dP}(\mathbf{X}_{\mu[N]}^{(1)}(\mathbf{x})) \right\}^p \cdot \left\| \mathbf{X}_{\mu[N]}^{(1)}(\mathbf{x}) - \mathbf{x} \right\|_\infty^{2 \cdot p} \right] \right)^{1/p} \right] \right\}$$

$$\geq \varliminf_{N \to \infty} N^{1/d} \cdot E_{\hat{\mathbf{X}}_{P[N]}} \left[ \frac{1}{L} \cdot \left( E_P \left[ \left\{ \frac{dQ}{dP}(\mathbf{X}_{\mu[N]}^{(1)}(\mathbf{x})) \right\}^p \cdot \left\| \mathbf{X}_{\mu[N]}^{(1)}(\mathbf{x}) - \mathbf{x} \right\|_\infty^p \right] \right)^{1/p} \right]$$

$$- E_{\hat{\mathbf{X}}_{P[N]}} \left[ \varlimsup_{N \to \infty} N^{1/d} \cdot \left\{ \frac{1}{2} \cdot e^{\mathrm{diam}(\Omega)} \right. \right.$$

$$\left. \left. \times L \cdot \left( E_P \left[ \left\{ \frac{dQ}{dP}(\mathbf{X}_{\mu[N]}^{(1)}(\mathbf{x})) \right\}^p \cdot \left\| \mathbf{X}_{\mu[N]}^{(1)}(\mathbf{x}) - \mathbf{x} \right\|_\infty^{2 \cdot p} \right] \right)^{1/p} \right\} \right]$$

$$= e^{-1} \cdot \frac{1}{L} \cdot \left\{ E_P \left[ \left\{ \frac{dQ}{dP}(\mathbf{x}) \right\}^p \right] \right\}^{1/p} . \tag{150}$$

Finally, from Equations (144), (148), and (150), we have

$$\lim_{N \to \infty} N^{p/d} \cdot E_{\hat{\mathbf{X}}_{P[N]}} \left[ \left\{ E_P \left| \frac{dQ}{dP}(\mathbf{x}) - \phi(\mathbf{x}) \right|^p \right\}^{1/p} \right]$$

$$\geq e^{-1} \cdot \frac{1}{L} \cdot \left\{ E_P \left[ \left\{ \frac{dQ}{dP}(\mathbf{x}) \right\}^p \right] \right\}^{1/p} - \text{diam}(\Omega) \cdot K. \tag{151}$$

Thus, it is shown that Equation (133) holds.

Next, we prove Equation (134).

First, we have

$$\left\{ E_P \left[ \left\{ \frac{dQ}{dP}(\mathbf{x}) \right\}^p \right] \right\}^{1/p} = \left\{ E_P \left[ \frac{dQ}{dP}(\mathbf{x}) \cdot \left\{ \frac{dQ}{dP}(\mathbf{x}) \right\}^{p-1} \right] \right\}^{1/p}$$

$$= \left\{ E_Q \left[ \left\{ \frac{dQ}{dP}(\mathbf{x}) \right\}^{p-1} \right] \right\}^{1/p}$$

$$= \left\{ E_Q \left[ e^{(p-1) \cdot \log \frac{dQ}{dP}(\mathbf{x})} \right] \right\}^{1/p}. \tag{152}$$

From Jensen's inequality,

$$\left\{ E_Q \left[ e^{(p-1) \cdot \log \frac{dQ}{dP}(\mathbf{x})} \right] \right\}^{1/p} \geq \left\{ e^{E_Q \left[ (p-1) \cdot \log \frac{dQ}{dP}(\mathbf{x}) \right]} \right\}^{1/p}$$

$$= \left\{ e^{(p-1) \cdot E_Q \left[ \log \frac{dQ}{dP}(\mathbf{x}) \right]} \right\}^{1/p}$$

$$= e^{\frac{p-1}{p} \cdot E_Q \left[ \log \frac{dQ}{dP}(\mathbf{x}) \right]}$$

$$= e^{\frac{p-1}{p} \cdot KL(Q||P)}. \tag{153}$$

From Equations (151), (152) and (153),

$$\lim_{N \to \infty} N^{p/d} \cdot E_{\hat{\mathbf{X}}_{P[N]}} \left[ \left\{ E_P \left| \frac{dQ}{dP}(\mathbf{x}) - \phi(\mathbf{x}) \right|^p \right\}^{1/p} \right]$$

$$\geq e^{-1} \cdot \frac{1}{L} \cdot \left\{ E_P \left[ \left\{ \frac{dQ}{dP}(\mathbf{x}) \right\}^p \right] \right\}^{1/p} - \text{diam}(\Omega) \cdot K$$

$$\geq \frac{1}{L} \cdot e^{\frac{p-1}{p} \cdot KL(Q||P) - 1} - \text{diam}(\Omega) \cdot K. \tag{154}$$

This completes the proof. $\square$

**Theorem C.21** (Theorem 4.8 restated). *Assume the same assumptions and notations as in Theorem C.20. Additionally, define*

$$\mathcal{F}_{K\text{-}Lip}^{(N)} = \left\{ \phi \in \mathcal{F}_{K\text{-}Lip} \;\middle|\; \exists \phi_* \in \widetilde{\mathcal{F}}_{K\text{-}Lip}^{(N)} \text{ such that } \phi = \phi_* + O_p \left( \frac{1}{\sqrt{N}} \right) \right\}. \tag{155}$$

*That is, $\mathcal{F}_{K\text{-}Lip}^{(N)}$ denotes the set of all functions that differ by at most $O_p(1/\sqrt{N})$ from some functions that minimize $\widetilde{\mathcal{L}}_f^{(N)}(\cdot)$.*

*Then, the same results as in Theorem C.20 hold for all $\phi \in \mathcal{F}_{K\text{-}Lip}^{(N)}$. Specifically:*

**(Upper Bound)** *Under Assumption C.4, Equation (132) holds for $1 \leq p \leq d/2$ such that for any $\phi \in \mathcal{F}_{K\text{-}Lip}^{(N)}$,*

$$\varlimsup_{N \to \infty} N^{1/d} \cdot \left\{ E_P \left| \frac{dQ}{dP}(\mathbf{x}) - \phi(\mathbf{x}) \right|^p \right\}^{1/p}$$

$$\leq L \cdot \mathrm{diam}(\Omega) \cdot \left\{ E_P \left[ \left\{ \frac{dQ}{dP}(\mathbf{x}) \right\}^{2 \cdot p} \right] \right\}^{1/(2 \cdot p)} + K \cdot \mathrm{diam}(\Omega). \tag{156}$$

**(Lower Bound)** *Under Assumption C.5, Equation (133) holds for any $\phi \in \mathcal{F}_{K\text{-}Lip}^{(N)}$, such that*

$$\lim_{N \to \infty} N^{1/d} \cdot E_{\hat{\mathbf{X}}_{P[N]}} \left[ \left\{ E_P \left| \frac{dQ}{dP}(\mathbf{x}) - \phi(\mathbf{x}) \right|^p \right\}^{1/p} \right]$$

$$\geq \frac{1}{L} \cdot \left\{ E_P \left[ \left\{ \frac{dQ}{dP}(\mathbf{x}) \right\}^p \right] \right\}^{1/p} - K \cdot \mathrm{diam}(\Omega) \tag{157}$$

$$\geq \frac{1}{L} \cdot e^{\frac{p-1}{p} \cdot KL(Q\|P) - 1} - K \cdot \mathrm{diam}(\Omega) \tag{158}$$

*Proof of Theorem C.21.* First, we prove Equation (156).

Let $\widetilde{\phi}$ be a member of $\mathcal{F}_{K\text{-}Lip}^{(N)}$. Then, there exists $\phi \in \mathcal{F}_{K\text{-}Lip}^{(N)}$ such that $\widetilde{\phi} = \phi + O_p(1/\sqrt{N})$.

Using the triangle inequality in the $L_p$ norm, we obtain

$$\left\{ E_P \left| \frac{dQ}{dP}(\mathbf{x}) - \widetilde{\phi}(\mathbf{x}) \right|^p \right\}^{1/p} = \left\{ E_P \left| \frac{dQ}{dP}(\mathbf{x}) - \phi(\mathbf{x}) + O_p \left( \frac{1}{\sqrt{N}} \right) \right|^p \right\}^{1/p}$$

$$\leq \left\{ E_P \left| \frac{dQ}{dP}(\mathbf{x}) - \phi(\mathbf{x}) \right|^p \right\}^{1/p} + \left\{ E_P \left| O_p \left( \frac{1}{\sqrt{N}} \right) \right|^p \right\}^{1/p}$$

$$= \left\{ E_P \left| \frac{dQ}{dP}(\mathbf{x}) - \phi(\mathbf{x}) \right|^p \right\}^{1/p} + O \left( \frac{1}{\sqrt{N}} \right). \tag{159}$$

From Equations (132) and (159), we have

$$\varlimsup_{N \to \infty} N^{1/d} \cdot \left\{ E_P \left| \frac{dQ}{dP}(\mathbf{x}) - \widetilde{\phi}(\mathbf{x}) \right|^p \right\}^{1/p}$$

$$\leq \varlimsup_{N \to \infty} N^{1/d} \cdot \left[ \left\{ E_P \left| \frac{dQ}{dP}(\mathbf{x}) - \phi(\mathbf{x}) \right|^p \right\}^{1/p} + O \left( \frac{1}{\sqrt{N}} \right) \right]$$

$$= \varlimsup_{N \to \infty} N^{1/d} \cdot \left\{ E_P \left| \frac{dQ}{dP}(\mathbf{x}) - \phi(\mathbf{x}) \right|^p \right\}^{1/p} + \varlimsup_{N \to \infty} N^{1/d} \cdot O \left( \frac{1}{\sqrt{N}} \right)$$

$$= \varlimsup_{N \to \infty} N^{1/d} \cdot \left\{ E_P \left| \frac{dQ}{dP}(\mathbf{x}) - \phi(\mathbf{x}) \right|^p \right\}^{1/p}$$

$$= L \cdot \mathrm{diam}(\Omega) \cdot \left\{ E_P \left[ \left\{ \frac{dQ}{dP}(\mathbf{x}) \right\}^{2 \cdot p} \right] \right\}^{1/(2 \cdot p)} + K \cdot \mathrm{diam}(\Omega). \tag{160}$$

Therefore, Equation (156) is proven.

Next, we prove Equation (157).

By applying the triangle inequality in the $L_p$ norm, we obtain

$$\left\{ E_P \left| \frac{dQ}{dP}(\mathbf{x}) - \widetilde{\phi}(\mathbf{x}) \right|^p \right\}^{1/p} = \left\{ E_P \left| \frac{dQ}{dP}(\mathbf{x}) - \phi(\mathbf{x}) + O_p \left( \frac{1}{\sqrt{N}} \right) \right|^p \right\}^{1/p}$$

$$\geq \left\{ E_P \left| \frac{dQ}{dP}(\mathbf{x}) - \phi(\mathbf{x}) \right|^p \right\}^{1/p} - \left\{ E_P \left| O_p \left( \frac{1}{\sqrt{N}} \right) \right|^p \right\}^{1/p}$$

$$= \left\{ E_P \left| \frac{dQ}{dP}(\mathbf{x}) - \phi(\mathbf{x}) \right|^p \right\}^{1/p} - O \left( \frac{1}{\sqrt{N}} \right). \tag{161}$$

In a similar manner to the derivation of Equation (160), we have

$$\lim_{N \to \infty} N^{1/d} \cdot E_{\hat{\mathbf{X}}_{P[N]}} \left[ \left\{ E_P \left| \frac{dQ}{dP}(\mathbf{x}) - \widetilde{\phi}(\mathbf{x}) \right|^p \right\}^{1/p} \right]$$

$$\geq \lim_{N \to \infty} N^{1/d} \cdot E_{\hat{\mathbf{X}}_{P[N]}} \left[ \left\{ E_P \left| \frac{dQ}{dP}(\mathbf{x}) - \phi(\mathbf{x}) \right|^p \right\}^{1/p} - O \left( \frac{1}{\sqrt{N}} \right) \right]$$

$$= \lim_{N \to \infty} N^{1/d} \cdot E_{\hat{\mathbf{X}}_{P[N]}} \left[ \left\{ E_P \left| \frac{dQ}{dP}(\mathbf{x}) - \phi(\mathbf{x}) \right|^p \right\}^{1/p} \right] - \varlimsup_{N \to \infty} N^{1/d} \cdot O \left( \frac{1}{\sqrt{N}} \right)$$

$$= \lim_{N \to \infty} N^{1/d} \cdot E_{\hat{\mathbf{X}}_{P[N]}} \left[ \left\{ E_P \left| \frac{dQ}{dP}(\mathbf{x}) - \phi(\mathbf{x}) \right|^p \right\}^{1/p} \right]$$

$$= \frac{1}{L} \cdot \left\{ E_P \left[ \left\{ \frac{dQ}{dP}(\mathbf{x}) \right\}^p \right] \right\}^{1/p} - K \cdot \mathrm{diam}(\Omega). \tag{162}$$

Therefore, Equation (157) is proven.

Equation (158) is obtained in the same manner as in the proof of Theorem C.20.

This completes the proof. $\qquad\square$

# D   DETAILS OF THE EXPERIMENTS IN SECTION 3

In this section, we provide detailed descriptions of the experiments reported in Section 3. Each dataset, experimental method, experimental result, and the neural network settings used in the experiments are presented in separate subsections.

## D.1   DATASETS.

We conducted two experiments: one for investigating the relationship between $L_p$ errors and KL-divergence in the data; the other for investigating the relationship between $L_p$ errors and the dimensionality of the data. In both, the datasets were generated from the following distributions: the numerator distribution was a multidimensional multimodal normal distribution, while the denominator distribution was a multidimensional standard normal distribution.

**Denominator Distribution:** The denominator datasets $\hat{\mathbf{X}}_{P[R]} = \{\mathbf{X}_P^1, \mathbf{X}_P^2, \ldots, \mathbf{X}_P^R\}$ were generated from the following $d$-dimensional standard normal distribution:

$$\mathbf{X}_P^i \overset{\mathrm{iid}}{\sim} \mathcal{N}(\mathbf{0}, I_d), \tag{163}$$

where $I_d$ denotes the $d$-dimensional identity matrix.

**Numerator Distribution:** The numerator datasets $\hat{\mathbf{X}}_{Q[S]} = \{\mathbf{X}_Q^1, \mathbf{X}_Q^2, \ldots, \mathbf{X}_Q^S\}$ were generated from the following $d$-dimensional, $M$-multimodal normal distribution:

$$\mathbf{X}_Q^i \overset{\mathrm{iid}}{\sim} \prod_{m=1}^{M} \mathcal{N}(\mu \cdot \mathbf{r}_m, I_d)^{Z_m}, \tag{164}$$

where for each mode $m$:

- $Z_m \sim \mathrm{Bernoulli}(1/M)$ and $\sum_{m=1}^{M} Z_m = 1$.
- $\mathbf{r}_m \sim \mathrm{Uniform}(\mathbb{S}^{d-1})$.

Here, Bernoulli$(1/M)$ denotes the Bernoulli distribution with parameter $1/M$, and Uniform$(\mathbb{S}^{d-1})$ denotes the uniform distribution on the $d$-dimensional unit surface $\mathbb{S}^{d-1} = \{\mathbf{x} \in \mathbb{R}^d : \|\mathbf{x}\| = 1\}$.

In the aforementioned setting when $M = 1$, the KL-divergence of the datasets is calculated as:

$$
\begin{aligned}
KL(P\|Q) &= E_P\left[\log\left(\frac{dP}{dQ}\right)\right] \\
&= E_{\mathcal{N}(\mathbf{0},I_d)}\left[\log\left(\frac{\mathcal{N}(\mathbf{0},I_d)}{\mathcal{N}(\mu \cdot \mathbf{r}_m, I_d)}\right)\right] \\
&= \frac{1}{2} \cdot \left[\log\frac{|\Sigma_p|}{|\Sigma_q|} - d + \mathrm{Tr}(\Sigma_p^{-1} \cdot \Sigma_q) + (\mu_p - \mu_q)^T \cdot \Sigma_p^{-1} \cdot (\mu_p - \mu_q)\right] \\
&= \frac{1}{2} \cdot \left[\log\frac{|I_d|}{|I_d|} - d + \mathrm{Tr}(I_d \cdot I_d) + (\mu \cdot \mathbf{r}_m)^T \cdot I_d \cdot (\mu \cdot \mathbf{r}_m)\right] \\
&= \frac{1}{2} \cdot \left(0 - d + d + \mu^2 \cdot \mathbf{r}_m^T \cdot \mathbf{r}_m\right) \\
&= \frac{1}{2} \cdot \mu^2.
\end{aligned}
\tag{165}
$$

From Equation (165), the KL-divergence of the datasets for $M > 1$ is calculated as:

$$
\begin{aligned}
KL(P\|Q) &= E_P\left[\log\left(\frac{dP}{dQ}\right)\right] \\
&= E_{\mathcal{N}(\mathbf{0},I_d)} E_{Z_m \sim \mathrm{Bernoulli}(1/M)}\left[\log\left(\frac{\mathcal{N}(\mathbf{0},I_d)}{\prod_{m=1}^{M}\mathcal{N}(\mu \cdot \mathbf{r}_m, I_d)^{Z_m}}\right)\right] \\
&= E_{\mathcal{N}(\mathbf{0},I_d)} E_{Z_m \sim \mathrm{Bernoulli}(1/M)}\left[\log\prod_{m=1}^{M}\left(\frac{\mathcal{N}(\mathbf{0},I_d)}{\mathcal{N}(\mu \cdot \mathbf{r}_m, I_d)}\right)^{Z_m}\right] \\
&= E_{\mathcal{N}(\mathbf{0},I_d)} E_{Z_m \sim \mathrm{Bernoulli}(1/M)}\left[\sum_{m=1}^{M}\log\left(\frac{\mathcal{N}(\mathbf{0},I_d)}{\mathcal{N}(\mu \cdot \mathbf{r}_m, I_d)}\right)\right] \\
&= E_{\mathcal{N}(\mathbf{0},I_d)}\left[\log\left(\frac{\mathcal{N}(\mathbf{0},I_d)}{\mathcal{N}(\mu \cdot \mathbf{r}_m, I_d)}\right)\right] \\
&= \frac{1}{2} \cdot \mu^2.
\end{aligned}
\tag{166}
$$

Thus, we set $\mu = \sqrt{2 \cdot KL(P\|Q)}$ in Equation (164) for $M = 1, 2, 3$, and 4, where $KL(P\|Q)$ denotes the KL-divergence of the datasets.

### D.2 EXPERIMENTAL PROCEDURE.

We trained neural networks using the training datasets by optimizing the KL-divergence or $\alpha$-divergence loss functions. Details of these two functions used in the experiments are provided below.

**KL-divergence loss function.** We used the following KL-divergence loss function, $\mathcal{L}_{\mathrm{KL}}(\cdot)$, in our experiments:

$$
\begin{aligned}
\mathcal{L}_{\mathrm{KL}}(T) &= \hat{E}_P\left[e^T\right] - \hat{E}_Q\left[T\right] \\
&= \frac{1}{S} \cdot \sum_{i=1}^{S} e^{T(\mathbf{X}_Q^i)} - \frac{1}{R} \cdot \sum_{i=1}^{R} T(\mathbf{X}_P^i).
\end{aligned}
\tag{167}
$$

$\alpha$**-divergence loss function.** We utilize an $\alpha$-divergence loss function originally proposed in our previous work (Kitazawa, 2024). The $\alpha$-divergence loss function is defined as:

The $\alpha$-divergence loss function is defined as:

$$
\begin{aligned}
\mathcal{L}_{\alpha\text{-divergence}}^{(R,S)}(T\,;\,\alpha) &= \frac{1}{\alpha} \cdot \hat{E}_{Q[S]}\left[e^{\alpha \cdot T_\theta}\right] + \frac{1}{1-\alpha} \cdot \hat{E}_{P[R]}\left[e^{(\alpha-1) \cdot T_\theta}\right] \\
&= \frac{1}{\alpha} \cdot \frac{1}{S} \cdot \sum_{i=1}^{S} e^{\alpha \cdot T(\mathbf{X}_Q^i)} + \frac{1}{1-\alpha} \cdot \frac{1}{R} \cdot \sum_{i=1}^{R} e^{(\alpha-1) \cdot T(\mathbf{X}_P^i)}.
\end{aligned} \tag{168}
$$

For further details and theoretical derivations of this loss function, we refer the reader to Kitazawa (2024).

$L_p$ **Errors vs. KL-Divergence in Data.** We initially created 100 training, validation, and test datasets, each consisting of 10000 samples, with a data dimensionality of 5 and KL-divergence values of 1, 2, 4, 8, 10, 12, and 14. The numerator datasets were generated with modalities of 1, 2, 3, and 4 using the aforementioned distributions. Neural networks were trained using the training datasets by optimizing both the $\alpha$-divergence and KL-divergence loss functions. Training was halted if the validation loss did not improve for an entire epoch. After training, the $L_p$ errors of the estimated density ratios for $p = 1$, 2, and 3 were measured using the test datasets. A total of 100 trials were conducted, and we reported the median $L_p$ errors along with the interquartile range (25th to 75th percentiles) for each KL-divergence and $\alpha$-divergence function.

$L_p$ **Errors vs. the Dimensions of Data.** We initially created 100 training datasets, each consisting of 20000 samples, and 100 validation and test datasets, each consisting of 5000 samples, with data dimensionalities of 50, 100, and 200, and a KL-divergence value of 3. Neural networks were trained using training datasets of sizes 1000, 2000, 4000, 8000, and 16000, by optimizing both the $\alpha$-divergence and KL-divergence loss functions. The numerator datasets were generated from the aforementioned distributions, with modalities $M = 1$, 2, 3, and 4. Training was halted if the validation loss did not improve for an entire epoch. After training, the $L_p$ errors of the estimated density ratios for $p = 1$, 2, and 3 were measured using the test datasets. A total of 100 trials were conducted, and we reported the median $L_p$ errors along with the interquartile range (25th to 75th percentiles) for each KL-divergence and $\alpha$-divergence function.

## D.3 Results.

$L_p$ **Errors vs. the KL-Divergence in Data.** The results for each multimodal case $M = 1, 2, 3$, and 4 of the numerator datasets are shown in Figure 3. The results for $M = 1$ were reported in Section 3.

As shown in Figure 3, the estimation errors for $p > 1$ increased significantly, with the rate of increase accelerating as $p$ becomes larger. In contrast, for $p = 1$, a relatively mild increase was observed. As indicated by Theorem 3.5, these results emphasize the impact of the KL-divergence in the data on $L_p$ errors for $p > 1$ in DRE with $f$-divergence loss functions. Additionally, only small differences were observed in the results among the modalities of the numerator datasets.

$L_p$ **Errors vs. the Dimensions of Data.** The results for each multimodal case $M = 1, 2, 3$, and 4 of the numerator datasets are shown in Figure 4 and 5. The results of $M = 1$ (the first and second rows in Figure 4) were reported in Section 3.

As shown in Figure 2, the $L_1$, $L_2$, and $L_3$ errors in DRE worsened as the data dimensionality increased for both the $\alpha$-divergence and KL-divergence loss functions. These results indicate that the curse of dimensionality affects all $L_p$ errors equally, as suggested by Theorem 3.5. Additionally, little difference was observed in the results across the modalities of the numerator datasets.

## D.4 Neural Network Architecture, Optimization Algorithm, and Hyperparameters.

$L_p$ **Errors vs. the KL-Divergence in Data.** The same neural network architecture, optimization algorithm, and hyperparameters were used for both the KL-divergence and $\alpha$-divergence loss functions. A 6-layer perceptron with ReLU activation was employed, with each hidden layer consisting of 1024 nodes. For optimization with both the KL-divergence and $\alpha$-divergence loss functions, the learning rate was set to 0.0001, and the batch size was 128. Early stopping was applied with a patience of 3

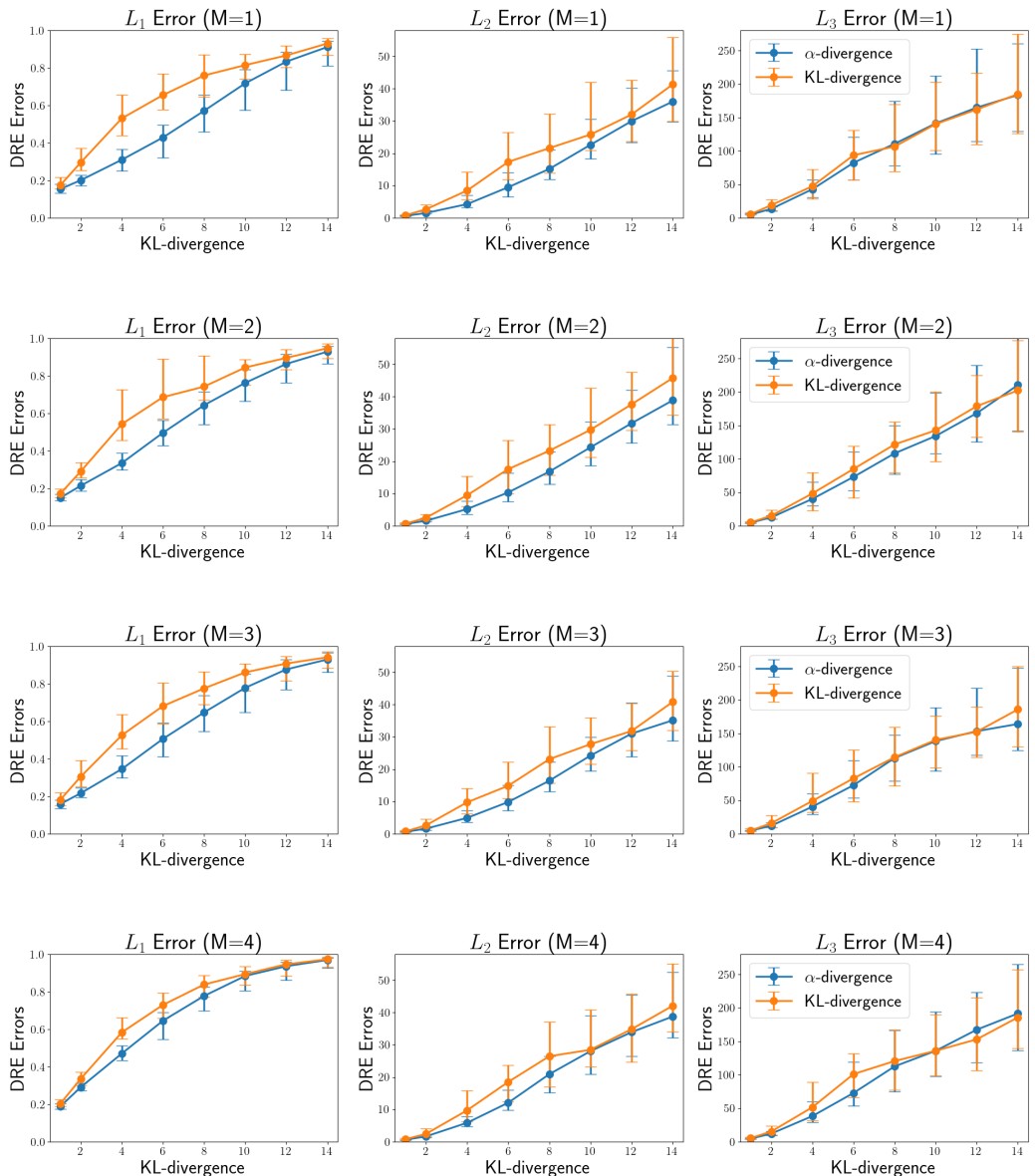

Figure 3: The experimental results of $L_p$ errors versus the KL-divergence in the data for each multimodal case $M = 1, 2, 3$, and 4 of the numerator datasets are presented, as discussed in Sections 3 and D. The results for $M = 1$ were reported in Section 3. The $x$-axis represents the KL-divergence of synthetic datasets with fixed dimensions. The $y$-axes of the left, center, and right graphs represent the $L_1$, $L_2$, and $L_3$ errors in DRE, respectively. The blue line represents errors using the $\alpha$-divergence loss function, and the orange line represents errors using the KL-divergence loss function. The error bars denote the interquartile range (25th to 75th percentiles) of the $y$-axis values. The plots show the median $y$-axis values corresponding to the KL-divergence levels in the synthetic datasets.

epochs, and the maximum number of epochs was set to 5000. The value of $\alpha$ for the $\alpha$-divergence loss function was set to 0.5, PyTorch (Paszke et al., 2017) library in Python was used to implement all models for DRE, with the Adam optimizer (Kingma, 2014) in PyTorch and an NVIDIA T4 GPU used for training the neural networks.

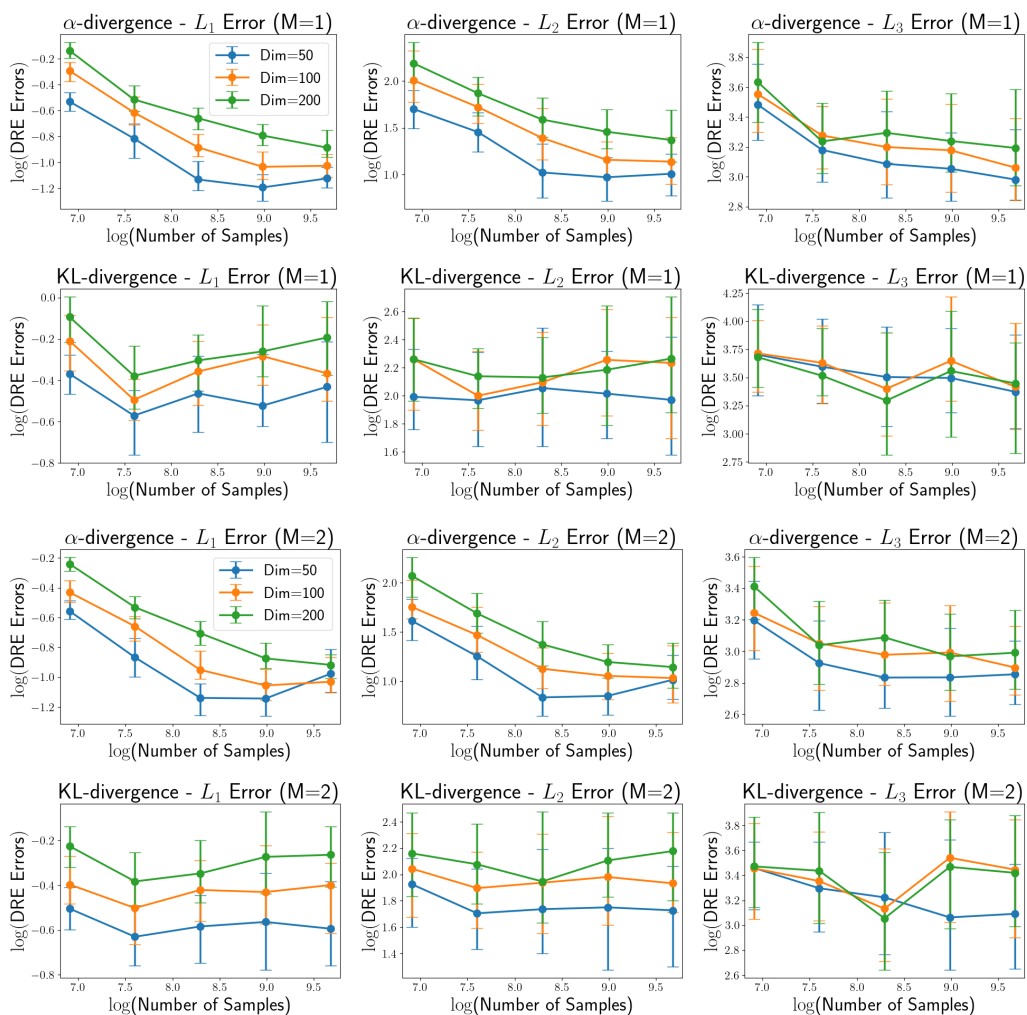

Figure 4: The experimental results of $L_p$ errors versus the dimensionality of the data for the multimodal cases $M = 1$ and 2 in the numerator datasets are presented, as discussed in Sections 3 and D. The results for $M = 1$ were reported in Section 3. The top row shows the results using the $\alpha$-divergence loss function, while the bottom row shows the results using the KL-divergence loss function. The $x$-axis represents the logarithm of the number of samples used for the optimizations for DRE. The $y$-axes of the left, center, and right graphs represent the $L_1$, $L_2$, and $L_3$ errors in DRE, respectively. The blue, orange, and green lines represent the results for data dimensionalities of 50, 100, and 200, respectively. The plots show the median $y$-axis values, and the error bars indicate the interquartile range (25th to 75th percentiles) of the $y$-axis values for the logarithm of the number of samples used in the optimizations for DRE.

$L_p$ **Errors vs. the Dimensions of Data.** The same neural network architecture, optimization algorithm, and hyperparameters were used for both the KL-divergence and $\alpha$-divergence loss functions. A 6-layer perceptron with ReLU activation was employed, with each hidden layer consisting of 1024 nodes. For optimization with both the KL-divergence and $\alpha$-divergence loss functions, the learning rate was set to 0.0001, and the batch size was 128. Early stopping was applied with a patience of 1 epoch, and the maximum number of epochs was set to 5000. The value of $\alpha$ for the $\alpha$-divergence loss function was set to 0.5, PyTorch (Paszke et al., 2017) library in Python was used to implement all models for DRE, with the Adam optimizer (Kingma, 2014) in PyTorch and an NVIDIA T4 GPU used for training the neural networks.

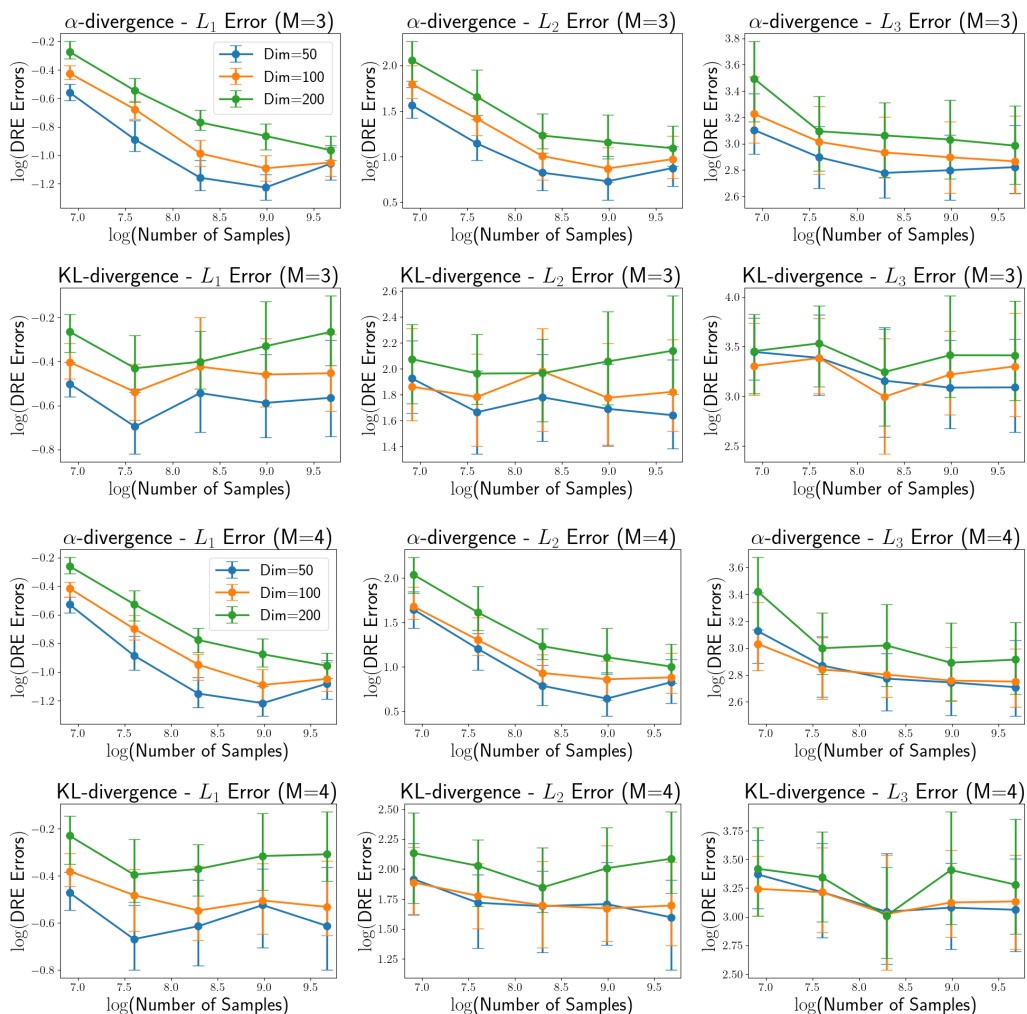

Figure 5: The experimental results of $L_p$ errors versus the dimensionality of the data for the multi-modal case $M = 3$ and 4 in the numerator datasets are presented, as discussed in Sections D. The top row shows the results using the $\alpha$-divergence loss function, while the bottom row shows the results using the KL-divergence loss function. The $x$-axis represents the logarithm of the number of samples used for the optimizations for DRE. The $y$-axes of the left, center, and right graphs represent the $L_1$, $L_2$, and $L_3$ errors in DRE, respectively. Blue, orange, and green lines represent the results for data dimensionalities of 50, 100, and 200, respectively. The plots show the median $y$-axis values, and the error bars indicate the interquartile range (25th to 75th percentiles) of the $y$-axis values for the logarithm of the number of samples used in the optimizations for DRE.

# E  FURTHER DISCUSSIONS RELATED TO THIS STUDY

In this section, we delve into further discussions related to this study. First, we compare the upper DRE bounds derived in this study with those reported in previous research. Next, we provide remarks on Assumption 3.3, highlighting its differences and similarities with related assumptions in prior work. Finally, we discuss the potential applications suggested by the findings of this study.

## E.1  COMPARISON WITH EXISTING DRE BOUNDS

We now compare our $L_p$ upper bound, as presented in Equation (4) of Theorem 3.5, to known DRE bounds from other methods.

The terms related to data dimensionality in our upper bound are tighter than the existing non-parametric minimax upper bounds in DRE. Furthermore, no prior work has included a term comparable to ours that involves the exponential of the KL-divergence, as shown in Equation (6) of Theorem 3.5.

Nguyen et al. (2010) presented a minimax upper bound rate of $O(1/N^{\frac{1}{2+d}})$ for the Hellinger distance between the true and estimated density ratio, obtained by optimizing a KL-divergence loss function. Since the Hellinger distance serves as an upper bound for the total variation distance (Sason & Verdú, 2016), the result from Nguyen et al. (2010) provides an upper bound on the $L_1$ error in DRE using the KL-divergence loss function. Kanamori et al. (2012) proposed an upper bound of $O(1/N^{\frac{1}{2+d}})$ for DRE using kernel unconstrained least-squares importance fitting (KuLSIF), their proposed DRE method. Under an assumption on the $\beta$-Hölder continuity of the probability ratio function, Kpotufe (2017) presented an upper bound of $O_P(\log N/N^{\frac{\beta}{\beta+d}})$ for DRE using an empirical distribution-based estimator, where our case corresponds to $\beta = 1$. A recent study (Lin et al., 2023) provided $L_1$ and $L_2$ error upper bounds of $O(1/N^{\frac{1}{2+d}})$ in DRE for an estimator using the $M$-th nearest neighbor, as $M$ increases along with the sample size.

In terms of comparison with our $L_p$ lower bound, a minimax $L_1$ lower bound of $O(1/N^{\frac{1}{2+d}})$, for example, was provided by Lin et al. (2023). This lower bound is larger than our lower bound in Equation (5) in Theorem 3.5 and appears tighter than ours. However, minimax lower bounds may not represent the true lower bounds and cannot be directly compared to our lower bound, as discussed in Section 1.

## E.2  REMARKS ON ASSUMPTION 3.3 AND RELATED ASSUMPTIONS IN PRIOR WORK

In the following, we provide remarks on Assumption 3.3 by comparing it with related assumptions in prior work.

An assumption closely related to Assumption 3.3 can be found in the pseudo-self-concordance property of losses introduced by Bach (2010). While the pseudo-self-concordance assumption guarantees that the original loss function is smooth and strongly convex proportional to its second derivative, Assumption 3.3 ensures these properties only for the expectation of the loss function.

First, we briefly review the pseudo-self-concordance assumption and a key property of loss functions that follows from it. Bach (2010) introduced the following pseudo-self-concordance assumption. [2]

**Assumption E.1** (Pseudo self-concordance). For any $u > 0$ and for any $r \in \mathbb{R}$, the loss $g(u)$ satisfies $|g'''(u + r)| \leq R \cdot r^2 \cdot g''(u)$, for some $R > 0$.

According to Proposition 1 in Bach (2010), under Assumption E.1, we have, for a sufficiently small $r_0 > 0$,

$$e^{-R \cdot r^2} \leq \frac{g''(u + r)}{g''(u)} \leq e^{R \cdot r^2}, \text{ for } 0 < r < r_0. \tag{169}$$

---

[2]In our discussion, we consider the pseudo-self-concordance assumption only for loss functions defined on a one-dimensional variable, whereas Bach (2010) introduced it for loss functions in a multidimensional domain. For a precise formulation, please refer to Propositions 1 and 2 in Bach (2010).

Now, let $G_u(r) = \{g(u+r) - g(u)\}/g''(u)$. From Equation (169),

$$\frac{1}{L} \le G''_u(r) \le L, \quad \text{for } 0 < r < r_0, \tag{170}$$

where $L = e^{R \cdot r_0^2}$.

Therefore, the pseudo-self-concordance property implies that $G_u(r)$ is both $L$-smooth and $1/L$-strongly convex on any interval of fixed length $r_0$, with $L$ independent of $u$. This property is considered a key characteristic of loss functions under the pseudo-self-concordance assumption.

Next, we discuss the properties of the loss function derived from our assumptions. Theorem C.9 in the appendix characterizes the local convexity of the loss function as follows:

$$\widetilde{l}_f\left(\frac{dQ}{dP}(\mathbf{x}) + r; \mathbf{x}\right) - \widetilde{l}_f\left(\frac{dQ}{dP}(\mathbf{x}); \mathbf{x}\right) = \frac{1}{2} \cdot f''\left(\frac{dQ}{dP}(\mathbf{x})\right) \cdot \frac{dP}{d\mu}(\mathbf{x}) \cdot r^2 + o\left(r^2\right). \tag{171}$$

Additionally, from Lemma C.8,

$$\widetilde{l}''_f\left(\frac{dQ}{dP}(\mathbf{x}); \mathbf{x}\right) = f''\left(\frac{dQ}{dP}(\mathbf{x})\right) \cdot \frac{dP}{d\mu}(\mathbf{x}), \tag{172}$$

where

$$\widetilde{l}''_f(u; \mathbf{x}) = \frac{d^2}{dr^2} \widetilde{l}_f(u + r; \mathbf{x})\Big|_{r=0}.$$

From Equations (171) and (172), as $r \to 0$,

$$\frac{\widetilde{l}_f\left(\frac{dQ}{dP}(\mathbf{x}) + r; \mathbf{x}\right) - \widetilde{l}_f\left(\frac{dQ}{dP}(\mathbf{x}); \mathbf{x}\right)}{\widetilde{l}''_f\left(\frac{dQ}{dP}(\mathbf{x}); \mathbf{x}\right)} = \frac{r^2}{2} + o_{\mathbf{x}}(1), \tag{173}$$

where $o_{\mathbf{x}}(1)$ denotes a quantity that converges to 0 as $r \to 0$, though not uniformly in $\mathbf{x}$; that is, $f(r) = o_{\mathbf{x}}(1)$ if and only if, for every $\varepsilon > 0$, there exists $\delta_{\mathbf{x}} > 0$ (depending on $\mathbf{x}$) such that $|f(r)| < \varepsilon$ for all $0 < r < \delta_{\mathbf{x}}$.

Now, let $G_{u(\mathbf{x})}(r) = \left\{\widetilde{l}_f(u(\mathbf{x}) + r; \mathbf{x}) - \widetilde{l}_f(u(\mathbf{x}); \mathbf{x})\right\} / \widetilde{l}''_f(u(\mathbf{x}); \mathbf{x})$, where $u(\mathbf{x}) = dQ/dP(\mathbf{x})$. From Equation (173), we have, for some $\delta_{\mathbf{x}} > 0$ and $L_{\mathbf{x}} \ge 1$,

$$\frac{1}{L_{\mathbf{x}}} \le G''_{u(\mathbf{x})}(r) \le L_{\mathbf{x}}, \quad \text{for } 0 < r < \delta_{\mathbf{x}}, \tag{174}$$

where $\delta_{\mathbf{x}} > 0$ and $L_{\mathbf{x}} \ge 1$ are determined at each point $\mathbf{x} \in \Omega$. Because $\delta_{\mathbf{x}}$ and $L_{\mathbf{x}}$ depend on $\mathbf{x}$, Equation (174) does not imply that $G_{u(\mathbf{x})}$ is $L$-smooth or $1/L$-strongly convex on any interval of a fixed length.

However, taking the expectation with respect to $\mu$ on both sides of Equation (171) yields

$$E_\mu\left[\widetilde{l}_f\left(\frac{dQ}{dP}(\mathbf{x}) + r; \mathbf{x}\right)\right] - E_\mu\left[\widetilde{l}_f\left(\frac{dQ}{dP}(\mathbf{x}); \mathbf{x}\right)\right] = \frac{1}{2} \cdot E_P\left[f''\left(\frac{dQ}{dP}\right)\right] \cdot r^2 + o\left(r^2\right). \tag{175}$$

From Equation (175), we have

$$\frac{d^2}{dr^2}\left\{E_\mu\left[\widetilde{l}_f\left(\frac{dQ}{dP}(\mathbf{x}) + r; \mathbf{x}\right)\right]\right\}\Big|_{r=0} = E_P\left[f''\left(\frac{dQ}{dP}\right)\right]. \tag{176}$$

Thus,

$$\overline{G}(r) = \frac{r^2}{2} + \frac{o\left(r^2\right)}{E_P\left[f''\left(\frac{dQ}{dP}\right)\right]}, \tag{177}$$

where

$$\overline{G}(r) = \frac{E_\mu\left[\widetilde{l}_f\left(\frac{dQ}{dP}(\mathbf{x}) + r; \mathbf{x}\right)\right] - E_\mu\left[\widetilde{l}_f\left(\frac{dQ}{dP}(\mathbf{x}); \mathbf{x}\right)\right]}{\frac{d^2}{dr^2}\left\{E_\mu\left[\widetilde{l}_f\left(\frac{dQ}{dP}(\mathbf{x}) + r; \mathbf{x}\right)\right]\right\}\Big|_{r=0}}.$$

From Equation (177), we deduce that, for some $L > 1$ and $r_0 > 0$,

$$\frac{1}{L} \leq \overline{G}''(r) \leq L, \ \text{ for } \ 0 < r < r_0. \tag{178}$$

Equation (178) implies that the expectation of the loss function is locally both smooth and strongly convex, with magnitudes proportional to its second derivative. In contrast, under the pseudo-self-concordance assumption, the original loss function is guaranteed to possess these properties (see Equation (170)).

In summary, under Assumption 3.3, the expectation of the loss function exhibits the same local smoothness and strong convexity properties (proportional to its second derivative) as those guaranteed by the pseudo-self-concordance assumption.

Furthermore, we note that the expression $E_P[f''(dQ/dP)]$ in Assumption 3.3 resembles the Fisher information when $f(u) = -\log u$, as shown in Equations (175) and (176). Thus, as an alternative perspective, we propose that Assumption 3.3 establishes an information-theoretic bound for estimation using $f$-divergence optimization.

### E.3  APPLICATIONS OF THIS STUDY

In this section, we provide a brief discussion of potential applications highlighted by our findings. The following two key applications can be derived from our results.

**Selecting a benchmark index for evaluating DRE methods.**  When evaluating the accuracy of DRE methods using synthetic datasets, the root mean squared error (RMSE) or mean squared error (MSE) is recommended over the mean absolute error (MAE). Prior works did not carefully consider the differences in their behavior with respect to the KL divergence of the datasets. For example, Kimura & Bondell (2024) used MAE, whereas Kato & Teshima (2021) used MSE.

**Fitting the distribution of base noise for $f$-GAN and Normalizing Flow.**  The optimization of $f$-GANs (Nowozin et al., 2016) could benefit from adjusting the base noise distribution to better match the data. Since the optimization of $f$-GANs is equivalent to DRE by optimizing the $f$-divergence (Uehara et al., 2016), the accuracy of generative models could be improved by fitting the base parametric models to the data in terms of KL divergence minimization (i.e., likelihood maximization). A similar approach could also be applied to the base models in Normalizing Flow (Papamakarios et al., 2021).

Table 2: List of $f'(\phi)$ and $f^*(f'(\phi))$ in Equation (1) together with convex functions, as discussed in Section 2.2. Part of the list of divergences and their convex functions is based on Nowozin et al. (2016).

| Name | convex function $f$ | $f'(\phi)$ | $f^*(f'(\phi))$ |
|------|---------------------|------------|------------------|
| KL | $u \cdot \log u$ | $\log(\phi) + 1$ | $\phi$ |
| Pearson $\chi^2$ | $(u-1)^2$ | $2 \cdot \phi - 2$ | $\phi^2 - 1$ |
| Squared Hellinger | $(\sqrt{u} - 1)^2$ | $1 - \phi^{-1/2}$ | $\phi^{1/2} - 1$ |
| GAN | $u \cdot \log u - (u+1) \cdot \log(u+1)$ | $-\log(1 + \phi^{-1})$ | $\log(1 + \phi)$ |

