# OpenReview forum: "Bounds on $L_p$ Errors in Density Ratio Estimation via $f$-Divergence Loss Functions"
_ICLR.cc/2025/Conference — ICLR 2025 Poster_

### Official Review · Reviewer_eBZV · 2024-11-02

**Soundness:** 3
**Presentation:** 2
**Contribution:** 2
**Rating:** 5
**Confidence:** 2

**Summary:**

This article presents an analysis of error rates for density ratio estimation (DRE) in the context of $L^p$-losses applied to $f$-divergence minimization. The study derives upper and lower error bounds, noting that these rates increase with \(p\) and exhibit the curse of dimensionality. Theoretical findings are supported by numerical illustrations, which help validate the proposed rates.

**Strengths:**

Problem is interesting. The proofs seem to be correct. Numerical illustrations support the theoretical findings.

**Weaknesses:**

While the work appears mathematically sound and introduces some interesting proof techniques, I am not inclined to recommend acceptance at this stage, primarily due to the following points:

- **Lack of Comparison with Recent DRE Literature**
    The paper would benefit from a more thorough comparison with recent work in DRE estimation, particularly research involving divergence-based losses. It is, at the current stage, not possible for me to set the results into the context of the large field of DRE (see questions).
- **Context for Techniques in Sections 4.1 and 4.2**
    The conceptual reformulation presented in Section 4.1 and the nearest-neighbor-based approach in Section 4.2 would benefit from contextualization. Comparing these methods with existing approaches would clarify their originality and contribution to the field.
- **Practical Applicability of Assumptions**
    The main results rely heavily on the Lipschitz continuity of both the function class used and the energy function of the distributions. However, ensuring these assumptions in practical settings may be challenging. This reliance makes the interpretability of the bounds somewhat limited and may constrain practical applicability.
- **Interpretability of Practical Implications**
    It remains unclear what practical conclusions can be drawn from the theoretical results. For example, are there implications for algorithmic design that might mitigate the curse of dimensionality or handle large sample sizes effectively? The implications of error rate growth with respect to $p$ should also be discussed.
- **Minor Issues and Clarifications**
    The parameter $N$ in Theorem 3.5 is undefined. It should be clarified in what sense equation (4) holds---is this with high probability?

**Questions:**

+ Are conceptual reformulation presented in Section 4.1 and the nearest-neighbor-based approach in Section 4.2 widely used or newly adapted in the paper?
+ How does Assumption 3.3 relate to the pseudo-self-concordance of losses as discussed in [1]?
+ Could the proof techniques introduced here be compared to those used in [1, 2]? Is the presented approach more general than these prior works?
 + Can the derived rates with respect to sample size be directly compared to those established in the literature?

[1] Zellinger, W., Kindermann, S., and Pereverzyev, S. V. "Adaptive Learning of Density Ratios in RKHS." Journal of Machine Learning Research, 24:1–28, 2023.

[2] Menon, A. and Ong, C. S. "Linking Losses for Density Ratio and Class-Probability Estimation." International Conference on Machine Learning, pp. 304–313, PMLR, 2016.

----------------------------------------------------------------------------------------------------------
**after the rebuttal, I raised my score to 5**

---

> ### Author Response · Authors · 2024-11-21
>
> We sincerely thank you for your constructive feedback and for highlighting important points of concern. Your comments have been invaluable in helping us identify areas for improvement, and we have carefully considered each of your points. Below, we provide detailed responses to address your concerns.
>
> > Are conceptual reformulation presented in Section 4.1 and the nearest-neighbor-based approach in Section 4.2 widely used or newly adapted in the paper?
>
> To the best of our knowledge, we are the first to propose this specific reformulation and to develop the corresponding proving approach, which had not been addressed in previous studies.
>
> > How does Assumption 3.3 relate to the pseudo-self-concordance of losses as discussed in [1]?
>
> We believe that both Assumption 3.3 and the pseudo-self-concordance of losses discussed in [1] assume some degree of the convexity of the loss functions, but they characterize these degrees differently.
>
> Specifically, according to Proposition 15 in [A9], the pseudo-self-concordance of the losses as discussed in [1] leads to the $L$-smoothness of the loss function (i.e., Lipschitz continuity of the gradient of the loss function), which is a deterministic property reflecting the strength of the loss convexity.
>
> On the other hand, from Theorem C.9 in the appendix, Assumption 3.3 provides a local property of the strength of the loss convexity such that
>
> $$
> \frac{\widetilde{l}\_{f} \left(u;\mathbf{x}\right) -  \widetilde{l}_{f} \left(\frac{dQ}{dP}(\mathbf{x}) ;\mathbf{x}\right)}{\left| u - \frac{dQ}{dP}(\mathbf{x})  \right|^2 }
> = \frac{1}{2} \cdot f''\left(\frac{dQ}{dP}(\mathbf{x})\right) \cdot \frac{dP}{d\mu}(\mathbf{x}) + o\left(\left| u - \frac{dQ}{dP}(\mathbf{x})  \right|^2 \right).
> $$
>
> Considering the expectation with respect to $\mu$ on both sides of this equation, we have
>
> $$
> E_{\mu} \left[ \frac{\widetilde{l}\_{f} \left(u;\mathbf{x}\right) -  \widetilde{l}\_{f} \left(\frac{dQ}{dP}(\mathbf{x}) ;\mathbf{x}\right)}{\left| u - \frac{dQ}{dP}(\mathbf{x})  \right|^2 } \right]
> = \frac{1}{2} \cdot E_P \left[  f''\left(\frac{dQ}{dP}(\mathbf{x})\right) \right] + o\left(E_{\mu} \left| u - \frac{dQ}{dP}(\mathbf{x})  \right|^2 \right),
> $$
>
> which implies that the strength of the loss convexity is statistically $E_{P}[f''(dQ/dP)]$.
>
> ($E_P[f''(dQ/dP)]$ can also be interpreted as an $f$-divergence information, such as the Fisher information when $f(u) = - \log u$.)
>
>
> Initially, we considered it necessary to impose conditions on the $L$-smoothness and $\mu$-strong convexity of the loss function, similarly to the assumption such as the pseudo-self-concordance of losses discussed in [1]. However, we found that only the strength of the convexity of the expected loss function is sufficient, and therefore our assumptions does not include such assumptions.
>
> [A9] Marteau-Ferey, U., Ostrovskii, D., Bach, F., & Rudi, A. (2019). Beyond Least-Squares: Fast Rates for Regularized Empirical Risk Minimization through Self-Concordance. Proceedings of Machine Learning Research vol, 99, 1-47.

---

> > ### Author Response · Authors · 2024-11-21
> >
> > > Could the proof techniques introduced here be compared to those used in [1, 2]? Is the presented approach more general than these prior works?
> >
> > We believe that the generality of two proving techniques can be compared only when results from one are derived from those of the other.  Thus, we consider it necessary to first examine whether the results in [1, 2] can be obtained from our findings.
> >
> > We believe that [1] presented the convergence rate of divergence estimation with a regularization term, which cannot be directly derived from our results.
> > Therefore, a comparison of the proof methods may not be meaningful.
> >
> > However, to address this issue more accurately, we believe it is important to further consider the following two points:
> >   1. Can a conceptual loss function generated from a convex function  $f$, as described in Equation (7) of our study, with a regularization term such as  $  \lambda  \cdot \| u^2 \| / 2$, be derived from a conceptual loss function constructed using another convex function?
> >
> >   2. Can the convergence rate of divergence estimation be obtained using the plug-in estimation of a probability density estimator derived from  $f$ -divergence optimization together with our upper and lower bounds?
> >
> > The plug-in estimation techniques in issue 2 follow a similar approach to what we discussed in response to a question raised by Reviewer tzbe:  for $ p $ defined as
> > $$
> >   p = \sup \left\\{ p > 1 \mid \inf_{a > 0} \lim_{x \rightarrow a} \frac{|f(x) - f(a)|}{|x - a|^p} = 0 \right\\},
> > $$
> > our upper bound is applied to the inequality $ |f(x) - f(a)| \leq |x - a|^p $.
> >
> > Additionally, we believe that these issues are significant and remain open questions for future research.
> >
> > > Can the derived rates with respect to sample size be directly compared to those established in the literature?
> >
> > We found it difficult to directly compare the established bounds in these studies with ours because their results, particularly in [1], provided upper bounds for divergence estimation, whereas our work presents bounds on error rates in DRE, a downstream task of divergence estimation. Thus, we would like to address the reviewer’s question by comparing our bounds to existing DRE bounds, although this may not directly answer the reviewer’s query.
> >
> > We believe that our bounds match the existing minimax non-parametric DRE bounds, as we clarified in response to a question from reviewer HPRf: the terms regarding data dimensionality in our upper and lower bounds are either identical to or tighter than those in the existing minimax non-parametric DRE bounds.

---

> > > ### Comment · Reviewer_eBZV · 2024-11-22
> > > **Further questions**
> > >
> > > Thank you for your detailed answers.
> > >
> > > I understand the argument of regularization. But, as far as I can see, the work [1] does not study divergence estimation but a general framework for deriving finite-sample bounds for density ratio estimators (see, e.g., Example 1 and their related work section). Did you mix it with the line of works following [zz]?
> > >
> > > [zz] Nguyen, XuanLong, Martin J. Wainwright, and Michael I. Jordan. "Estimating divergence functionals and the likelihood ratio by convex risk minimization." IEEE Transactions on Information Theory 56.11 (2010): 5847-5861.

---

> ### Author Response · Authors · 2024-11-25
>
> We sincerely thank the reviewers for their valuable comments and questions.
>
> > I understand the argument of regularization. But, as far as I can see, the work [1] does not study divergence estimation but a general framework for deriving finite-sample bounds for density ratio estimators (see, e.g., Example 1 and their related work section). Did you mix it with the line of works following [zz]?
>
> We respectfully disagree with the reviewer's assertion that [1] primarily studied a general framework for deriving finite-sample bounds for density ratio estimators rather than focusing  bounds on divergence estimation. We believe that this claim does not accurately reflect the contributions of [1] in the context of prior research.
>
> First, as outlined in Section 1.2 of [1], and particularly in Equation (4), it is evident that the primary result of [1] provides an upper bound specifically for divergence estimation.
>
> Furthermore, Example 1 in [1] enumerates the following four examples:
>
> 1. The kernel unconstrained least-squares importance fitting procedure (KuLSIF)  ([A10]): $f(u) = (u - 1)^2/2$.
>
> 2. The logistic regression ([A11], Section 7):
>   $f(u) =  u \cdot \log u - (1 + u) \cdot \log(1 + u)$.
>
> 3. The exponential function ([A12], Section 7): $f(u) = u^{-3/2}$.
>
> 4. The square loss  ([A12], Table 1): $f(u) = 1(/2\cdot u + 2)$.
>
>
> The reviewer suggests that [1] focuses on finite-sample bounds for density ratio estimators because minimizing the KuLSIF (item 1 in the above list) is equivalent to minimizing the $ L_2 $ error in DRE, as shown in Equation (3) of [1].
>
> However, since the results in [1] do not provide a comparison of convergence rates between minimizing the KuLSIF and optimizing other divergences (e.g., items 2–4 in the above list), no $ L_2 $ error bounds in DRE for $f$-divergence minimization were presented, except in the case of the KuLSIF.
>
>
> ---
> [A10] Takafumi Kanamori, Taiji Suzuki, and Masashi Sugiyama. Statistical analysis of kernelbased least-squares density-ratio estimation. Machine Learning, 86(3):335–367, 2012.
>
> [A11] Steffen Bickel, Michael Br¨uckner, and Tobias Scheffer. Discriminative learning under covariate shift. Journal of Machine Learning Research, 10(9), 2009.
>
> [A12] Aditya Menon and Cheng Soon Ong. Linking losses for density ratio and class-probability estimation. In International Conference on Machine Learning, pages 304–313. PMLR, 2016.

---

> > ### Comment · Reviewer_eBZV · 2024-11-25
> >
> > I thank the authors for the discussion. It is still hard for me to place the research in the literature. Some of my original concerns (especially regarding related research following [Menon\&Ong]) have been addressed. I raise my score to 5.

---

> ### Author Response · Authors · 2024-11-26
>
> Dear Reviewer eBZV,
>
> Thank you very much for taking the time to carefully review our manuscript and for providing valuable feedback.
>
> We are pleased to hear that the concerns raised during the review process have been satisfactorily addressed.
> Your detailed suggestions and observations regarding the related research following [Menon&Ong] and [A9] have allowed us to refine and clarify key aspects of our research, and we are grateful for your recognition of these improvements.
>
> Once again, we sincerely thank you for your time and effort in evaluating our submission.
>
> Best regards,
>
> Authors of Submission 12415

---

> > ### Author Response · Authors · 2024-11-28
> >
> > Dear Reviewer eBZV,
> >
> > We sincerely thank you for your thoughtful feedback on our manuscript.
> >
> > We would like to address an inaccuracy in one part of our previous response, which has now been corrected in Sections E.1 and E.2 of the appendix.
> >
> > In Section E.1, we have updated our discussion regarding comparisons with existing studies on the upper bounds of DRE. Initially, based on our response to Reviewer HPRf, we mentioned the existence of prior work establishing a minimax rate equivalent to our upper bound. However, we have found that no prior work has established a minimax rate equivalent to the term in our upper bound. We have clarified this point in the updated section and kindly invite you to review the revised discussion.
> >
> > Additionally, in Section E.2, we have elaborated on the relationship between the pseudo-self-concordance of losses and Assumption 3.3. Your insightful feedback has deepened our understanding of the roles of both pseudo-self-concordance and our assumption, and we have reflected this in the revised discussion.
> >
> > We would greatly appreciate it if you could take a moment to review the updated sections. Please let us know if there is anything further we can clarify or assist with.
> >
> > Sincerely,
> >
> > Authors of Submission 12415

---

### Official Review · Reviewer_tzbe · 2024-11-03

**Soundness:** 4
**Presentation:** 4
**Contribution:** 4
**Rating:** 8
**Confidence:** 4

**Summary:**

The paper proposes an approach to density ratio estimation based on $f$-divergences. This idea originated in a paper by Nguyen, Wainwright, and Jordan, who introduced this methodology, which also forms the foundation of generative adversarial networks (GANs). The dual representation of the $f$-divergence involves optimization over a function space, where the optimizer—obtained, for instance, using a neural network—provides an estimator for the likelihood ratio when evaluated on samples. The consistency of this estimator was established in Nguyen et al., and various convergence results have been proved over recent years. The main contribution of this paper is to provide upper and lower bounds on the $L^p$-error, assuming both measures have compact support and that the log-likelihood function is bi-Lipschitz. These bounds are the first of their kind.

**Strengths:**

The paper is a novel and original contribution to the field of density estimation and highly relevant to recent advances in generative modeling (for example through generative adversarial networks). To my knowledge this are the first $L^p$-error bounds in this context.
The proof of the result is very clever and relies on a representation of the $f$-divergence (called the conceptual loss function) which provides access the density ratio.  The estimates are then obtained using a "nearest neighbor" approach and by showing that the conceptual loss function and the real loss function are close to each other.  I believe that the techniques of proofs will find other applications where $f$-divergences are used (e.g. in generative modeling).  I found the presentation of the results to be excellent  both, conceptually, in the main text and, technically in the supplementary material.  Also notable is that this result is practical as the implementation of the estimator is totally straightforward using a neural network architecture.

**Weaknesses:**

+ There is a very unfortunate typo in the informal theorem 3.5 where in the lower bound  $(1-p)$ should be $(p-1)$.

+ An explanation of why the result does not depend on the $f$-divergence should be provided.  I believe this is because of the Lipschitz assumption on the log-likelihood and the compactness of the support. In that way the behavior of $f$ at $0$ and at $\infty$ are irrelevant.

+ Some head-to-head experimental comparison with other density ratio estimators would have been helpful.

**Questions:**

+ Can you explain why the choice of $f$ in the divergence does not matter?  Does it only change the constant?

+ What are the implications of your results on the analysis of GANs?

+ What are the implications of your results neural estimation of $f$-divergence (see e.g. the recent https://jmlr.org/beta/papers/v23/21-1212.html)?

---

> ### Author Response · Authors · 2024-11-21
>
> We sincerely appreciate your thoughtful review and valuable suggestions. Your comments have helped us to identify areas for improvement, and we have made the necessary changes to address them in this revised version. Please find our detailed responses below.
>
> > Can you explain why the choice of in the $f$-divergence does not matter? Does it only change the constant?
>
> We believe that the reason our bounds are independent of the choice of the $f$-divergence is that the convergence rates of the upper and lower bounds of DRE are dominated by the data dimensionality, leading to convergence rates much larger than the statistically optimal rate (i.e., $1/\sqrt{N}$) when $d \geq 3$.
>
> Theorem C.9 in the appendix shows that the convergence rate for $f$-divergence optimization is $1/\sqrt{N}$, with the constant $E_P[f''(dQ/dP)]$ appearing as a coefficient in this rate. Thus, the choice of $f$-divergence is expected to influence the convergence rate for DRE when the data dimensionality is one or two.
>
> However, under our assumptions on the data, we have not identified any definitive reasons why our bounds are dominated by the data dimensionality. In other words, we believe that the existing $f$-divergence optimization approach could potentially be further refined, leading to convergence rates that are not dominated by the data dimensionality, such as $1/\sqrt{N}$ in an extremely fortunate case.
>
> > What are the implications of your results on the analysis of GANs?
>
> As we addressed a question from reviewer mVFa, optimization of $f$-GANs could benefit from adjusting the base noise distribution to data.
>     The accuracy of generative models could be improved by fitting the base parametric models to the data in terms of KL-divergence minimization.
>
> Additionally, although careful theoretical considerations regarding the relationship between GAN's metrics, such as the inception score, and the $L_p$ error for DRE are required, the Truncation Trick proposed in W-GAN ([A8]) might be partially interpreted as a reduction in the KL-divergence between the training data and the base distribution.
>
> [A8] Brock, A. (2018). Large Scale GAN Training for High Fidelity Natural Image Synthesis. arXiv preprint arXiv:1809.11096.
>
> > What are the implications of your results neural estimation of $f$-divergence (see e.g. the recent https://jmlr.org/beta/papers/v23/21-1212.html)?
>
> The reviewer’s question is extremely important. We appreciate the valuable suggestions.
>
> One implication of our study on the neural estimation of $ f $-divergence is that a term involving the exponential of KL divergence of data might emerge in its upper and lower bounds, reflecting the strength of the convexity of the $ f $-divergence.
>
> If there exists $1 \leq p  < \infty$ such that $|f(x) - f(a)| \geq |x - a|^p$, it might be possible to derive the following expression using this inequality:
>
>
> $$
>     \left\\{ E_{P}\left|f\left(\hat{r}(\mathbf{x})\right) - f\left(\frac{dP}{dQ}(\mathbf{x})\right)\right| \right\\}^{1/p}
>     \ge \left\\{E_{P} \left\| \hat{r}(\mathbf{x}) - \frac{dP}{dQ}(\mathbf{x}) \right\|^p \right\\}^{1/p}
>     \ge \frac{1}{N^{1/d}} \cdot \left\\{ \frac{1}{L} \cdot e^{\frac{(p - 1)}{p} \cdot KL(P \| Q) - 1} - K \cdot \mathrm{diag}(\Omega) \right\\}, \tag{B1}
> $$
>
> where $\hat{r}$ represents the density ratio estimator obtained from the $f$-divergence optimization.
>
> More specifically, $p$ is defined as a constant greater than or equal to 1 that satisfies the following conditions:
>
> $$
>     p = \inf \left\\{ p > 1 \mid \inf_{a > 0} \lim_{x \rightarrow a} \frac{|f(x) - f(a)|}{|x - a|^p} \neq 0 \right\\},
> $$
>
> or equivalently,
>
> $$
>     p = \inf \Big\\{ p > 1 \mid \inf_{a > 0} \left( {f'(a)}^p \cdot {f(a)}^{1 - p} \right)  \neq 0 \Big\\}.
> $$
>
>
> For example, in the case of the KL divergence, $f(u) = - \log u$, we have $p = 1$. In the case of the Pearson $\chi^2$ divergence, $f(u) = (1 - u)^2$, we have $p = 2$.
>
> Thereafter, by applying our lower bound to the inequality $|f(x) - f(a)| \geq |x - a|^p$, Equation (B1) is obtained.

---

> > ### Author Response · Authors · 2024-11-28
> >
> > Dear Reviewer tzbe,
> >
> > In the revised version of our manuscript, we have addressed the typographical errors you kindly pointed out, which were extremely helpful, as we had not noticed them ourselves.
> >
> > We apologize for the delay in submitting the revised manuscript, and for the time taken to express our gratitude for your helpful feedback.
> >
> > If there is anything further we can assist with or clarify, please do not hesitate to let us know. Once again, thank you for your constructive feedback and invaluable support.
> >
> > Sincerely,
> >
> > Authors of Submission 12415

---

### Official Review · Reviewer_mVFa · 2024-11-04

**Soundness:** 3
**Presentation:** 3
**Contribution:** 3
**Rating:** 5
**Confidence:** 3

**Summary:**

* The paper provides new upper and lower bounds on the Lp errors of a specific density ratio estimator. (Equations 4,5, and 6).
This estimator is constructed by minimizing an f-divergence based loss function.

These bounds provide new insights about how the dimensionality of the data, and the KL divergence between the distributions affect the error of the estimator.

**Strengths:**

* As far as I know, the derived lower and upper bounds on the density ratio estimator are new.
* The paper is well-written, and easy to follow.
* The presentation is clear.

**Weaknesses:**

The theoretical results are interesting.
In my opinion, what is missing from the paper is to show that these theoretical results can make a difference in some important applications. The paper contains a nice list of motivations for density ratio estimation, but the experimental section only contains some simple toy problems, and therefore the impact of the theory in real applications is not perfectly clear.

**Questions:**

* Equation 5 (the upper bound) contains the expectation operator on the left-hand side. This expectation is missing from the lower side (Eq 4). Is it a typo?
* In Equations 4 and 5, is it possible to create bounds for dimensions 1, and 2 as well?
* The upper and lower bounds are only derived for a specific density ratio estimator. Is it known how the convergence rate of this estimator compares to other density ratio estimators?
* How can we use these theoretical results in applications?

---

> ### Author Response · Authors · 2024-11-21
>
> Thank you for your insightful comments and suggestions. We have carefully reviewed and incorporated your feedback into the revised manuscript to enhance its clarity and quality. A detailed point-by-point response is provided below.
>
> > Equation 5 (the upper bound) contains the expectation operator on the left-hand side. This expectation is missing from the lower side (Eq 4). Is it a typo?
>
> No, it is not a typo.
> In our evaluation of the lower bound, Equation (5), some randomness remained.
> By taking the expectation over the lower bound, we have eliminated that randomness.
> Specifically, in the proof of Theorem 4.4 (Theorem C.17 in the appendix),
> we observed that Equation (91) asymptotically converges to the Poisson distribution.
> To evaluate the lower bound as a deterministic value, we performed the integration.
>
> > In Equations 4 and 5, is it possible to create bounds for dimensions 1, and 2 as well?
>
> No, we believe that the same bounds derived in this study do not hold for these dimensions. Instead, bounds of $1/\sqrt{N}$ might be obtained. Additionally, we believe that bounds for dimensions 1 and 2 might be more complex than the upper and lower bounds presented in this study, since $1/\sqrt{N}$ is a statistically optimal rate. For example, terms related to $f$-divergence might emerge in the coefficients of these bounds.
>
> > The upper and lower bounds are only derived for a specific density ratio estimator. Is it known how the convergence rate of this estimator compares to other density ratio estimators?
>
> As we addressed a question from reviewer HPRf, the terms regarding data dimensionality in our upper and lower bounds are identical to or tighter than the existing minimax non-parametric DRE bounds. However, to the best of our knowledge, no prior work has provided the same term as ours concerning the exponential of the KL-divergence.
>
>
> > How can we use these theoretical results in applications?
>
> The following two apprications can be considered:
> 1.  Selecting a benchmark index for evaluating DRE methods:
>   When evaluating the accuracy of DRE methods using synthetic datasets, the root mean squared error (RMSE) or mean squared error (MSE) should be used rather than the mean absolute error (MAE). Prior works did not take into account the differences in their behavior regarding the KL divergence of the datasets. For example, [A3] used MAE, whereas [A4] used MSE.
>
> 2.  Fitting the distribution of base noise for f-GAN and Normalizing Flow:
>     Optimization of $f$-GANs ([A5]) could benefit from adjusting the base noise distribution to data. Since the optimization of $f$-GANs is equivalent to DRE by optimizing the $f$-divergence ([A6]), the accuracy of generative models could be improved by fitting the base parametric models to the data in terms of KL divergence minimization (i.e., likelihood maximization). The same approach can be applied to the base models in Normalizing Flow ([A7]).
>
> [A3] Kimura, M., & Bondell, H. (2024). Density Ratio Estimation via Sampling along Generalized Geodesics on Statistical Manifolds. arXiv preprint arXiv:2406.18806.
>
> [A4] Kato, M., & Teshima, T. (2021, July). Non-negative bregman divergence minimization for deep direct density ratio estimation. In International Conference on Machine Learning (pp. 5320-5333). PMLR.
>
> [A5] Nowozin, S., Cseke, B., & Tomioka, R. (2016). f-gan: Training generative neural samplers using variational divergence minimization. Advances in neural information processing systems, 29.
>
> [A6] Uehara, M., Sato, I., Suzuki, M., Nakayama, K., & Matsuo, Y. (2016). Generative adversarial nets from a density ratio estimation perspective. arXiv preprint arXiv:1610.02920.
>
> [A7] Papamakarios, G., Nalisnick, E., Rezende, D. J., Mohamed, S., & Lakshminarayanan, B. (2021). Normalizing flows for probabilistic modeling and inference. Journal of Machine Learning Research, 22(57), 1-64.

---

> > ### Author Response · Authors · 2024-11-28
> >
> > Dear Reviewer mVFa,
> >
> > We would like to extend our sincere gratitude for your valuable feedback and time spent reviewing our manuscript.
> >
> > Upon revisiting our previous responses, we found that our reply contained inaccuracies. Specifically, besed on our response to Reviewer HPRf, we metioned the existence of prior work establishing a minimax rate equivalent to our upper bound.
> > However, we found no prior work establishing a minimax rate equivalent to the term in our upper bound.
> >
> > We have addressed and corrected this error in the revised manuscript, particularly in Section E.1 of the appendix, and would appreciate it if the reviewer could kindly take a look at the updated disccussion.
> >
> > If there is anything further we can assist with or clarify, please do not hesitate to let us know. Thank you once again for your constructive feedback and support.
> >
> > Sincerely,
> >
> > Authors of Submission 12415

---

> > > ### Comment · Reviewer_mVFa · 2024-12-02
> > > **Re: New Official Comment**
> > >
> > > Dear Authors,
> > >
> > > Thanks for your detailed reviews. I still believe that to show the significance of the work, the paper should have some numerical experiments that can demonstrate how these results can make a difference in some important real applications. Therefore, I keep my original scores.

---

> > > > ### Author Response · Authors · 2024-12-03
> > > >
> > > > Dear Reviewer mVFa,
> > > >
> > > > Thank you for your valuable feedback and insightful suggestions regarding our submission. We sincerely apologize for the delay in our response and greatly appreciate the time and effort you have devoted to reviewing our work.
> > > >
> > > > In response to your comments, we have added a brief section discussing the potential applications highlighted by our findings. These revisions have been incorporated into the updated manuscript to enhance the quality and clarity of our research.
> > > >
> > > > Thank you once again for your constructive review.
> > > >
> > > > Sincerely,
> > > > Authors of Submission 12415

---

### Official Review · Reviewer_HPRf · 2024-11-08

**Soundness:** 2
**Presentation:** 2
**Contribution:** 3
**Rating:** 6
**Confidence:** 3

**Summary:**

This paper establishes L_p​ error bounds of the density ratio in Density Ratio Estimation under f-divergence loss. The bounds scale with $N^{-1/d}$ and certain p-Renyi divergence, but is independent from the specific choice of fff-divergence.

**Strengths:**

The problem of uncertainty quantification in density ratio estimation is an important fundamental question. This work contributes a new understanding of its theoretical limits by deriving L_p​ error bounds of the density ratio. The nearly matching upper and lower bounds are a strong point, and the independence on the specific f-divergence within the loss function is also intriguing.

**Weaknesses:**

The bound’s dependence on $N^{−1/d}$ means that in high-dimensional spaces (where d is large), the error bound becomes practically useless. There is also a lack of downstream application or implication of the theoretical results. As a whole, this raises questions on the meaningfulness of the bounds.

The experiments presented exhibit high variance, which limits the interpretability and robustness of the empirical validation.

**Questions:**

How do these bounds compare to other known bounds in DRE or related density estimation methods?

---

> ### Author Response · Authors · 2024-11-21
>
> We are grateful for your thorough review and constructive remarks. Your feedback has been instrumental in refining our work, and we have addressed each of your points in the updated manuscript. Please see our responses below.
>
> > How do these bounds compare to other known bounds in DRE or related
>     density estimation methods?
>
> The terms regarding data dimensionality in our upper and lower bounds are identical to or tighter than the existing minimax non-parametric DRE bounds. However, to the best of our knowledge, no prior work has provided the same term as ours concerning the exponential of the KL-divergence.
>
> From existing minmax DRE upper bounds, [A1] provided $L_1$ and $L_2$ upper bounds in DRE for the $M$-th nearest neighbor-based method.
> According to the results in [A1], the $L_1$ upper bound is $N^{-1/d}$ for all $M \geq 1$ (Theorem 4.3, case (i), p.11 in [A1]), which is the same as the term regarding the data dimensionality in our upper bound. Furthermore, its $L_2$ upper bound is $N^{-1/(2+d)}$ as $M$ increases along with the sample size (Theorem 4.3, case (iii), p.11 in [A1]).
>
> In addition, in [A2], for another example, an $L_1$ upper bound for probability density estimation was presented as $N^{-1/(2+d)}$ for a nearest neighbor-based method.
>
> As an example of DRE minimax lower bounds in prior works, an $L_2$ error bound of $N^{-1/d}$ was provided (e.g., Proposition 4.1, p.11 in [A1]).
> However, minimax lower bounds do not represent the true lower bounds of DRE, as we noted in lines 47–48 in this study.
>
> [A1] Lin, Z., Ding, P., & Han, F. (2023). Estimation based on nearest neighbor matching: from density ratio to average treatment effect. Econometrica, 91(6), 2187-2217.
>
> [A2] Kpotufe, S. (2017, April). Lipschitz density-ratios, structured data, and data-driven tuning. In Artificial Intelligence and Statistics (pp. 1320-1328). PMLR.

---

> > ### Comment · Reviewer_HPRf · 2024-11-26
> >
> > Thank you for your response. I appreciate the clarification from the authors and will keep my score.

---

> > > ### Author Response · Authors · 2024-11-27
> > >
> > > Dear Reviewer HPRf,
> > >
> > > Thank you very much for taking the time to carefully review our manuscript and for addressing the concerns regarding it. We greatly appreciate your constructive feedback, which has been instrumental in improving our work.
> > >
> > > Currently, we are in the process of preparing a revised version of the manuscript, incorporating the suggestions received from other reviewers. This includes addressing typographical errors and other minor corrections that were pointed out.
> > >
> > > In response to your and other reviewers' suggestions regarding the lack of comparison with other known DRE bounds, we plan to include a brief discussion based on our response to your review. This additional content will be placed in the appendix of the revised manuscript. We apologize for the delay in submitting the revised version.
> > >
> > > Once again, we sincerely thank you for your time and effort in evaluating our submission.
> > >
> > > Best regards,
> > >
> > > Authors of Submission 12415

---

> > > > ### Author Response · Authors · 2024-11-28
> > > >
> > > > Dear Reviewer HPRf,
> > > >
> > > > We sincerely thank the reviewer for their thoughtful feedback on our manuscript.
> > > >
> > > > We would like to address an inaccuracy in one part of the review, which which has been corrected in Section E.1 of the appendix. Specifically, we found no prior work establishing a minimax rate equivalent to the term in our upper bound.
> > > >
> > > > We have clarified this point in Section E.1 of the appendix and would appreciate it if the reviewer could kindly take a look at the updated discussion.
> > > >
> > > > Sincerely,
> > > >
> > > > Authors of Submission 12415

---

### Author Response · Authors · 2024-11-28

We sincerely thank the reviewers for their valuable comments and suggestions during the review process, which have greatly contributed to improving our work. We have carefully revised our paper based on the discussions and feedback provided.

We deeply apologize for the delay in submitting the revised version.

Below, we outline the key revisions made to the manuscript in response to your comments:

1. Comparison with existing DRE bounds (Section E.1 in the appendix)
   - We have added a detailed comparison between the results of this study and existing DRE bounds in Section E.1 in the appendix. We have carefully revised our comparison with prior work and found that our upper bounds are tighter than the existing minimax bounds in both $L_1$ and $L_2$ errors.
   - Additionally, we identified additional prior works which are particularly relevant to this study and have included them as references.

2. Remarks on Assumption 3.3 and related assumptions in prior work (Section E.2 in the appendix)
   - In Section E.2 in the appendix, we have included a detailed discussion on the relationship between Assumption 3.3 and the pseudo-self-concordance of losses to address the concern raised by Reviewer eBZV.
   - The discussion has been refined, and we believe that it provides a more rigorous and clearer explanation regarding this issue.

3. Applications of this study (Section E.3 in the appendix)
   - We have added a discussion on potential applications suggested by this research in Section E.3 in the appendix based on the discussion during the review process. This new section elaborates on how the findings could contribute to related fields.

4. Formatting and Minor Revisions:
   - We have updated the Related Work in Section 1 by adding important relevant citations.
   - We have corrected grammatical errors and typos for improved clarity. While the main content of the paper remains unchanged, we have also made adjustments to the formatting due to space constraints, such as relocating figures in the manuscript.

---

### Meta-Review · Area_Chair_r2d9 · 2024-12-20

**Metareview:**

The paper provides (nearly matching) upper and lower bounds on $L_p$ error in density ration estimation (DRE) for a natural class of $f$-divergence-based estimators. The bounds are independent of the specific $f$-divergence used, which is intriguing. They are instead dominated by the dimensional dependence, scaling with $1/N^{1/d}$. The proof techniques appear novel and interesting. The main reservations about the paper stemmed from the limitations of the $1/N^{1/d}$ scaling in high dimensions, lack of proper comparison with prior work, and lack of applications. The latter two points were mostly addressed during the rebuttal period, though the authors are advised to further expand on those discussions in the next revision. For the first point, I believe there is value to the results, despite such scaling, for the following reasons. First, given that the considered estimators are natural and commonly used, it is helpful to understand their limitations through the provided lower bounds. Second, as was revealed during the discussion, there are no alternative upper bounds that provide a better scaling with the dimension.

**Additional Comments On Reviewer Discussion:**

The main reservations about the paper stemmed from the lack of applications/insufficient experiments and insufficient comparison to prior work. The authors added these points in the rebuttal phase and hopefully can further expand on in the next revision. The reviewers with lower scores were not opposed to acceptance following the discussion period (though were still not enthusiastic). Reviewer tzbe championed the paper, based on its technical novelty, which seemed convincing to me.

---

### Decision · Program_Chairs · 2025-01-22

Accept (Poster)